# Can MLLMs Perform Text-to-Image In-Context Learning?

Yuchen Zeng[*1] , Wonjun Kang[*2,3] , Yicong Chen[1] , Hyung Il Koo[2,4] , and Kangwook Lee[1]

[1]University of Wisconsin-Madison      [2]FuriosaAI
[3]Seoul National University      [4]Ajou University

## Abstract

The evolution from Large Language Models (LLMs) to Multimodal Large Language Models (MLLMs) has spurred research into extending In-Context Learning (ICL) to its multimodal counterpart. Existing such studies have primarily concentrated on image-to-text ICL. However, the Text-to-Image ICL (T2I-ICL), with its unique characteristics and potential applications, remains underexplored. To address this gap, we formally define the task of T2I-ICL and present **CoBSAT**, the first T2I-ICL benchmark dataset, encompassing ten tasks. Utilizing our dataset to benchmark six state-of-the-art MLLMs, we uncover considerable difficulties MLLMs encounter in solving T2I-ICL. We identify the primary challenges as the inherent complexity of multimodality and image generation, and show that strategies such as fine-tuning and Chain-of-Thought prompting help to mitigate these difficulties, leading to notable improvements in performance. Our code and dataset are available at `https://github.com/UW-Madison-Lee-Lab/CoBSAT`.

## 1 Introduction

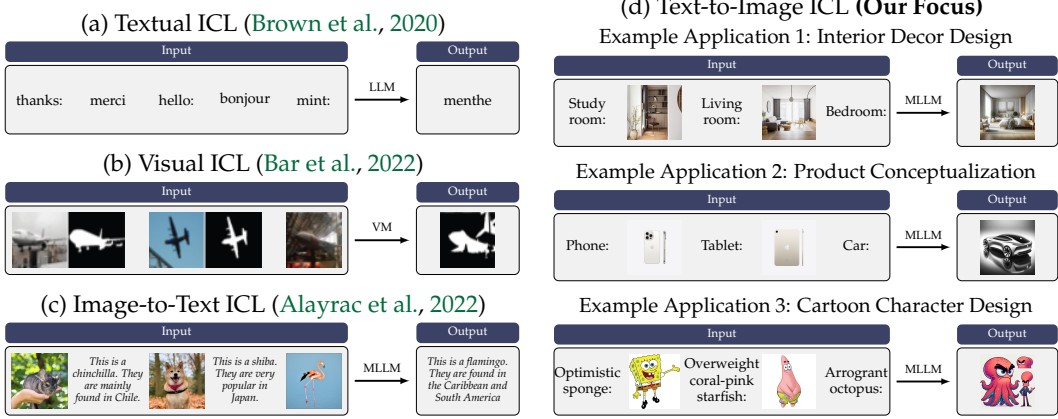

Figure 1: **Comparison of various In-Context Learning (ICL) settings.** (a) Textual ICL, where both the input and output in each example are textual. (b) Visual ICL, where both input and output in each demonstration are presented as images. (c) Image-to-Text ICL (I2T-ICL), featuring images as input and texts as output in each demonstration. (d) Text-to-Image ICL (T2I-ICL, our focus), which involves text input and image output in each demonstration. T2I-ICL introduces greater complexities and presents different potential applications. The examples in (d) provide three potential applications of T2I-ICL, with the output generated using ChatGPT-4 (OpenAI, 2023) with DALL-E 3 (Betker et al., 2023) capabilities.

---

[*]Equal contribution. Emails: `yzeng58@wisc.edu, kangwj1995@furiosa.ai`

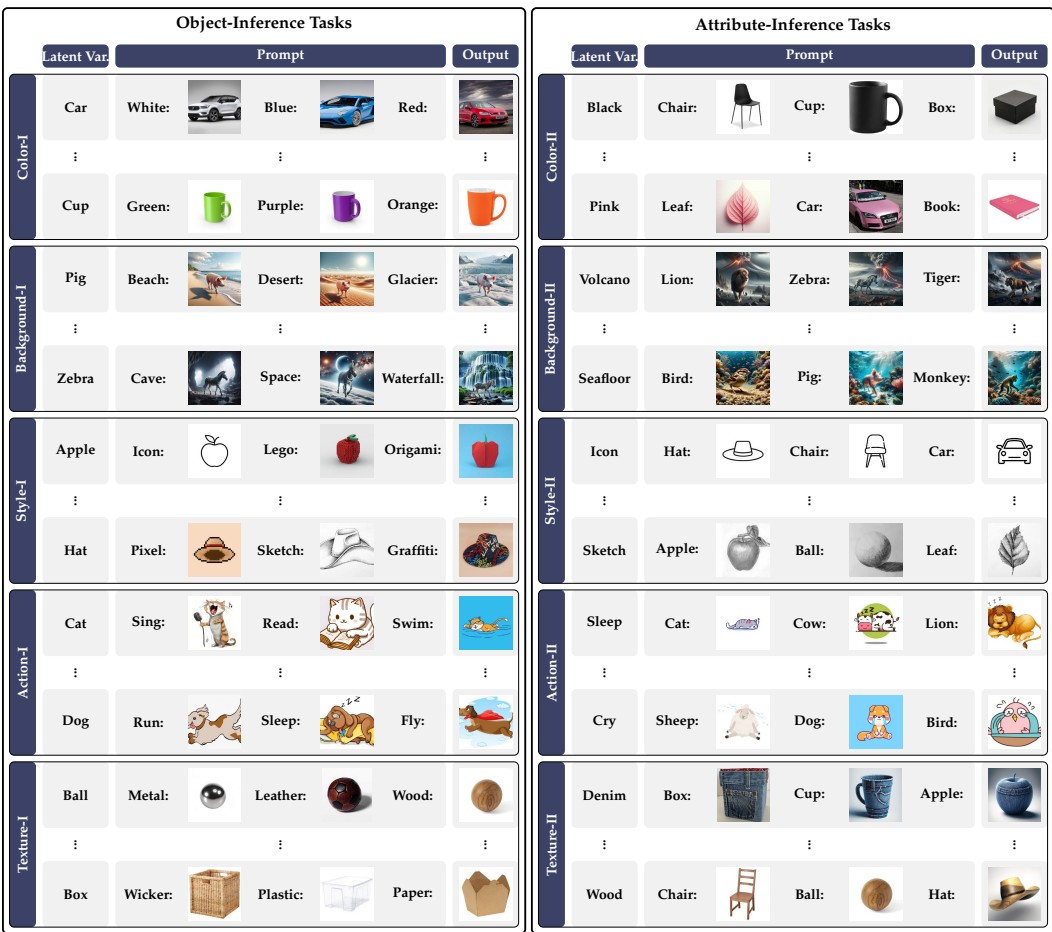

Figure 2: **Overview of example prompts in the CoBSAT benchmark.** CoBSAT covers five themes: color, background, style, action, and texture, each with two different emphases: object-inference and attribute-inference. In object-inference tasks, the attribute (e.g., color) is directly provided in the textual input, and the model is required to infer the object (e.g., car) from the images. In other words, the latent variable (denoted as "Latent Var." in the figure) of object-inference tasks is the object. Conversely, in attribute-inference tasks, the object is specified in the text. The model is tasked with inferring the attribute from the images in the demonstrations, i.e., the attribute serves as the latent variable in attribute-inference tasks.

In the rapidly evolving landscape of artificial intelligence, Multimodal Large Language Models (MLLMs) (Ge et al., 2023b; Koh et al., 2023; Sun et al., 2023c; OpenAI, 2023; Liu et al., 2023a; Bai et al., 2023b; Gemini Team Google: Anil et al., 2023; Li et al., 2023; Anthropic, 2024) extend the frontier of Large Language Models (LLMs) (Devlin et al., 2019; Radford et al., 2019; Brown et al., 2020; OpenAI, 2023; Touvron et al., 2023) by handling not only text but also images, videos, and audio. This multimodal capability enables MLLMs to undertake complex tasks, integrating visual, auditory, and textual cues. The versatility of MLLMs makes them powerful tools in AI, offering context-rich interpretations across various domains.

In-Context Learning (ICL) (see Figure 1(a)) is a prevalent technique that enables predictions based on context through a sequence of input-output pairs, termed *demonstrations*, without requiring any model parameter updates. This capability was initially identified and applied by Brown et al. (2020) and has since become a widely used standard prompt engineering method to enhance LLM inference performance for various downstream tasks. This method has been applied in computer vision to produce output images contextually aligned with

provided image-image pair examples, termed Visual ICL (V-ICL) (see Figure 1(b)) (Bar et al., 2022; Wang et al., 2023a). In another development, Tsimpoukelli et al. (2021) introduced Multimodal ICL (M-ICL) for the first time for image-to-text generation tasks, including applications such as visual question answering and image captioning. Unlike ICL, which is exclusively text-focused, and V-ICL, which is solely image-oriented, M-ICL uses demonstrations that incorporate samples from two modalities.

The majority of existing M-ICL work (Tsimpoukelli et al., 2021; Alayrac et al., 2022; Monajatipoor et al., 2023; Chen et al., 2023b; Zhao et al., 2023) has mainly centered on the performance of image-to-text tasks, the goal of which is transforming high-dimensional, image-based input into low-dimensional, text-based output. However, when the roles are reversed, models can exhibit significantly different performance characteristics. To distinguish between these two tasks, we refer to M-ICL for image-to-text generation as *Image-to-Text ICL* (I2T-ICL) (see Figure 1(c)), and M-ICL for text-to-image generation as *Text-to-Image ICL* (T2I-ICL) (see Figure 1(d)), with the latter being the focus of our work. It is important to note that potential applications of T2I-ICL are completely different from I2T-ICL, which include areas like product design and personalized content creation.

**Our Contributions.**    We summarize our main contributions as follows.

- **Identifying an Important Problem: T2I-ICL.** Our work first identifies the important yet underexplored ICL setting on text-to-image generation, termed T2I-ICL.

- **Introducing the CoBSAT Benchmark.** To systematically assess the T2I-ICL capability of MLLMs, we introduce a comprehensive benchmark featuring ten tasks across five different themes — **Co**lor, **B**ackground, **S**tyle, **A**ction, and **T**exture, which is named as **CoBSAT** (see Figure 2).

- **Benchmarking MLLMs in T2I-ICL.** We utilize our dataset to evaluate the T2I-ICL capabilities of ten state-of-the-art MLLMs. This includes Emu (Sun et al., 2023c), GILL (Koh et al., 2023), SEED-LLaMA (Ge et al., 2023b), Qwen-VL (Bai et al., 2023b), Gemini (Gemini Team Google: Anil et al., 2023), Claude (Anthropic, 2024), and GPT-4V (OpenAI, 2023), which are elaborated upon in the main paper, alongside Emu2 (Sun et al., 2023a), LLaVA-1.5 (Liu et al., 2023a), and LLaVA-NeXT (Liu et al., 2024), detailed in the appendix. We observe that the T2I-ICL performance of these models is significantly influenced by their respective training paradigms. Among them, SEED-LLaMA, Qwen-VL, Gemini, Claude, and GPT-4V demonstrate the capability to perform T2I-ICL. Yet, except for Gemini, their accuracy rates hover around or fall below 60% in most scenarios.

- **Understanding Challenges in T2I-ICL.** We then investigate the key factors contributing to the underperformance of MLLMs in T2I-ICL. Our findings point to two principal challenges: (i) the intrinsic complexity involved in processing multimodal data, and (ii) the inherent difficulties associated with the task of image generation.

- **Enhancing MLLMs' T2I-ICL Capabilities.**  To augment MLLMs' T2I-ICL capabilities, we delve into various potential techniques. Our study demonstrates that fine-tuning and Chain-of-Thought (CoT) (Wei et al., 2022) significantly boost T2I-ICL performance.

## 2   Related Works

**Unimodal ICL.**    Ever since Brown et al. (2020) demonstrated that language models are in-context learners (see Figure 1(a)), there has been substantial interest in comprehending this capability, both empirically (Liu et al., 2022; Min et al., 2022b; Chen et al., 2022; Mishra et al., 2022; Lampinen et al., 2022; Garg et al., 2022; Hendel et al., 2023) and theoretically (Xie et al., 2022; Wies et al., 2023; Akyürek et al., 2023; Von Oswald et al., 2023; Bai et al., 2023c; Ahn et al., 2023; Zhang et al., 2023b). Textual ICL (T-ICL) enables the adaptation of LLMs to downstream tasks simply by providing a few illustrative examples, bypassing any need for updating model parameters. The concept of V-ICL is then employed in computer vision, starting with the introduction of visual prompts (see Figure 1(b)). The pioneering works by Bar et al. (2022); Wang et al. (2023a) propose to automatically generate output images that are contextually aligned with provided examples. Specifically, Bar et al. (2022) developed

a method that combines three images - an example input, its corresponding output, and a query - into a single composite image. In this layout, the example input is placed in the upper left, the example output in the upper right, the query image in the bottom left, and the bottom right patch is left blank for output construction via an image inpainting model. Bar et al. (2022) demonstrated the effectiveness of V-ICL in tasks like edge detection, colorization, inpainting, etc. Unlike T-ICL and V-ICL which are limited to handling unimodal inputs, M-ICL integrates demonstrations encompassing both text and images.

**Image-to-Text ICL.** Most existing work on M-ICL focuses on image-to-text generation, i.e., I2T-ICL (Tsimpoukelli et al., 2021; Alayrac et al., 2022; Monajatipoor et al., 2023; Chen et al., 2023b; Zhao et al., 2023). In particular, Tsimpoukelli et al. (2021) were the first to extend ICL from the text domain to the multimodal domain, focusing on image-to-text generation such as visual question-answering (see Figure 1(c)). Alayrac et al. (2022) introduced Flamingo, an MLLM that achieves good performance in a variety of image and video understanding tasks using I2T-ICL with 32 demonstrations, implying the efficacy of I2T-ICL in performance enhancement in their model. Concurrently, efforts have been made to develop datasets specifically designed for evaluating the I2T-ICL capability of MLLMs (Zhao et al., 2023).

**Text-to-Image ICL.** There are limited attempts to evaluate MLLMs based on their T2I-ICL capabilities. A notable exception is concurrent research by Sun et al. (2023a). They evaluated the performance of their model on T2I-ICL with DreamBooth dataset (Ruiz et al., 2023). However, it is important to note that the DreamBooth dataset, primarily developed for fine-tuning models to modify image contexts, was not specifically designed for T2I-ICL applications, making it more challenging and mostly focusing on background altering. The complexity, as seen in style transfer examples that emulate artists like Vincent van Gogh or Michelangelo, can pose challenges even for human interpretation.

**MLLMs.** Recently, there has been a surge in the release of MLLMs, which are designed to address more challenging multimodal tasks, thereby enabling the perception of images, videos, and audios (Li et al., 2022; Alayrac et al., 2022; Hao et al., 2022; Laurençon et al., 2023; Huang et al., 2023b; Peng et al., 2023b; Li et al., 2023; Ge et al., 2023b; Koh et al., 2023; Zhu et al., 2023a; Sun et al., 2023c; Zheng et al., 2023a; OpenAI, 2023; Liu et al., 2023b;a; Bai et al., 2023b; Sun et al., 2023a; Driess et al., 2023; Gemini Team Google: Anil et al., 2023; Borsos et al., 2023; Huang et al., 2023a; Chen et al., 2023a; Zhang et al., 2023a; Anthropic, 2024).

Since our main focus is T2I-ICL, we only consider models capable of processing both text and multiple images. We consider two types of MLLMs: (i) proficient in generating both text and images, including Emu (Sun et al., 2023c), Emu2 (Sun et al., 2023a), GILL (Koh et al., 2023), and SEED-LLaMA (Ge et al., 2023b), and (ii) those limited to text generation, including GPT-4V (OpenAI, 2023), LLaVA-1.5 (Liu et al., 2023b), LLaVA-NeXT (Liu et al., 2024), Gemini (Gemini Team Google: Anil et al., 2023), Claude (Anthropic, 2024) and Qwen-VL (Bai et al., 2023b). For text-only MLLMs, we evaluate their capacity to infer visual outputs by prompting them to describe the anticipated image. Conversely, for MLLMs capable of image generation, we not only elicit image outputs but also ask for descriptive text, ensuring an apple-to-apple comparison with text-only models.

Owing to page constraints, we provide a more detailed overview of related works in Sec. B.

## 3 Dataset: CoBSAT

We start by describing the definition of in-context learning. Consider a task with data $(x, y)$, where input $x \in \mathcal{X}$, output $y \sim f_\theta(x)$, where distribution $f_\theta$ is parameterized by latent variable $\theta \in \Theta$. We denote the model by $M$. For in-context demonstrations, we are given $N$ input-output pairs $\{(x_n, y_n)\}_{n=1}^{N}$ and one test query $x_{N+1}$. In-context learning make the prediction by incorporating these demonstrations $\{(x_n, y_n)\}_{n=1}^{N}$ and the test query $x_{N+1}$ in the prompt. The prediction made by model $M$ is formulated as $\hat{y}_{N+1} = M(x_1, y_1, x_2, y_2, \ldots, x_N, y_N, x_{N+1})$. In this work, we mainly focus on scenarios where the input $x$ is textual data and output $y$ corresponds to an image. We use notation

[Image: **description**] to denote an image corresponding to the text description. For instance, [Image: **red car**] refers to an image depicting a red car.

**Dataset Structure.** We begin by outlining the structure of our dataset, which evaluates whether models are capable of learning the mapping from textual input to visual output, based on the given in-context demonstrations. For instance, task Color-I in our experiment involves generating an image of an object of a particular color, where the object to be drawn is not explicitly stated in the text query $x_{N+1}$. The information of the object is instead implicitly contained in $\theta$ (and hence in $y_i$'s since $y_i \sim f_\theta(x_i)$ for all $i = 1, \ldots, N$), which can be learned from the demonstrations. An example prompt when $\theta =$ "car" is

$$\text{"} \overbrace{\underbrace{\text{red:}}_{x_1} \; \underbrace{\text{[Image: red car]}}_{y_1}}^{\text{example 1}} \; \overbrace{\underbrace{\text{blue:}}_{x_2} \; \underbrace{\text{[Image: blue car]}}_{y_2}}^{\text{example 2}} \; \overbrace{\underbrace{\text{pink:}}_{x_3}}^{\text{query}} \text{."}$$

Ideally, MLLMs can learn the object $\theta$ from the context, and generate an image of a pink car.

CoBSAT comprises ten tasks, divided into two categories: (i) *object-inference tasks*, which give the attributes (e.g., color, texture) in the text input and require identifying objects (e.g., car, cup) from images, and (ii) *attribute-inference tasks*, which provide the object to be drawn in the text input but require identifying the common attribute from the images (see Figure 2). Each task has predefined lists for text inputs and latent variables, denoted as $\mathcal{X}$ and $\Theta$, each containing ten distinct items. For instance, in the Color-I task, the predefined list for the latent variable (i.e., the object) is $\Theta = \{$leaf, hat, cup, chair, car, box, book, ball, bag, apple$\}$, and the predefined list for the text input (i.e., the attribute) is $\mathcal{X} = \{$yellow, white, red, purple, pink, orange, green, brown, blue, black$\}$. The predefined lists for all tasks are provided in Sec. C. In our experiment, for each specified number of shots (i.e., 2, 4, 6, 8 in our experiments), we create 1,000 prompts per task. This is accomplished by randomly selecting a latent variable $\theta$ from the predefined list $\Theta$ and a sequence of textual inputs $(x_n)_{n=1}^{N+1}$ from $\mathcal{X}^{N+1}$. Then, we pair each textual input $x_n$ with the corresponding image $y_n \sim f_\theta(x_n)$ to instruct in-context demonstrations.

**Data Collection.** For each task, we gather one image for every possible pairing of the textual input $x \in \mathcal{X}$ and latent variable $\theta \in \Theta$, resulting in $|\mathcal{X}| \times |\Theta| = 10 \times 10 = 100$ images for each task. For instance, for task Color-I, we collect an image of a red car to correspond to the case where $x =$ "red" and $\theta =$ "car," and likewise for other images. It is noteworthy that the tasks with the same theme, such as Color-I (object-inference task) and Color-II (attribute-inference task), share the same images. In addition, all object lists and attribute lists, along with the images, are carefully selected so that LLaVA can correctly identify the specified objects and the corresponding attributes (i.e., color, background, texture, action, and style) of the images. This ensures an appropriate level of difficulty for T2I-ICL tasks and allows LLaVA to perform reliable evaluations on generated images. In total, we collect 500 images from the web and DALL-E 3 (Betker et al., 2023). We then construct in-context prompts for 2, 4, 6, and 8 shots as previously described, with each shot resulting in 10,000 prompts.

## 4 Methodology

**MLLMs.** In our study, we assess the performance of models in T2I-ICL, specifically Emu (Sun et al., 2023c), Emu2 (Sun et al., 2023a), SEED-LLaMA (Ge et al., 2023b), and GILL (Koh et al., 2023), which can generate images. In addition to image generation scenarios, we instruct the text-only generation models — Qwen-VL (Bai et al., 2023b), LLaVA-1.5 (Liu et al., 2023a), LLaVA-NeXT (Liu et al., 2024), Gemini (Gemini Team Google: Anil et al., 2023), Claude (Anthropic, 2024), and GPT-4V (OpenAI, 2023)), together with aforementioned models capable of generating images, to generate textual descriptions for expected images. This assesses if they learn the mapping from low-dimensional textual input to high-dimensional visual output based on the demonstrations. An extensive review of these

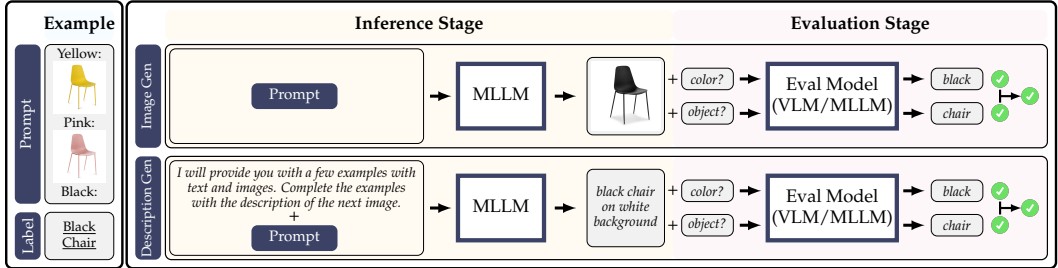

Figure 3: **Benchmarking pipline for MLLMs in T2I-ICL with CoBSAT.** (i) For MLLMs with image generation capabilities, we feed prompts from our dataset into the MLLM under evaluation to prompt image generation. If the MLLM accurately interprets the text-image relationship in the provided demonstrations, it should produce an image of a "black chair." To verify this alignment, we employ one evaluation model, it could be either a Vision-Language Model (VLM, e.g., CLIP) or an MLLM adept at visual question answering (e.g., LLaVA). This allows us to determine whether the generated image accurately corresponds to the target label. (ii) For MLLMs that do not generate images, we modify the process by instructing the MLLMs to describe the image textually, following the same evaluation criteria as in the image generation scenario.

MLLMs, and detailed information about the prompts used for each model, are provided in Sec. A and Sec. D.1, respectively.

In particular, since LLaVA models are primarily designed for visual question answering (Liu et al., 2023a; 2024) and are tailored to work with single-image inputs accompanied by questions, they do not perform well on T2I-ICL tasks as expected. Furthermore, Emu2 requires a significant amount of memory, especially for cases with a large number of demonstrations, which limits our ability to obtain comprehensive results due to resource constraints. Therefore, we defer the results of LLaVA models, as well as the partial results obtained for Emu2 in two-shot and four-shot cases, to Sec. F. In the main body of the paper, we primarily focus on discussing the other seven models.

**Evaluation.** Our evaluation pipeline is depicted in Figure 3, where we leverage both VLM and MLLM to assess whether the generated images or descriptions accurately represent the intended objects (e.g., "car" in the first example in Figure 2) and attributes (e.g., "red" in the same example). Specifically, we employ CLIP for its proficiency in vision-and-language tasks (Hessel et al., 2021; Ruiz et al., 2023), and MLLMs including LLaVA, Qwen-VL, and Gemini to determine the accuracy of the generated content. For CLIP's evaluation, we identify the main object and attribute in the generated content by calculating the similarity between the embeddings of the generated content and the embeddings of all entries within our object and attribute lists. The items with the highest similarity are deemed the predicted labels. In the case of MLLMs, the generated content is embedded into the input, prompting MLLMs to identify the main object and attribute in the generated content, which are then assigned as the predicted labels. We then measure the accuracy of these predictions against the true labels to determine the correctness of the generated content.

In Sec. E, we compare these evaluation models in terms of alignment with human evaluation, and find Gemini > LLaVA-1.5 > CLIP > Qwen-VL in terms of alignment. Since Gemini is not open-sourced and there is a high correlation between the accuracies of LLaVA-1.5 and Gemini, we use free and open-sourced LLaVA-1.5 for all accuracy evaluations in our paper, unless otherwise stated. Additionally, we find that LLaVA-1.5 accurately identifies the correct object and attribute for all images in our dataset, ensuring the reliability of our evaluations. We provide more details such as prompts utilized for evaluation in Sec. D.2.

## 5   Benchmarking MLLMs in T2I-ICL

We visualize the T2I-ICL performance of the considered MLLMs in Figure 4.

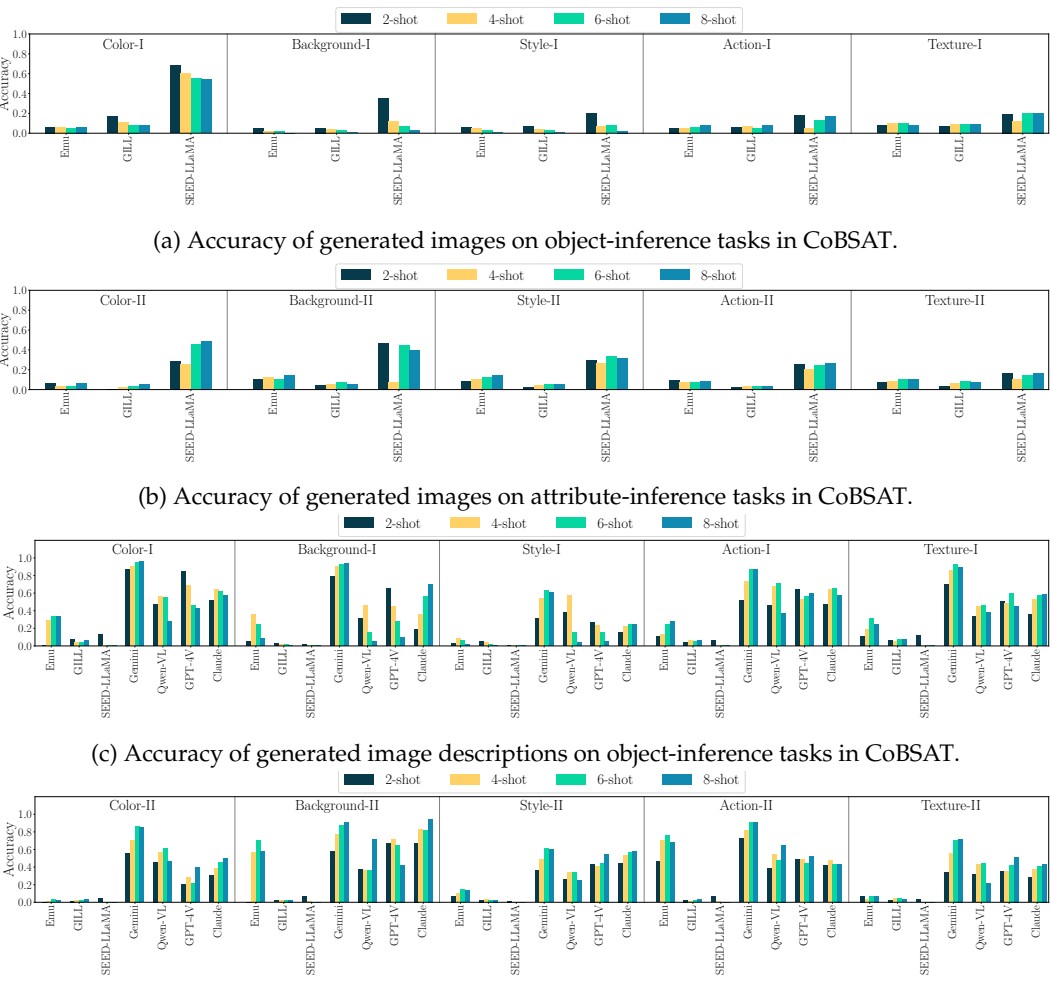

(a) Accuracy of generated images on object-inference tasks in CoBSAT.

(b) Accuracy of generated images on attribute-inference tasks in CoBSAT.

(c) Accuracy of generated image descriptions on object-inference tasks in CoBSAT.

(d) Accuracy of generated image descriptions on attribute-inference tasks in CoBSAT.

Figure 4: T2I-ICL performance of MLLMs on CoBSAT with 2,4,6,8 demonstrations.

**Assessing Generated Images.**    In terms of image generation, we focus on the three MLLMs that have this capability: Emu, GILL, and SEED-LLaMA. Among these, SEED-LLaMA significantly outperforms the others, as evidenced by Figure 4(a) and (b), where it attains accuracies exceeding or nearing 20% across various tasks. Notably, on the Color-I task, SEED-LLaMA reaches an impressive 68% accuracy. In contrast, Emu and GILL exhibit low performance, achieving accuracies around or even below 10%.

GILL's limited performance can be attributed to its training paradigm, which is not optimized for tasks requiring a unified understanding and generation of multimodal content (Ge et al., 2023b). Specifically, this limitation stems from its training that omits interleaved image-text data and the absence of an image generation model during its training process (Koh et al., 2023). In contrast, SEED-LLaMA benefits from instruction fine-tuning across a broad range of datasets, including both multimodal and text-to-image generation datasets such as Instructpix2pix (Brooks et al., 2023), MagicBrush (Zhang et al., 2024), JourneyDB (Sun et al., 2024), DiffusionDB (Wang et al., 2023c), LAION-Aesthetics (LAION, 2022), and VIST (Huang et al., 2016). Emu, on the other hand, has been fine-tuned exclusively on the LLaVA dataset (Liu et al., 2023b) in the context of image-text tasks. This expansive and varied instruction fine-tuning likely accounts for SEED-LLaMA's enhanced performance in T2I-ICL tasks when compared to Emu.

| Model | Shot | Method | Object-Inference Task | | | | | Attribute-Inference Task | | | | |
|---|---|---|---|---|---|---|---|---|---|---|---|---|
| | | | Color-I | Background-I | Style-I | Action-I | Texture-I | Color-II | Background-II | Style-II | Action-II | Texture-II |
| Gemini | 2 | T2I-ICL | .865 | .794 | .315 | .517 | .704 | **.555** | **.583** | .360 | **.725** | .340 |
| | | T-ICL | **.979** | **.907** | **.692** | **.895** | **.764** | .150 | .410 | **.645** | .468 | **.361** |
| | 4 | T2I-ICL | .904 | .908 | .540 | .737 | .861 | .709 | .773 | .484 | **.818** | .553 |
| | | T-ICL | **.988** | **.965** | **.888** | **.965** | **.927** | **.777** | **.780** | **.835** | .783 | **.812** |

Table 1: **Comparison of T2I-ICL v.s. T-ICL accuracy** (see Table 7 for the full version). To perform T-ICL on our dataset, we replace all images in the prompts with their corresponding descriptions. Underlined numbers indicate the highest accuracy achieved for each model and task across various shot numbers, while bold numbers indicate the highest accuracy for each specific combination of model, task, and shot count.

**Assessing Generated Image Descriptions.** Figures 4(c) and (d) reveal that Gemini, Qwen-VL, Claude, and GPT-4V stand out by significantly surpassing other MLLMs in most tasks. It is observed that MLLMs with image-generation capabilities often struggle with generating image descriptions. Among these leading models, Claude, Qwen-VL and GPT-4V show comparable results, whereas Gemini outperforms all of them. Given the lack of detailed information on the training datasets and paradigms for Gemini, Claude, and GPT-4V, our analysis can only extend to Qwen-VL. Notably, Qwen-VL benefits from pretraining on a broader dataset than Emu, GILL, and SEED-LLaMA, contributing to its enhanced performance (Bai et al., 2023b).

**Impact of Number of Demonstrations.** An interesting observation from Figure 4 is the lack of a consistent pattern in how performance is influenced by an increase in the number of demonstrations. For example, the accuracy in generating image descriptions for models such as Emu and Qwen-VL first increases and then decreases with an increasing number of demonstrations generally. Conversely, SEED-LLaMA's accuracy first decreases and then increases. This non-monotonic performance trend with a growing number of demonstrations can potentially be attributed to two factors. Firstly, with a higher number of demonstrations, there may be an insufficient number of pertaining samples featuring the corresponding number of image inputs. Secondly, existing evidence indicates that an increase in demonstrations does not necessarily correlate with enhanced performance (Xie et al., 2022; Brown et al., 2020; Lin & Lee, 2024). Brown et al. (2020) demonstrate that for some datasets (e.g., LAMBADA, HellaSwag, PhysicalQA, RACE-m, CoQA/SAT analogies for smaller models), GPT-3's zero-shot performance may surpass one-shot performance. Similarly, Xie et al. (2022) found that zero-shot scenarios can sometimes outperform few-shot ones, although performance tends to recover with the addition of more examples. Lin & Lee (2024) provided a theoretical explanation for this phenomenon by considering in-context learning as a process that involves both task retrieval and task learning.

We offer a more in-depth analysis in Sec. F.1, which delves further into the discussion above, and additionally (i) explores the impact of textual and visual information on predictions, (ii) investigates the performance of MLLMs in accurately generating the objects and attributes, respectively, and (iii) presents results for a more challenging variant of the CoBSAT benchmark.

## 6 Understanding Challenges in T2I-ICL

In Sec. 5, we observe that most MLLMs still face challenges in performing T2I-ICL effectively. Notably, SEED-LLaMA, Gemini, and Qwen-VL are notable free models, each capable of performing T2I-ICL tasks; SEED-LLaMA performs well for image generation scenarios, whereas Gemini and Qwen-VL specialize in image description generation scenarios. Therefore, unless otherwise stated, our subsequent investigations concentrate on these three models, specifically utilizing SEED-LLaMA for image generation scenarios and Gemini and Qwen-VL for image description generation.

In this section, our goal is to understand the main difficulties leading to this suboptimal performance in T2I-ICL. We hypothesize that the primary difficulties lie in (i) the complexity

| Model | Shot | Precise Textual Inputs | Object-Inference Task | | | | | Attribute-Inference Task | | | | |
|---|---|---|---|---|---|---|---|---|---|---|---|---|
| | | | Color-I | Background-I | Style-I | Action-I | Texture-I | Color-II | Background-II | Style-II | Action-II | Texture-II |
| SEED-LLaMA | 0 | ✓ | .730 | .456 | .356 | .264 | .275 | .582 | .314 | .298 | .207 | .286 |
| | 2 | ✗ | .680 | .348 | .203 | .182 | .196 | .287 | .467 | .297 | .261 | .163 |
| | | ✓ | **.801** | .409 | .241 | .192 | **.326** | .385 | **.485** | **.393** | **.317** | .268 |
| | 4 | ✗ | .482 | .211 | .141 | .053 | .122 | .252 | .076 | .268 | .207 | .105 |
| | | ✓ | .669 | .318 | .284 | .161 | .286 | .608 | .441 | .299 | .278 | .248 |

Table 2: **Accuracy comparison: with or without providing precise textual inputs** (see Table 8 for the full version). Bold numbers represent the highest accuracy for each task and shot count, comparing scenarios with and without descriptive textual inputs. Underlined numbers indicate the highest accuracy for each task across various shots.

| Model | Shot | Fine-tuned | Object-Inference Task | | | | | Attribute-Inference Task | | | | |
|---|---|---|---|---|---|---|---|---|---|---|---|---|
| | | | Color-I | Background-I | Style-I | Action-I | Texture-I | Color-II | Background-II | Style-II | Action-II | Texture-II |
| Qwen-VL | 2 | ✗ | .540 | .236 | **.248** | .412 | .372 | .276 | .244 | .112 | .232 | .224 |
| | | ✓ | **.852** | **.744** | .212 | **.856** | .532 | .516 | **.344** | **.148** | **.520** | **.284** |
| | 4 | ✗ | .680 | .492 | **.448** | .228 | .556 | .512 | **.448** | **.240** | .320 | .420 |
| | | ✓ | **.876** | **.604** | .216 | **.812** | **.588** | **.696** | .308 | .088 | **.656** | **.480** |

Table 3: **T2I-ICL accuracy comparison of pretrained-only versus fine-tuned (FT) MLLM** (see Table 9 for the full version). Underlined numbers denote the highest performance achieved across different methods and shots for each task, while bold numbers indicate the top performance for each shot across various methods within their tasks.

inherent to multimodality, and (ii) the intrinsic challenges of the image generation task itself, which might be independent of the T2I-ICL process. We test these hypotheses as below.

**Is Multimodality a Primary Challenge in T2I-ICL?** The low performance of MLLMs in T2I-ICL is in contrast to the impressive results their underlying LLM demonstrated in T-ICL (Touvron et al., 2023; Bai et al., 2023a). To study whether multimodality is one primary challenge for T2I-ICL, we consider a textual version of our tasks by replacing every image in the prompts with corresponding detailed descriptions, which are initially created by LLaVA and ChatGPT and reviewed and updated by humans. Results in Table 1 show that T-ICL significantly improves the accuracy, especially in the 4-shot scenario. This improvement is also observed in the performance of Qwen-VL and SEED-LLaMA. For an in-depth exploration of the performance of Qwen-VL and SEED-LLaMA, detailed experimental settings, and comprehensive discussion, refer to Sec. F.2.1. These findings validate our hypothesis that multimodality is a principal challenge in T2I-ICL.

**Is the Image Generation a Primary Challenge in T2I-ICL?** We conduct an experiment with 0, 2, and 4-shot image generation tasks, with textual inputs updated as precise labels. For example, in the initial scenario from Figure 2, the terms "White," "Blue," and "Red" are updated to "White car," "Blue car," and "Red car," respectively. The results, as shown in Table 2, reveal that even when precise textual inputs are provided, the accuracies of SEED-LLaMA remain below 50% in most scenarios, maintaining a similar relative performance across different tasks to scenarios without these inputs. This indicates that the task of image generation itself poses a significant challenge for current MLLMs, contributing to their underperformance on the CoBSAT dataset. Similar investigations with Emu and GILL yield consistent conclusions (see Sec. F.2.2).

# 7 Enhancing MLLMs' T2I-ICL Capabilities

In the previous sections, we observed the suboptimal performance of MLLMs in executing T2I-ICL and investigated the primary challenges involved. This section delves into exploring techniques that could potentially enhance the performance of MLLMs in T2I-ICL. Additional details on our experiments, including choices of hyperparameters, prompt templates, results of other MLLMs, and other interesting technique explorations, are provided in Sec. F.3.

| Model | Shot | CoT | Object-Inference Task | | | | | Attribute-Inference Task | | | | |
|---|---|---|---|---|---|---|---|---|---|---|---|---|
| | | | Color-I | Background-I | Style-I | Action-I | Texture-I | Color-II | Background-II | Style-II | Action-II | Texture-II |
| SEED-LLaMA | 2 | ✗ | .680 | **.348** | .203 | **.182** | .196 | **.287** | .467 | **.297** | .261 | **.163** |
| | | ✓ | **.781** | .179 | **.206** | .167 | **.222** | .179 | .389 | .195 | **.300** | .154 |
| | 4 | ✗ | .482 | .211 | .141 | .053 | .122 | .252 | .076 | .268 | .207 | .105 |
| | | ✓ | **.650** | **.353** | **.244** | **.242** | **.208** | **.303** | **.370** | **.335** | **.241** | **.171** |

Table 4: **Accuracy comparison between T2I-ICL with v.s. without CoT** (see Table 10 for the full version). Numbers in bold highlight the highest accuracy achieved for each model, number of shots, and task, and underlined numbers indicate the highest accuracy achieved for each model and task across different numbers of shots.

**Fine-tuning MLLMs on CoBSAT.** Building on the work of Min et al. (2022a), which demonstrates that tuning models on a collection of ICL tasks enables them to learn new tasks in context at test time, we fine-tune two instances of Qwen-VL, one on a 2-shot dataset and the other on a 4-shot dataset, and then compare their performances with their non-fine-tuned counterparts on the T2I-ICL test set. Note that all objects and attributes in the test set are not present in the training set. The results are summarized in Table 3. The results indicate a significant improvement in Qwen-VL's T2I-ICL performance post fine-tuning. A similar trend is observed with SEED-LLaMA, as discussed in Sec. F.3.1. This suggests that fine-tuning MLLMs on a T2I-ICL dataset enhances T2I-ICL capability of MLLMs. Furthermore, a more challenging training-test dataset split is considered in Sec. F.3.1 to study the generalizability of the fine-tuned models in terms of T2I-ICL.

**Intergrating Chain-of-Thought with T2I-ICL.** Another widely utilized method in prompt engineering is Chain-of-Thought (CoT) (Wei et al., 2022). This approach involves incorporating a simple instruction, such as "let's think step by step," prompting the model to sequentially generate concise sentences that outline the reasoning process, commonly referred to as reasoning chains or rationales. The chains are subsequently embedded into the subsequent prompt to obtain the final answer. In this experiment, we investigate the impact of integrating CoT on the T2I-ICL performance of MLLMs. The results are reported in Table 4. With the integration of CoT, SEED-LLaMA shows significant improvement in T2I-ICL performance across all ten tasks in the 4-shot scenario. Similar improvement is observed for Gemini, see Sec. F.3.2.

## 8 Conclusion and Future Works

In this work, we identify an important yet underexplored problem — T2I-ICL, and explore the capability of MLLMs to solve it. To facilitate this investigation, we introduce CoBSAT, a comprehensive benchmark dataset. Our experimental evaluation of MLLMs on this dataset reveals that many MLLMs have difficulty in effectively performing T2I-ICL. We identify two key challenges in T2I-ICL: (i) the integration and understanding of multimodal information; and (ii, particularly for image generation models) the actual process of image creation. To improve MLLMs' performance in T2I-ICL, we carry out additional experimental studies, which suggest that fine-tuning and CoT can substantially enhance T2I-ICL capabilities.

As we identify T2I-ICL as an important problem for the first time, many interesting questions remain open. First, the impact of demonstration selection on T2I-ICL performance is yet to be fully understood. Furthermore, the application of other prevalent prompt engineering techniques to T2I-ICL remains open. While our dataset only covers basic themes, we identify expanding the themes of our dataset and extending it for image editing tasks as two interesting future directions. For a more in-depth discussion, please refer to Sec. G.

## Acknowledgement

The work of Kangwook Lee is supported in part by NSF CAREER Award CCF-2339978, Amazon Research Award, and a grant from FuriosaAI. We would like to express our

appreciation to Prof. Dimitris Papailiopoulos, Hanrong Ye, Changho Shin, Mu Cai, and anonymous reviewers for their insightful comments.

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

# Appendix

# A    In-Depth Overview of MLLMs

In this section, we provide a detailed overview of the MLLMs used in our experiments, including (i) four MLLMs with image generation capabilities: Emu (Sun et al., 2023c), Emu2 (Sun et al., 2023a), SEED-LLaMA (Ge et al., 2023a;b), and GILL (Koh et al., 2023), and (ii) five state-of-the-art MLLMs that can only generate text: Qwen-VL (Bai et al., 2023b), LLaVA models (LLaVA-1.5 (Liu et al., 2023a) and LLaVA-NeXT (Liu et al., 2024)), Gemini (Gemini Team Google: Anil et al., 2023), and GPT-4V (OpenAI, 2023).

**Emu (Sun et al., 2023c).**    Emu integrates EVA-CLIP (Fang et al., 2023) as the Visual Encoder, the Causal Transformer, LLaMA-13B (Touvron et al., 2023), and Stable Diffusion v1.5 as the Visual Decoder. Given any sequence including images and texts, the images are encoded into dense visual features via EVA-CLIP (Fang et al., 2023). These features are then transformed into visual causal embeddings via a Causal Transformer, which converts 2D spatial visual signals into 1D causal sequences. Two special image tokens, `[IMG]` and `[/IMG]`, are prepended and appended to the visual causal embeddings of each image. The visual causal embeddings are then combined with the text tokens and fed into the LLaMA. In the output generated by LLaMA, the visual embeddings in-between image tokens `[IMG]` and `[/IMG]` are decoded using the fine-tuned Stable Diffusion 1.5. All components of Emu are further trained from their initial state using image-text pairs from LAION-2B (Schuhmann et al., 2022a) and LAION-COCO (Schuhmann et al., 2022b), video-text pairs from WebVid-10M (Bain et al., 2021), interleaved image and text from MMC4 (Zhu et al., 2023b), an expanded version of the text-only C4 (Raffel et al., 2020), and interleaved video and text from YT-Storyboard-1B (Zellers et al., 2022; Sun et al., 2023c). Furthermore, Emu can also process videos by treating various frames as a sequence interspersed with text and images.

**Emu2 (Sun et al., 2023a).**    Emu2 represents a upscaled version of its predecessor, Emu, featuring significant upgrades in its component architecture. Unlike Emu, which utilized EVA-CLIP, LLaMA-13B, and Stable Diffusion v1.5 for its Visual Encoder, Multimodal Modeling, and Visual Decoder, respectively, Emu2 employs larger versions: EVA-02-CLIP-E-plus (Sun et al., 2023b) for the Visual Encoder, LLaMA-33B for Multimodal Modeling, and SDXL (Podell et al., 2023) as the Visual Decoder. Moreover, Emu2 replaced Emu's C-Former with mean pooling followed by a linear projection for connecting Visual Encoder and Multimodal modeling. Its pretraining regime also differs, utilizing datasets that includes image-text pairs from LAION-2B (Schuhmann et al., 2022a) and CapsFusion-120M (Yu et al., 2023b), video-text pairs from WebVid-10M (Bain et al., 2021), interleaved image-text data from MMC4 (Zhu et al., 2023b), interleaved video-text data from YT-Storyboard-1B (Zellers et al., 2022; Sun et al., 2023c), grounded image-text pairs from GRIT-20M (Peng et al., 2023b) and CapsFusion-grounded-100M (Yu et al., 2023b), and language-focused data from Pile (Gao et al., 2020).

**SEED-LLaMA (Ge et al., 2023b).**    SEED-LLaMA introduces a tokenizer named SEED, which consists of a ViT encoder (Dosovitskiy et al., 2021) derived from the pretrained BLIP-2 (Li et al., 2023), a Causal Q-Former, a VQ Codebook (van den Oord et al., 2017), a multi-layer perception, and a UNet decoder (Ronneberger et al., 2015) derived from the Stable Diffusion model. When given an input that includes both text and images, the images are first transformed into 2D raster-ordered features by the ViT encoder. These features are then converted into a sequence of causal semantic embeddings via the Causal Q-Former, discretized by the VQ Codebook, and projected by a multi-layer perceptron. The resulting embeddings are integrated with the text embeddings and fed into the LLaMA. The generated image embeddings are subsequently inputted into the Stable Diffusion model to generate realistic images. All components, except for the embedding layer, have been further trained on datasets including COCO Caption (Chen et al., 2015), CC3M (Sharma et al., 2018b), Unsplash (Unsplash Team, 2023), LAION-COCO (Schuhmann et al., 2022b), MMC4 (Zhu et al., 2023b), OBELISC (Laurençon et al., 2023), and WebVid (Bain et al., 2021). Additionally, 26 datasets are employed for supervised instruction tuning of SEED-LLaMA to align it with human instructions.

**GILL (Koh et al., 2023).** GILL employs a pretrained visual backbone and linear projection mapping to process image input, while a tokenizer is used for text input. These inputs are concatenated and fed into OPT-6.7B (Zhang et al., 2022a). The output image embeddings are then processed by a decision model to determine whether to retrieve real images or generate realistic fake ones. For generating realistic images, GILL proposes a GILLMapper, which encompasses a Transformer Encoder that receives image embeddings, and a Transformer Decoder that processes the Encoder's outputs along with certain learned queries. The sequences produced by the Decoder are transformed through a linear layer to generate the predicted embeddings, which are then provided to the Stable Diffusion v1.5 model to create realistic images. For image retrieval, GILL projects the image embeddings via a linear layer and then measures the similarity between these embeddings and those of potential image candidates obtained through the CLIP ViT-L model (Radford et al., 2021). The image exhibiting the highest similarity score is then selected for output. GILL is pretrained on the CC3M dataset (Sharma et al., 2018b).

The three models previously mentioned are MLLMs capable of generating images. Next, we will describe MLLMs that can only generate text.

**Qwen-VL (Bai et al., 2023b).** Qwen-VL is an extension of the Qwen-7B language model (Bai et al., 2023a), equipped with visual capabilities. To achieve this, Qwen-VL incorporates a Vision Transformer (ViT) (Dosovitskiy et al., 2021) with weights initialized from OpenCLIP's ViT-bigG (Ilharco et al., 2021), and a single-layer cross-attention module to convert images into a feature sequence that can be directly fed into Qwen-7B. Qwen-VL is pre-trained using (i) a variety of web-crawled image-text datasets, including LAION-5B, LAION-COCO (Schuhmann et al., 2022a), DataComp (Gadre et al., 2023), Coyo (Byeon et al., 2022), CC12M (Changpinyo et al., 2021), CC3M (Sharma et al., 2018a), SBU (Ordonez et al., 2011), COCO Caption (Chen et al., 2015), and in-house data (Bai et al., 2023b); and (ii) other visual question-answering datasets and visual reasoning datasets, including GQA (Hudson & Manning, 2019), VGQA (Krishna et al., 2017), VQAv2 (Goyal et al., 2019), DVQA (Kafle et al., 2018), OCR-VQA (Mishra et al., 2019), DocVQA (Mathew et al., 2021), GRIT (Peng et al., 2023a), Visual Genome (Krishna et al., 2017), RefCOCO (Kazemzadeh et al., 2014), RefCOCO+, and RefCOCOg (Mao et al., 2016).

**LLaVA (Liu et al., 2023a).** LLaVA is built upon the Vicuna-v1.5-13B LLM (Zheng et al., 2023b). To enable the visual perceiving capability, it incorporates a vision encoder, specifically the CLIP-ViT-L-336px (Radford et al., 2021), along with an MLP projection to encode visual features into image embeddings. These image embeddings, along with text embeddings encoded by tokenization, are then concatenated and fed into the LLM to generate the textual output. Its training follows a two-stage protocol. First, during the vision-language alignment pretraining stage, the model leverages the image-text pairs dataset CC3M (Sharma et al., 2018a) to align the visual features with the language model's word embedding space. Second, the visual instruction tuning stage involves tuning the model on visual instructions to enable it to follow users' diverse requests involving visual content. For this stage, LLaVA utilizes GPT-4V (OpenAI, 2023) to expand the existing COCO (Chen et al., 2015) bounding box and caption dataset into a multimodal instruction-following dataset, which includes three types of instruction-following data: conversational-style QA, detailed description, and complex reasoning. LLaVA-NeXT (Liu et al., 2024) is an improved version of LLaVA, particularly in reasoning, OCR, and world knowledge. It achieves this by increasing the input image resolution to capture more visual details and utilizing Mistral-7B and Nous-Hermes-2-Yi-34B as the additional backbones. Moreover, LLaVA-NeXT utilizes a better mixture of visual instruction tuning data, comprising high-quality user instructions and multimodal document/chart data.

**Claude (Anthropic, 2024).** Claude series is one of the leading LLMs developed by Anthropic. Anthropic recently introduced Claude 3, a family of MLLMs: Claude 3 Opus, Claude 3 Sonnet, and Claude 3 Haiku. Claude 3 can understand multimodal inputs such as photos, tables, and graphs. Besides multimodality, Claude 3 shows better fluency, especially for non-English languages. We chose Claude 3 Haiku for our experiment due to its speed and cost-effectiveness.

**Gemini (Gemini Team Google: Anil et al., 2023).** Gemini, a family of MLLMs developed by Google, is built on Transformer decoders and trained on extensive images, audio, video, and text datasets (including natural images, charts, screenshots, and PDFs). With a 32k context length support, it provides three variants: Ultra, Pro, and Nano, with Ultra offering the highest capabilities and Nano excelling in efficiency. We employ Gemini-pro in our paper.

**GPT-4V (OpenAI, 2023).** GPT-4V has emerged as one of the most proficient MLLMs, demonstrating exceptional performance and achieving human-level results on a majority of professional and academic examinations. Despite being a closed-source MLLM, with undisclosed details about its architecture and dataset construction, GPT-4V is included in our evaluation due to its superior performance compared to other MLLMs (Bai et al., 2023b).

## B Extended Related Works

This section provides detailed related works.

**Textual ICL.** Ever since Brown et al. (2020) demonstrated that language models are in-context learners (see Figure 1(a)), there has been substantial interest in comprehending this capability, both empirically (Liu et al., 2022; Min et al., 2022b; Chen et al., 2022; Mishra et al., 2022; Lampinen et al., 2022; Garg et al., 2022; Hendel et al., 2023) and theoretically (Xie et al., 2022; Wies et al., 2023; Akyürek et al., 2023; Von Oswald et al., 2023; Bai et al., 2023c; Ahn et al., 2023; Zhang et al., 2023b). Textual ICL (T-ICL) enables the adaptation of LLMs to downstream tasks simply by providing a few illustrative examples, bypassing any need for updating model parameters. The existing works indicate that LLMs possess the capability to comprehend context and perform reasoning through T-ICL (Brown et al., 2020).

**Visual ICL.** The concept of V-ICL is then employed in computer vision, starting with the introduction of visual prompts (see Figure 1(b)). The pioneering works by Bar et al. (2022); Wang et al. (2023a) propose to automatically generate output images that are contextually aligned with provided examples. Specifically, Bar et al. (2022) developed a method that combines three images - an example input, its corresponding output, and a query - into a single composite image. In this layout, the example input is placed in the upper left, the example output in the upper right, the query image in the bottom left, and the bottom right patch is left blank for output construction via an image inpainting model. Bar et al. (2022) demonstrated the effectiveness of V-ICL in tasks like edge detection, colorization, inpainting, segmentation, and style transfer. Wang et al. (2023a) introduced a similar approach and trained a generalist model named "Painter," which exclusively uses visual prompts without any textual data for V-ICL. Experiments on standard computer vision benchmarks revealed competitive performance against task-specific models. Nguyen et al. (2023) further applied visual prompts to image editing by inverting visual prompts into text-based editing directions, leveraging the pre-trained capabilities of diffusion models.

A subsequent empirical study by Zhang et al. (2023e) highlighted that the success of V-ICL significantly depends on the choice of in-context demonstrations. The aspect of demonstration selection was further explored by Sun et al. (2023d), who also examined the impact of prompt fusion on performance. Their findings indicate a high sensitivity of performance to the arrangement of sub-images in in-context learning. Moreover, innovative approaches to structuring V-ICL, such as the concept of "visual sentences," have been introduced in recent studies, notably by Bai et al. (2023d). Unlike V-ICL which only handles images, M-ICL integrates demonstrations encompassing both text and images.

**MLLMs.** In light of the significant success of LLMs, there has been an increase in the release of MLLMs. These models are designed to address more challenging multimodal tasks, thereby enabling the perception of images (Li et al., 2022; Alayrac et al., 2022; Hao et al., 2022; Laurençon et al., 2023; Huang et al., 2023b; Peng et al., 2023b; Li et al., 2023; Ge et al., 2023b; Koh et al., 2023; Zhu et al., 2023a; Sun et al., 2023c; Zheng et al., 2023a; OpenAI, 2023; Liu et al., 2023b;a; Bai et al., 2023b; Sun et al., 2023a; Gemini Team Google: Anil et al.,

2023; Driess et al., 2023; Anthropic, 2024), videos (Li et al., 2022; Alayrac et al., 2022; Li et al., 2023; Sun et al., 2023c; Gemini Team Google: Anil et al., 2023), and audio (Hao et al., 2022; Borsos et al., 2023; Huang et al., 2023a; Chen et al., 2023a; Zhang et al., 2023a; Gemini Team Google: Anil et al., 2023). Existing models capable of handling images can be categorized as follows: (i) those that use language as a general interface and directly employ LLMs without altering the model architectures (Dinh et al., 2022; Cai et al., 2023; Aghajanyan et al., 2022; Yu et al., 2023a; Huang et al., 2023b; Mirchandani et al., 2023; Cai et al., 2023); (ii) those that add one or more modules before feeding the input sequence into the LLM to perceive multimodal inputs (Tsimpoukelli et al., 2021; Alayrac et al., 2022; Awadalla et al., 2023; Laurençon et al., 2023; Li et al., 2023; Hao et al., 2022; Liu et al., 2023b;a; Zhu et al., 2023a; Gemini Team Google: Anil et al., 2023; Liu et al., 2024); (iii) those that add one or more modules after the LLM processing for generating multimodal outputs (Pan et al., 2023); (iv) those that add modules to both inputs and outputs of the LLMs to process the multimodal input and generate multimodal outputs (Dong et al., 2023; Sun et al., 2023c; Koh et al., 2023; Ge et al., 2023b; Zheng et al., 2023a; Sun et al., 2023a).

In this paper, our main focus is T2I-ICL. We aim to investigate whether MLLMs can learn to transform low-dimensional textual input into high-dimensional visual output based on demonstrations, and to accurately generate images from new textual queries. Consequently, we focus on models capable of processing both text and multiple images. We consider two types of MLLMs: (i) proficient in generating both text and images, including Emu (Sun et al., 2023c), Emu2 (Sun et al., 2023a), GILL (Koh et al., 2023), and SEED-LLaMA (Ge et al., 2023b), and (ii) those limited to text generation, including GPT-4V (OpenAI, 2023), Gemini (Gemini Team Google: Anil et al., 2023), Claude (Anthropic, 2024), LLaVA-1.5 (Liu et al., 2023a), LLaVA-NeXT (Liu et al., 2024), and Qwen-VL (Bai et al., 2023b). For text-only MLLMs, we evaluate their capacity to infer visual outputs by prompting them to describe the anticipated image. Conversely, for MLLMs capable of image generation, we not only elicit image outputs but also ask for descriptive text, ensuring an apple-to-apple comparison with text-only models.

**Image-to-Text ICL in MLLMs.**    Most existing work on M-ICL focuses on the image-to-text generation, i.e., I2T-ICL, which involves mapping from high-dimensional input (i.e., images) to low-dimensional output (i.e., text). In particular, Tsimpoukelli et al. (2021) were the first to extend ICL from the text domain to the multimodal domain, focusing on image-to-text generation such as visual question-answering (see Figure 1(c)). Alayrac et al. (2022) introduced Flamingo, an MLLM that achieves state-of-the-art performance in a variety of image and video understanding tasks using I2T-ICL with 32 demonstrations, implying the efficacy of I2T-ICL in performance enhancement in their model. In contrast, Monajatipoor et al. (2023) explores whether the in-context capabilities of LLMs can be seamlessly extended to I2T-ICL by incorporating a visual encoder. Chen et al. (2023b) conducted a systematic study on the importance of visual and textual information in I2T-ICL. Concurrently, efforts have been made to develop datasets specifically designed for evaluating I2T-ICL capability of MLLMs (Zhao et al., 2023). In contrast, there have been only a few attempts (Sun et al., 2023a) to evaluate the T2I-ICL capability of MLLMs, a domain that remains relatively unexplored compared to its image-to-text counterpart.

**Zero-Shot Image Generation in MLLMs.**    A relatively small number of MLLMs are capable of image generation (Yu et al., 2023a; Dong et al., 2023; Zheng et al., 2023a; Sun et al., 2023c; Ge et al., 2023b; Koh et al., 2023; Pan et al., 2023; Sun et al., 2023a). Zero-shot text-to-image generation typically generates images directly from textual descriptions without relying on any examples. This does not require the model to integrate a combination of textual and visual inputs. Another common task for MLLMs in image generation is context modifications. In this more complex scenario, the model receives visual inputs (e.g., an image of a dog) along with associated textual instructions (e.g., "swimming underwater"). This task requires a nuanced understanding and manipulation of the image, guided by the textual instructions, thereby blending image comprehension with contextual transformation based on text. Unlike zero-shot image generation, our focus is on studying whether MLLMs can learn the implicit relationship between the input and output from multiple in-context demonstrations.

**Text-to-Image ICL in MLLMs.** There are limited attempts to evaluate MLLMs based on their T2I-ICL capabilities. A notable exception is concurrent research by Sun et al. (2023a). They evaluated the performance of their model on T2I-ICL with DreamBooth dataset (Ruiz et al., 2023). However, it is important to note that the DreamBooth dataset, primarily developed for fine-tuning models to modify image contexts, was not specifically designed for T2I-ICL applications. This leads to certain constraints, such as its concentrated emphasis on altering backgrounds only and a level of complexity that may not align well with T2I-ICL. In contrast, our dataset spans five themes and provides well-designed prompts to assess whether models can understand both visual and textual information, learn mappings from demonstrations, and make inferences.

**Image Evaluation Metrics.** A variety of metrics exist for assessing the quality of generated images. Classical ones like Peak Signal-to-Noise Ratio (PSNR) (Wang et al., 2004) evaluate the quality of reconstructed images or videos by measuring pixel-level errors compared to the target images. Fréchet Inception Distance (FID) (Parmar et al., 2022) gauges the quality of images produced by generative models, such as Generative Adversarial Networks, by calculating the similarity between the distributions of generated and real images. However, these metrics are not entirely suitable for our purpose, where no single definitive ground-truth target image exists but rather a textual label (e.g., "red car" in the first example of Figure 2).

In the realm of text-to-image generation, the CLIP similarity (Radford et al., 2021) metric has gained popularity (Ruiz et al., 2023). It measures the cosine similarity between the CLIP embeddings of the textual ground truth and the visual output. Meanwhile, there is a growing trend of utilizing MLLMs for evaluation (Zhang et al., 2023c; Hu et al., 2023), showing promising results in text-to-image tasks. Our study both approaches, utilizing CLIP (Radford et al., 2021) and MLLMs including LLaVA-1.5 (Liu et al., 2023a), Gemini (Gemini Team Google: Anil et al., 2023), and Qwen-VL (Bai et al., 2023b) to assess the accuracy of generated images. To be more specific, we utilize CLIP and MLLMs to identify the object (e.g., "car") and attribute (e.g., "red") in the image generated by MLLMs and then compare these identifications with the actual label (e.g., "red car" for the first example in Figure 2). The details are provided in Sec. 4. Unless specified otherwise, the accuracy reported in our studies is primarily estimated using LLaVA-1.5, whose effectiveness has been validated by by its ability to accurately recognize objects and attributes, achieving a 100% accuracy rate within our dataset, and closely aligning with human evaluation, as detailed in our analysis in Sec. E.

## C   More Details of CoBSAT Dataset

**Detailed Structure.** The detailed structure of all tasks in our dataset is provided in Table 5.

**Copyright Considerations.** It is important to note that the images generated using DALL-E 3 for our dataset are not subject to copyright restrictions. As per the content policy and terms of the DALL-E 3 service, users retain ownership rights over the images they create, including the rights to reprint, sell, and merchandise, irrespective of whether the images were generated using free or paid credits (OpenAI, 2023).

## D   Detailed Experiment Setup

In this section, we provide the details of our experiment setup, including prompt template design for model inference (Sec. D.1) and prompt design for model evaluation (Sec. D.2).

### D.1   Prompt Templates for Model Inference

For generating images based on in-context input-output pairs, we employ the prompt template depicted in Figure 3 for SEED-LLaMA and Emu. This template simply includes

| Category | Task | Text Input $x \in \mathcal{X}$ | Latent Variable $\theta \in \Theta$ | Image Output $y \sim f_\theta(x)$ |
|---|---|---|---|---|
| Object-Inference | Color-I | [Text: **color** $\in$ {yellow, white, red, purple, pink, orange, green, brown, blue, black}] | object $\in$ {leaf, hat, cup, chair, car, box, book, ball, bag, apple} | [Image: **object $\theta$ of color $x$**] |
| | Background-I | [Text: **background** $\in$ {beach, desert, glacier, volcano, park, gym, waterfall, space, cave, seafloor}] | animal $\in$ {zebra, tiger, sheep, pig, monkey, lion, dog, cow, cat, bird} | [Image: **animal $\theta$ in background $x$**] |
| | Style-I | [Text: **style** $\in$ {watercolor, sketch, pixel, origami, lego, icon, graffiti, futuristic, wireframe, old}] | object $\in$ {leaf, hat, cup, chair, car, box, book, ball, bag, apple} | [Image: **object $\theta$ in style $x$**] |
| | Action-I | [Text: **action** $\in$ {swim, sleep, sing, run, read, fly, eat, drink, cry, angry}] | animal $\in$ {zebra, tiger, sheep, pig, monkey, lion, dog, cow, cat, bird} | [Image: **animal $\theta$ doing $x$**] |
| | Texture-I | [Text: **texture** $\in$ {wood, wicker, sequined, plastic, paper, metal, leather, lace, denim, ceramic}] | object $\in$ {leaf, hat, cup, chair, car, box, book, ball, bag, apple} | [Image: **object $\theta$ in texture $x$**] |
| Attribute-Inference | Color-II | [Text: **object** $\in$ {leaf, hat, cup, chair, car, box, book, ball, bag, apple}] | color $\in$ {yellow, white, red, purple, pink, orange, green, brown, blue, black} | [Image: **object $x$ of color $\theta$**] |
| | Background-II | [Text: **animal** $\in$ {zebra, tiger, sheep, pig, monkey, lion, dog, cow, cat, bird}] | background $\in$ {beach, desert, glacier, volcano, park, gym, waterfall, space, cave, seafloor} | [Image: **animal $x$ in background $\theta$**] |
| | Style-II | [Text: **object** $\in$ {leaf, hat, cup, chair, car, box, book, ball, bag, apple}] | style $\in$ {watercolor, sketch, pixel, origami, lego, icon, graffiti, futuristic, wireframe, old} | [Image: **object $x$ in style $\theta$**] |
| | Action-II | [Text: **animal** $\in$ {zebra, tiger, sheep, pig, monkey, lion, dog, cow, cat, bird}] | action $\in$ {swim, sleep, sing, run, read, fly, eat, drink, cry, angry} | [Image: **animal $x$ doing $\theta$**] |
| | Texture-II | [Text: **object** $\in$ {leaf, hat, cup, chair, car, box, book, ball, bag, apple}] | texture $\in$ {wood, wicker, sequined, plastic, paper, metal, leather, lace, denim, ceramic} | [Image: **object $x$ in texture $\theta$**] |

Table 5: **Task summary of CoBSAT.** We use [Text: **description**] to denote the text providing the corresponding description. For instance, [Text: **color**] could refer to terms such as "red" and "black." Each task is characterized by the input space $\mathcal{X}$, and the latent variable space $\Theta$. For $N$-shot inference, we generate 1,000 prompts. Each prompt is obtained by randomly sampling $\theta \in \Theta$ and $(x_n)_{n=1}^{N+1} \in \mathcal{X}^{N+1}$, followed by collecting the corresponding images $(y_n)_{n=1}^{N}$, where $y_n \sim f_\theta(x_n)$.

the in-context samples and the text query, without any additional instructions. For GILL, we add an additional system message: "*You are a professional assistant who can generate a new image based on the sequence.*"

In the subsequent subsections, we present our prompts for instructing MLLMs to generate image descriptions, continuing from the discussion in Sec. 4, and prompts for articulating the text-to-image relationship, continuing from the discussion in Sec. 7.

### D.1.1 Instructing MLLMs for Generating Image Descriptions

In this part, we provide the prompt templates used for instructing all considered models to generate image descriptions:

- **Emu**: We add the instruction as a system message: "*Based on the sequence, describe the next image clearly, including attributes such as the main object, color, texture, background, action, style, if applicable.*"

- **Emu2**: We append "*Based on the sequence, describe the next image clearly, including details such as the main object, color, texture, background, action, style, if applicable.*" to the end of the input.

- **GILL**: We insert "*You are a professional assistant and always answer my question directly and perfectly without any excuses.*" at the beginning of the prompt and append "*Based on the sequence, describe what the next image should be clearly, including attributes such as the main object, color, texture, background, action, style, if applicable. Your response should only contain a description of the image, and any additional information can cause significant loss.*" at the end of the input.

- **SEED-LLaMA**: We insert *"I will provide you a few examples with text and image. Complete the example with the description of next image. Tell me only the text prompt and I'll use your entire answer as a direct input to A Dalle-3. Never say other explanations."* at the beginning of the prompt.

- **LLaVA-1.5 & LLaVA-NeXT**: We add *"Based on the sequence, describe the next image to be generated clearly, including attributes such as the main object, color, texture, background, action, style, if applicable."* at the end of the prompt.

- **Qwen-VL**: We insert *"You are a professional assistant and always answer my question directly and perfectly without any excuses."* to the start of the prompt and append *"Based on the sequence, describe what the next image should be clearly, including attributes such as the main object, color, texture, background, action, style, if applicable. Your response should only contain a description of the image, and all other information can cause huge loss."* to the end of the input.

- **Gemini**: We append *"Based on the sequence, describe the next image clearly, including details such as the main object, color, texture, background, action, style, if applicable."* at the end of the prompt.

- **Claude**: We prepend *"I will provide you a few examples with text and image. Complete the example with the description of next image. Never say other explanations. "* to the beginning of the prompt, and append *"Give me the description of the your predicted next image."* at the end of the prompt.

- **GPT-4V**: We add *"I will provide you with a few examples with text and images. Complete the example with the description of the next image. The description should be clear with main object, and include attributes such as color, texture, background, style, and action, if applicable. Tell me only the text prompt and I'll use your entire answer as a direct input to A Dalle-3. Never say other explanations."* at the start of the input.

### D.1.2 Articulating the Text-to-Image Relationship in Prompts

We now present the instructions for articulating the text-to-image relationship for the experiment presented in Sec. 7.

For image generation, we add the following sentences to the start of the prompts for each task.

- **Color-I**: *"Please identify the common main object in the images, and generate another image of this object of the requested color."*

- **Color-II**: *"Please identify the common color in the images, and generate another image of the requested object in the same color."*

- **Background-I**: *"Please identify the common animal in the images, and generate another image of this animal walking in the requested background."*

- **Background-II**: *"Please identify the common background in the images, and generate another image of the requested animal walking in the same background."*

- **Style-I**: *"Please identify the common object in the images, and generate another image of this object in the requested style."*

- **Style-II**: *"Please identify the common style in the images, and generate another image of the requested object in the same style."*

- **Action-I**: *"Please identify the common animal in the images, and generate another image of this animal doing the requested action."*

- **Action-II**: *"Please identify the common action/mood the animal is doing in the images, and generate another image of the requested animal doing the same action/mood."*

- **Texture-I**: *"Please identify the common main object in the images, and generate another image of this object of the requested texture."*

- **Texture-II**: *"Please identify the common texture of the objects in the images, and generate another image of the requested object in the same texture."*

For image description, we add the following sentences to the start of the prompts for each task.

- **Color-I**: *"Please identify the common main object in the images, and describe the next image to be generated based on the sequence below. Your description of the image should contain the description of the common main object and the requested color."*

- **Color-II**: *"Please identify the common main color in the images, and describe the next image to be generated based on the sequence below. Your description of the image should contain the description of the requested object and the common color."*

- **Background-I**: *"Please identify the common animal in the images, and describe the next image to be generated based on the sequence below. Your description of the image should contain the description of the common animal and the requested background."*

- **Background-II**: *"Please identify the common background in the images, and describe the next image to be generated based on the sequence below. Your description of the image should contain the description of the requested animal and the common background."*

- **Style-I**: *"Please identify the common object in the images, and describe the next image to be generated based on the sequence below. Your description of the image should contain the description of the common object and the requested style."*

- **Style-II**: *"Please identify the common style in the images, and describe the next image to be generated based on the sequence below. Your description of the image should contain the description of the requested object and the common style."*

- **Action-I**: *"Please identify the common animal in the images, and describe the next image to be generated based on the sequence below. Your description of the image should contain the description of the common animal and the requested action."*

- **Action-II**: *"Please identify the common action/mood the animal is doing in the images, and describe the next image to be generated based on the sequence below. Your description of the image should contain the description of the requested animal and the common action/mood."*

- **Texture-I**: *"Please identify the common main object in the images, and describe the next image to be generated based on the sequence below. Your description of the image should contain the description of the common main object and the requested texture."*

- **Texture-II**: *"Please identify the common texture of the objects in the images, and describe the next image to be generated based on the sequence below. Your description of the image should contain the description of the requested object and the common texture."*

### D.2 Prompt Templates for Model Evaluation

In this section, we present our prompt templates for model evaluation. The evaluation encompasses two scenarios: (i) assessing the generated images, and (ii) assessing the generated image descriptions.

**Assessing Generated Images.** Unless otherwise stated, we employ LLaVA-1.5 to evaluate the generated images in terms of whether they generated the right object (e.g., "car" in the first example in Figure 2) and attribute (e.g., "red" in the first example in Figure 2). To facilitate this evaluation, we design specific prompts for LLaVA. Here are the prompts designed for tasks Color-I and II:

- Object Identification: *"[Image: **generated image**] What is the main object in this image? Answer from the following options: (1)leaf (2)hat (3)cup (4)chair (5)car (6)box (7)book (8)ball (9)bag (10)apple. Answer the number only and do not include any other texts (e.g., 1)."*

- Attribute Identification: *"[Image: **generated image**] What is the color (of the main object) in this image? Answer from the following options: (1)yellow (2)white (3)red (4)purple (5)pink (6)orange (7)green (8)brown (9)blue (10)black. Answer the number only and do not include any other texts (e.g., 1)."*

For other tasks involving different themes, the options and the attribute category (e.g., replace "color" in the attribute inference prompt with "style" for tasks Style-I and II) are updated correspondingly.

**Assessing Generated Image Descriptions.**   We also use LLaVA-1.5 to evaluate the generated image descriptions. However, in this case, we modify the prompts used for assessing generated images by replacing "[Image: **generated image**]" with "Image caption: [Text: **generated description**]."

# E   Comparison of T2I-ICL Evaluation Metrics

In our experiments, we leverage LLaVA-1.5 for estimating the accuracy of the output of T2I-ICL. However, there are also many other alternatives such as CLIP, Gemini, and Qwen-VL. In this experiment, we study and compare the effectiveness of different models in terms of evaluating the performance of T2I-ICL.

**Evaluation Metrics.**   This comparison focuses on the accuracy metrics derived from CLIP and MLLMs including Gemini, LLaVA-1.5, and Qwen-VL, with results gathered from SEED-LLaMA's 2-shot T2I-ICL on CoBSAT. *MLLM accuracy* is determined by using MLLM to identify the main object and specific attribute (e.g., color) in the generated images or descriptions leveraging prompts provided in Sec. D.2, which are then matched against the true labels. *CLIP accuracy* is computed based on CLIP similarity. CLIP similarity measures the cosine similarity between the true label's CLIP embedding and that of the generated content. CLIP accuracy involves selecting the most similar object and attribute from the predefined list based on their CLIP embedding's cosine similarity with the generated image or description. These selections are then compared with the true labels to determine accuracy.

**Alignment of T2I-ICL Evaluation Metrics with Human Evaluation.**   We first investigate their alignments with human evaluation. We manually labeled 100 images generated by SEED-LLaMA through T2I-ICL, selecting ten random images from each task to serve as a baseline. It is important to note that some images were of suboptimal quality, presenting ambiguities that could be interpreted both as correct or incorrect. Despite these difficulties, our evaluations using the LLaVA-1.5 show strong alignment with human assessments, achieving a consistency rate of 89% (computed by the ratio of agreement between the two methods). Notably, other MLLMs, especially Gemini, also exhibited commendable performance, as shown in Table 6.

| Model | CLIP | LLaVA-1.5 | Qwen-VL | Gemini |
|---|---|---|---|---|
| Consistency Rate to Human Evaluation | .85 | .89 | .78 | **.92** |

Table 6: Alignment between human evaluations and automatic evaluations performed by CLIP, LLaVA-1.5, Qwen-VL, and Gemini.

**Comparison among Evaluation Metrics.**   We further conducted a scaled statistical study with 20,000 images to compare the performance of these automatic metrics, particularly focusing on how other metrics relate to Gemini's results, given its closest alignment with human evaluations.

Figure 6, 7, and 8 depict the alignment between the accuracy estimates of Gemini and those provided by CLIP, Qwen-VL, and LLaVA-1.5, respectively. The analyses demonstrate a robust correlation between the accuracy estimates of LLaVA-1.5 and Gemini, highlighted by the narrow confidence interval represented by the purple shadow in the figures. This correlation strengthens our confidence in LLaVA-1.5 as a reliable and accessible open-source evaluation alternative to closed-source models in evaluating MLLMs' T2I-ICL performance.

# F   Detailed and Extended Results of Experiments

In this section, we supplement the experimental details, extended experiments, and discussions that could not be addressed in the main body due to space limitations. Specifically,

Sec F.1, F.2, and F.3 provide additional experiment results and discussions for Sec 5, 6, and 7, respectively.

## F.1 Benchmarking MLLMs in T2I-ICL (Detailed Version of Sec. 5)

This is an extended discussion of Section 5.

In this section, we present and analyze our experimental results on the evaluation of all the considered MLLMs' performance on T2I-ICL, including the MLLMs that are not discussed in the main paper, i.e., LLaVA-1.5, LLaVa-NeXT, and Emu2. The full evaluation results are visualized in Figure 9. In addition to supplementing more detailed information on top of the main body, we also present a comparison of textual and visual information in Sec. F.1.4, and a comparison of object and attribute generation in Sec. F.1.5. Furthermore, we explore a more complex variation of our dataset, with detailed descriptions of the experiments and results presented in Section F.1.6.

### F.1.1 Assessing Generated Images

In terms of image generation, we focus on the four MLLMs that have this capability: Emu, Emu2, GILL, and SEED-LLaMA. Among these, SEED-LLaMA significantly outperforms the others, as evidenced by Figure 9(a) and (b), achieving a score of 68% on Color-I tasks. In contrast, Emu, Emu2, and GILL exhibit low performance, achieving accuracies around or even below 10%. For a more tangible understanding, we present specific prompts alongside the images generated using Emu, Emu2, GILL, and SEED-LLaMA in Figure 14, 15, 16, 17, and 18. We observe that while Emu, Emu2, and GILL exhibit low performance, GILL does manage to generate images that either align with the textual query (e.g., "pink" in the fourth example of Figure 14(a)) or adhere to common visual patterns (e.g., "monkey" in the fourth example of Figure 15(a)). Conversely, Emu occasionally generates random images, as seen in the fourth example of Figure 14(a). On the other hand, Emu2's generated images more closely resemble a blend of the input images in the prompt, such as the fifth example of Figure 16(b).

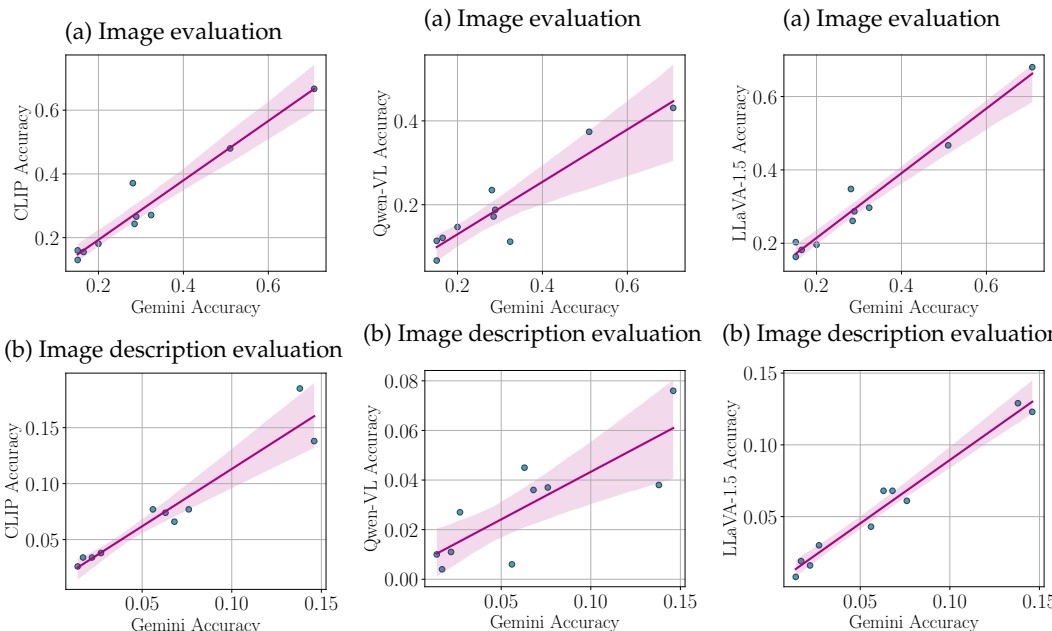

Figure 6: Accuracy estimated by CLIP versus accuracy estimated by Gemini.

Figure 7: Accuracy estimated by Qwen-VL versus accuracy estimated by Gemini.

Figure 8: Accuracy estimated by LLaVA-1.5 versus accuracy estimated by Gemini.

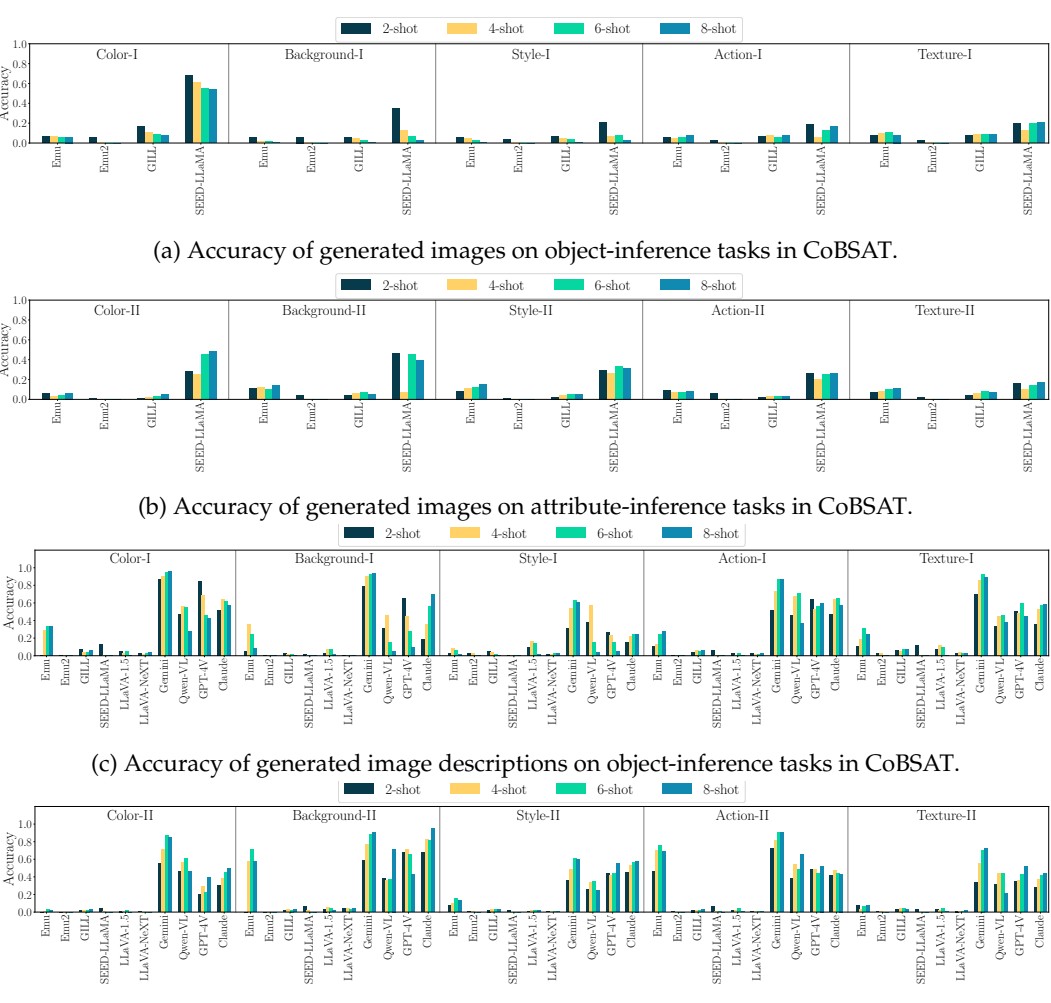

(a) Accuracy of generated images on object-inference tasks in CoBSAT.

(b) Accuracy of generated images on attribute-inference tasks in CoBSAT.

(c) Accuracy of generated image descriptions on object-inference tasks in CoBSAT.

(d) Accuracy of generated image descriptions on attribute-inference tasks in CoBSAT.

Figure 9: T2I-ICL performance of all evaluated MLLMs on the CoBSAT benchmark with 2,4,6,8 in-context demonstrations.[1]

GILL's limited performance can be attributed to its training paradigm, which is not optimized for tasks requiring a unified understanding and generation of multimodal content (Ge et al., 2023b). Specifically, this limitation stems from its training that omits interleaved image-text data and the absence of an image generation model during its training process (Koh et al., 2023). Meanwhile, both Emu and Emu2 update all components in their model, and there is empirical evidence showing that they can better understand multimodal prompts. During instruction fine-tuning, Emu has been fine-tuned exclusively on the LLaVA dataset (Liu et al., 2023b) in the context of image-text tasks, while Emu2 is fine-tuned on more image-text pair data, including LLaVA and LLaVAR (Zhang et al., 2023d). In contrast, SEED-LLaMA benefits from instruction fine-tuning across a broad range of datasets, including both multimodal and text-to-image generation datasets such as Instructpix2pix (Brooks et al., 2023), MagicBrush (Zhang et al., 2024), JourneyDB (Sun et al., 2024), DiffusionDB (Wang et al., 2023c), LAION-Aesthetics (LAION, 2022), and VIST (Huang et al., 2016). This specific text-to-image generation dataset for instruction fine-tuning likely accounts for SEED-LLaMA's enhanced performance in T2I-ICL tasks when compared to Emu and Emu2.

---

[1]Warning: it should be noted that Emu2 has results for 2 and 4-shot scenarios, but results for 6 and 8-shot scenarios are unavailable due to resource constraints, as Emu2 demands excessive memory.

### F.1.2 Assessing Generated Image Descriptions

For image description generation, Figure 9(c) and (d) illustrate the performance of MLLMs in performing T2I-ICL for object-inference and attribute-inference tasks, respectively. We observe that Gemini, Qwen-VL, Claude, and GPT-4V stand out by significantly surpassing other MLLMs in most tasks. Among these leading models, Qwen-VL, Claude, and GPT-4V show comparable results, whereas Gemini outperforms them all.

To further investigate the performance of each model, we offer examples of prompts and their corresponding image descriptions, generated by MLLMs in Figure 19, 20, 21, 22, and 23. We observe that SEED-LLaMA and GILL often struggle to produce relevant textual output. GILL, tends to produce disjointed sentences like "person - bird on the beach - watercolor painting - watercolor," as exemplified in Figure 20(a). SEED-LLaMA, on the other hand, predominantly generates images, defaulting to the text "I have generated an image," regardless of varying instructions. Emu, Emu2, LLaVA-1.5, and LLaVA-NeXT all tend to describe the images contained in the prompt instead of making predictions. This is expected for LLaVA models since they are mostly trained for single image inputs with related questions and answers. Their primary function is to describe and answer questions related to the single image inputs rather than making predictions. In terms of image-text datasets, Emu and Emu2 are also instruction fine-tuned on LLaVA and LLaVAR datasets, thus sharing the same property as LLaVA models. However, they perform slightly better than LLaVA models. For instance, Emu makes the correct prediction in the second example in Figure 19(a). This improvement can be attributed to their pretraining on many other datasets, including interleaved image and text datasets such as Multimodal-C4 (Zhu et al., 2023b). For the leading models, which include Gemini, Claude, Qwen-VL, and GPT-4V, Qwen-VL is the only one that includes detailed information on the training datasets and paradigms. Notably, Qwen-VL benefits from pretraining on a broader dataset than Emu, GILL, SEED-LLaMA, LLaVA-1.5, and LLaVA-NeXT, contributing to its enhanced performance (Bai et al., 2023b).

### F.1.3 Impact of Number of Demonstrations

In this part, we analyze how the number of demonstrations affects the performance of T2I-ICL. An interesting observation from Figure 4 is the lack of a consistent pattern in how performance is influenced by an increase in the number of demonstrations. For example, the accuracy in generating image descriptions for models such as Emu, Qwen-VL, and LLaVA initially increases and then decreases with an increasing number of demonstrations generally. Conversely, SEED-LLaMA's accuracy first decreases and then increases.

This non-monotonic performance trend with a growing number of demonstrations can potentially be attributed to two factors. Firstly, with a higher number of demonstrations, there may be an insufficient number of pertaining samples featuring the corresponding number of image inputs. For example, LLaVA encounters a context length limitation when presented with eight image inputs, resulting in the model generating only empty strings in 8-shot cases. Secondly, existing evidence indicates that an increase in demonstrations does not necessarily correlate with enhanced performance (Xie et al., 2022; Brown et al., 2020). Brown et al. (2020) demonstrate that for some datasets (e.g., LAMBADA, HellaSwag, PhysicalQA, RACE-m, CoQA/SAT analogies for smaller models), GPT-3's zero-shot performance may surpass one-shot performance. Similarly, Xie et al. (2022) found that zero-shot scenarios can sometimes outperform few-shot ones, although performance tends to recover with the addition of more examples. Xie et al. (2022) posit that an initial decrease in accuracy may be due to the distracting structure of prompts in such settings. Theoretical insights from Lin & Lee (2024) shed light on this phenomenon, suggesting that models initially rely on task retrieval and prior knowledge for predictions with a low number of demonstrations, shifting towards task learning as the number of demonstrations increases. The presence of a limited number of initial demonstrations might result in the retrieval of an incorrect task, potentially causing a decline in ICL performance. As more demonstrations are added, performance is anticipated to improve, as the model increasingly depends on task learning, which is improved by a greater number of demonstrations. However, in the MLLM scenario,

due to the scarcity of prompts with multiple images in the pretrained dataset, we do not anticipate observing this phenomenon.

### F.1.4 Textual Information v.s. Visual Information

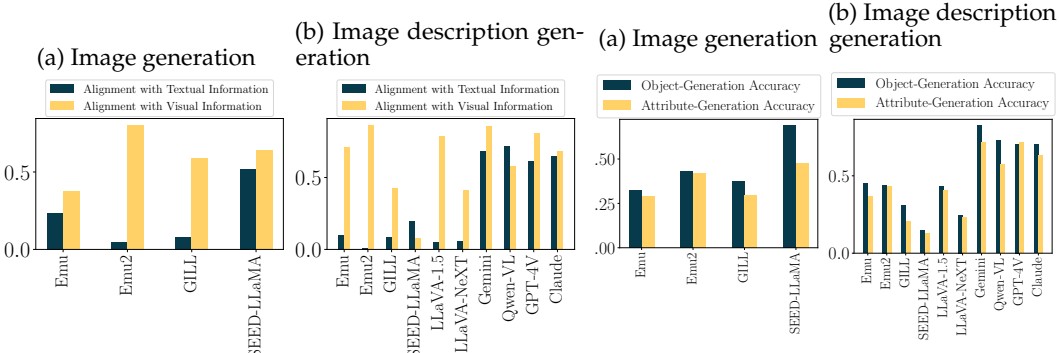

Figure 11: Comparison of alignment with textual information versus visual information when MLLMs perform two-shot T2I-ICL on the CoBSAT tasks.

Figure 12: Comparison of object generation accuracy and attribute generation accuracy when MLLMs perform two-shot T2I-ICL on the CoBSAT tasks.

In this part, we investigate whether textual or visual information contributes more to the prediction of MLLMs. We assess this by evaluating how well the output of MLLMs aligns with both types of information. For textual alignment, we concentrate on how accurately the models generate images or image descriptions that match the given textual query. As an example, consider the scenario in Figure 2, where the text instructs "red." In this context, we consider the output textually aligned if it features a red object. We employ a similar approach to measure the visual alignment of the outputs. Similarly, for visual alignment, we examine whether the generated images or their descriptions accurately incorporate elements from the images presented in the prompts. Taking the same example, an output is visually aligned if it correctly represents aspects like "car," which is the common feature in the demonstration images. Employing these criteria allows us to determine which models are more influenced by textual queries and which lean toward visual cues.

Figure 11 reveals distinct patterns in how MLLMs respond to these inputs. Models such as Emu, Emu2, GILL, LLaVA-1.5, and LLaVA-NeXT demonstrate a marked reliance on visual information in their inputs. This aligns with our findings discussed in Sec. F.1.1 and F.1.2. As we discussed in Sec. F.1.1, GILL's training exclusively on the CC3M dataset (Sharma et al., 2018b), an image-caption corpus, limits its predictive capabilities. For Emu, Emu2, LLaVA-1.5, and LLaVA-Next, they consistently generate descriptions of the images present in the prompt rather than predicting the next image based on the sequence, thus ignoring the textual query in the prompt. In contrast, the models that perform well, such as SEED-LLaMA for image generation and Qwen-VL, GPT-4V, Claude, and Gemini for image description generation, demonstrate a more balanced use of both textual and visual information compared to the other models.

### F.1.5 Object Generation v.s. Attribute Generation

We are also interested in evaluating the proficiency of different MLLMs in inferring objects (e.g., car, chair) and attributes (e.g., color, style). As such, we report the accuracy of MLLMs in generating the correct objects and attributes. These results are depicted in Figure 12. Our observations reveal that all MLLMs perform better in generating correct objects, indicating that the task of generating accurate attributes presents a greater challenge compared to generating correct objects.

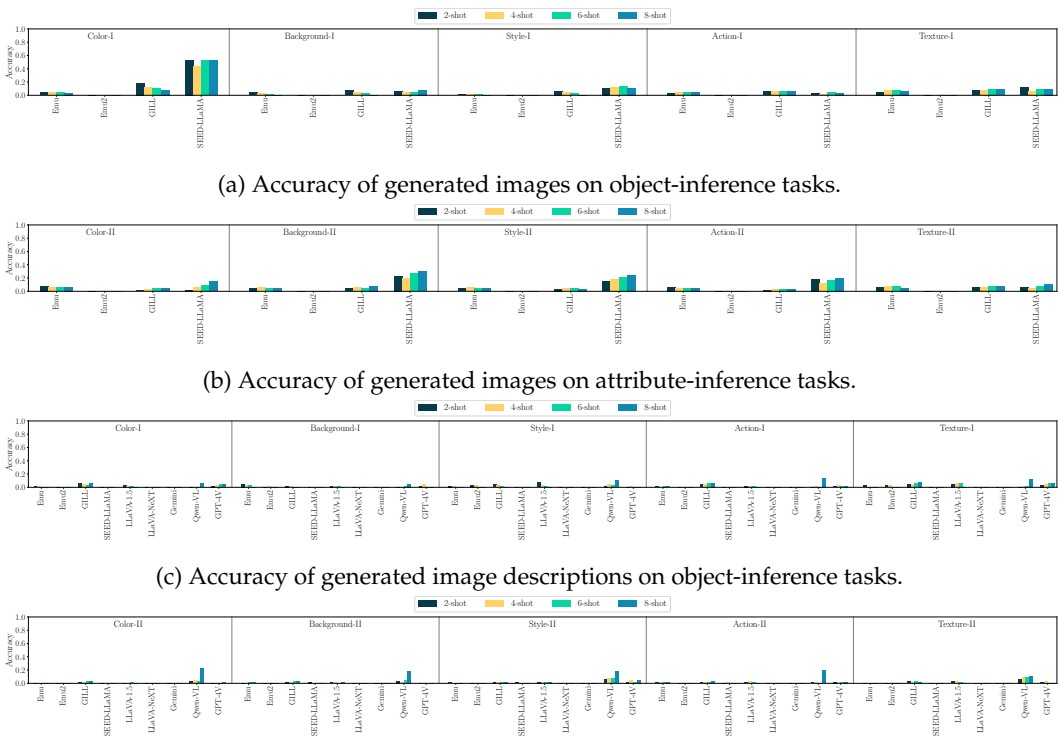

(a) Accuracy of generated images on object-inference tasks.

(b) Accuracy of generated images on attribute-inference tasks.

(c) Accuracy of generated image descriptions on object-inference tasks.

(d) Accuracy of generated image descriptions on attribute-inference tasks.

Figure 13: **Performance of considered MLLMs on the challenging version of the CoBSAT dataset with misleading information in the textual inputs.** We evaluate the T2I-ICL performance of various MLLMs with 2,4,6,8 in-context demonstrations. Low performance is observed across almost all evaluated MLLMs on this variant of our dataset, indicating a limited capacity of existing MLLMs in filtering out misleading information from prompts. [2]

### F.1.6  A Challenging Version of CoBSAT

We also investigate a more challenging task type that introduces misleading information into the textual input. This aims to evaluate whether MLLMs can accurately identify and ignore irrelevant information. Note that Claude is not considered in this experiment.

**Prompt Design.**  To this end, we consider inputs of the form $(x, \tilde{\theta})$ instead of just $x$, where $\tilde{\theta} \in \Theta$ represents the misleading information that does not affect the output $y$. Here is an example with 4-shot input when $\theta =$ "car:"

$$\text{``} \underbrace{\underbrace{\text{red}}_{x_1} \underbrace{\text{box:}}_{\tilde{\theta}_1} \underbrace{\text{[Image: \textbf{red car}]}}_{y_1}}_{\text{example 1}} \underbrace{\underbrace{\text{blue}}_{x_2} \underbrace{\text{chair:}}_{\tilde{\theta}_2} \underbrace{\text{[Image: \textbf{blue car}]}}_{y_2}}_{\text{example 2}} \underbrace{\underbrace{\text{yellow}}_{x_3} \underbrace{\text{leaf:}}_{\tilde{\theta}_3} \underbrace{\text{[Image: \textbf{yellow car}]}}_{y_3}}_{\text{example 3}} \underbrace{\underbrace{\text{black}}_{x_4} \underbrace{\text{book:}}_{\tilde{\theta}_4} \underbrace{\text{[Image: \textbf{black car}]}}_{y_4}}_{\text{example 4}} \underbrace{\underbrace{\text{green}}_{x_2} \underbrace{\text{bag:}}_{\tilde{\theta}_2}}_{\text{query}} \text{.''}$$

For prompt generation, we base it on the original prompt design, but with an added twist: for each prompt created with a sampled latent variable $\theta$, we introduce a misleading instruction for each example within the prompt. This is done by sampling the misleading information $\tilde{\theta} \in \Theta / \{\theta\}$ without replacement, thereby adding an extra layer of complexity to the task. In the case of prompts with misleading textual inputs, for each prompt with a sampled $\theta$, we further sample $\tilde{\theta} \in \Theta / \{\theta\}$ without replacement for each example.

---

[2]Warning: it should be noted that Emu2 has results for 2 and 4-shot scenarios, but results for 6 and 8-shot scenarios are unavailable due to resource constraints, as Emu2 demands excessive memory.

**Results.**    Figure 13 illustrates the performance of MLLMs on this challenging version of the CoBSAT dataset. We note a significantly poor performance across all MLLMs, with the exception of SEED-LLaMA in image generation. However, even SEED-LLaMA's performance shows a decline in most tasks compared to those in the CoBSAT dataset, which is visualized in Figure 4. These results suggest that current MLLMs also heavily rely on textual instructions and struggle to filter out misleading information. We anticipate this to be a challenging task for future MLLMs to overcome.

### F.2    Understanding Challenges in T2I-ICL (Detailed Version of Sec. 6)

In this section, we add more comprehensive details related to the experiments in Sec 6, to better understand the challenges in T2I-ICL. Our further experiments will mainly explore SEED-LLaMA, Gemini, and Qwen-VL. Previously, we identified SEED-LLaMA as the leading free model for image generation, whereas Gemini and Qwen-VL excel as the top free models for image description generation scenarios.

#### F.2.1    Is Multimodality a Primary Challenge in T2I-ICL?

In Sec. 5, we find that SEED-LLaMA and Qwen-VL achieve only around or less than 50% accuracy on most tasks. This is in contrast to the impressive results their underlying LLM demonstrates in Textual ICL (T-ICL) (Touvron et al., 2023; Bai et al., 2023a). In this experiment, our objective is to determine whether multimodality is the primary cause of this reduced performance, or whether MLLMs intrinsically struggle with these tasks even for T-ICL.

**Prompt Design.**    In this part, we evaluate SEED-LLaMA's capability in image generation and Qwen-VL and Gemini's proficiency in image description generation by modifying the prompts to be entirely textual. We achieve this by replacing every image in the prompts with corresponding detailed descriptions, which are initially created by LLaVA and ChatGPT. These descriptions are then reviewed and corrected by humans to ensure their accuracy. For example, in the first example depicted in Figure 2, the image [Image: **red car**] is replaced with a descriptive text: *"The image portrays a red Volkswagen Golf R, a compact sports car, stationed on a wet road under a dark sky, with its vivid red color prominently contrasting the background."* Furthermore, to guide SEED-LLaMA in generating images rather than their descriptions, we append the following instruction at the beginning of the prompt: *"We provide a few examples, each with an input, and an output of the image description. Based on the examples, predict the next image description and visualize it."* Similarly, for Qwen-VL and Gemini, we include an instruction: *"We provide a few examples, each with an input, and an output of the image description. Based on the examples, predict the next image description,"* with the focus on predicting the next image description without the visualization component. This distinction aims to direct SEED-LLaMA towards image generation, whereas Qwen-VL and Gemini are instructed to generate image descriptions.

**Results.**    The results are reported in Table 7. For 2-shot cases, SEED-LLaMA exhibits similar accuracies in both T2I-ICL and T-ICL, but in 4-shot instances, T-ICL surpasses T2I-ICL in eight out of ten tasks. The disparity is even more evident for Qwen-VL and Gemini; under T-ICL, it significantly outperforms T2I-ICL, especially in 4-shot situations. These findings confirm our first hypothesis, indicating that multimodality is the primary challenge in T2I-ICL.

#### F.2.2    Is the Image Generation a Primary Challenge in T2I-ICL?

To verify the second hypothesis, that image generation itself presents a primary challenge in T2I-ICL, we conduct an experiment with 0, 2, and 4-shot image generation tasks, with textual inputs updated as precise labels. For example, in the initial scenario from Figure 2, the terms "White," "Blue," and "Red" are updated to "White car," "Blue car," and "Red car," respectively. For this experiment, we exclude MLLMs that do not generate images. Instead, we focus on MLLMs that are capable of generating images, including Emu, GILL, and SEED-LLaMA.

| Model | Shot | Method | Object-Inference Task | | | | | Attribute-Inference Task | | | | |
|---|---|---|---|---|---|---|---|---|---|---|---|---|
| | | | Color-I | Background-I | Style-I | Action-I | Texture-I | Color-II | Background-II | Style-II | Action-II | Texture-II |
| SEED-LLaMA | 2 | T2I-ICL | **_.680_** | .348 | .203 | .182 | .196 | .287 | **_.467_** | **_.297_** | **_.261_** | .163 |
| | | T-ICL | .614 | **.380** | **.246** | **.279** | **.265** | **.531** | .315 | .206 | .184 | **.192** |
| | 4 | T2I-ICL | .482 | .211 | .141 | .053 | .122 | .252 | .076 | **.268** | **.207** | .105 |
| | | T-ICL | **.584** | **_.404_** | **_.289_** | **_.317_** | **_.276_** | **_.667_** | **.343** | .266 | .195 | **_.228_** |
| Qwen-VL | 2 | T2I-ICL | .475 | .313 | .378 | .464 | .338 | **.457** | .379 | .258 | .388 | .316 |
| | | T-ICL | **.854** | **.822** | **.692** | **.892** | **.679** | .272 | **.409** | **.559** | **.428** | **.431** |
| | 4 | T2I-ICL | .560 | .459 | .571 | .679 | .454 | .568 | .364 | .341 | .546 | .434 |
| | | T-ICL | **_.973_** | **_.851_** | **_.857_** | **_.972_** | **_.890_** | **_.740_** | **_.805_** | **_.793_** | **_.719_** | **_.827_** |
| Gemini | 2 | T2I-ICL | .865 | .794 | .315 | .517 | .704 | **.555** | **.583** | .360 | **.725** | .340 |
| | | T-ICL | **.979** | **.907** | **.692** | **.895** | **.764** | .150 | .410 | **.645** | .468 | **.361** |
| | 4 | T2I-ICL | .904 | .908 | .540 | .737 | .861 | .709 | .773 | .484 | **_.818_** | .553 |
| | | T-ICL | **_.988_** | **_.965_** | **_.888_** | **_.965_** | **_.927_** | **_.777_** | **_.780_** | **_.835_** | .783 | **_.812_** |

Table 7: **Comparison of Text-to-Image ICL (T2I-ICL) versus Textual ICL (T-ICL) accuracy on our dataset.** To perform T-ICL on our dataset, we replace all images in the prompts with their corresponding detailed descriptions. Underlined numbers indicate the highest accuracy achieved for each model and task across various shot numbers, while bold numbers indicate the highest accuracy for each specific combination of model, task, and shot count. In this experiment, we focus on three MLLMs: SEED-LLaMA, which is used for image generation; Qwen-VL and Gemini, utilized for generating image descriptions. MLLMs demonstrate notably superior performance in T-ICL compared to T2I-ICL, particularly in the 4-shot scenario.

| Model | Shot | Precise Textual Inputs | Object-Inference Task | | | | | Attribute-Inference Task | | | | |
|---|---|---|---|---|---|---|---|---|---|---|---|---|
| | | | Color-I | Background-I | Style-I | Action-I | Texture-I | Color-II | Background-II | Style-II | Action-II | Texture-II |
| SEED-LLaMA | 0 | ✓ | .730 | _.456_ | _.356_ | _.264_ | .275 | .582 | .314 | .298 | .207 | _.286_ |
| | 2 | ✗ | .680 | .348 | .203 | .182 | .196 | .287 | .467 | .297 | .261 | .163 |
| | | ✓ | **_.801_** | **.409** | **.241** | **.192** | **_.326_** | **.385** | **_.485_** | **_.393_** | **_.317_** | **.268** |
| | 4 | ✗ | .482 | .211 | .141 | .053 | .122 | .252 | .076 | .268 | .207 | .105 |
| | | ✓ | **.669** | **.318** | **.284** | **.161** | **.286** | **_.608_** | **.441** | **.299** | **.278** | .248 |
| Emu | 0 | ✓ | _.094_ | _.102_ | .052 | .064 | .047 | .054 | .075 | .069 | _.160_ | .028 |
| | 2 | ✗ | **.065** | .051 | .057 | .052 | .078 | .062 | **.109** | **.081** | **.092** | **.074** |
| | | ✓ | .050 | **.086** | **.101** | **_.070_** | **_.116_** | **.122** | .087 | .074 | .079 | .060 |
| | 4 | ✗ | **.063** | .018 | .045 | .048 | **.097** | .037 | **_.122_** | **_.109_** | **.077** | **_.088_** |
| | | ✓ | .061 | **.069** | **_.136_** | **.056** | .091 | **_.136_** | .083 | .076 | .072 | .081 |
| GILL | 0 | ✓ | _.341_ | _.286_ | _.244_ | _.135_ | _.237_ | _.297_ | _.223_ | _.178_ | _.176_ | _.226_ |
| | 2 | ✗ | .171 | .054 | .069 | .063 | .074 | .010 | .043 | .024 | **.022** | .040 |
| | | ✓ | **.245** | **.112** | **.100** | **.066** | **.108** | **.023** | **.092** | **.054** | .021 | **.075** |
| | 4 | ✗ | .106 | .044 | .041 | **.073** | .087 | .022 | .059 | .044 | .032 | .067 |
| | | ✓ | **.178** | **.084** | **.125** | .064 | **.133** | **.072** | **.092** | **.055** | **.037** | **.095** |

Table 8: **Accuracy comparison on SEED-LLaMA, Emu, and GILL: with or without providing precise textual inputs.** Bold numbers represent the highest accuracy for each task and shot count, comparing scenarios with and without descriptive textual inputs. Underlined numbers indicate the highest accuracy for each task across various shots.

**Results.** Table 8 presents a comparative analysis of three considered MLLMs's performance in T2I-ICL, both with and without the inclusion of precise textual inputs. We observe that in scenarios with both 2 and 4 shots, the presence of precise textual inputs leads to significantly higher accuracy in image generation compared to when these inputs are absent for SEED-LLaMA and GILL, whereas Emu's performance does not follow a discernible trend. Crucially, the analysis shows that, even with precise inputs, all models sustain a comparable level of performance across different tasks, with accuracies remaining under 50% in most cases. This suggests that the task of image generation remains a considerable challenge for contemporary MLLMs, affecting their efficacy on the CoBSAT dataset.

## F.3 Enhancing MLLMs' T2I-ICL Capabilities (Detailed Version of Sec. 7)

In this section, we supplement Sec 7 with additional experimental details, discussion, and expanded experiment results, exploring techniques that could potentially enhance the performance of MLLMs in T2I-ICL.

### F.3.1 Fine-tuning MLLMs on CoBSAT

In this experiment, we investigate the impact of fine-tuning MLLMs on our dataset in improving its T2I-ICL capabilities. We focus on SEED-LLaMA Qwen-VL for this investiga-

tion. Consequently, we compare the T2I-ICL performance of the pretrained-only version of Qwen-VL nad SEED-LLaMA with their corresponding variant that is fine-tuned on our dataset.

**Training Setup.** We fine-tune two instances of both SEED-LLaMA and Qwen-VL, one on a 2-shot dataset and the other on a 4-shot dataset, and then compare their performances with their non-fine-tuned counterparts on the T2I-ICL test set. For both models, we fine-tune their LLM backbone only using LoRA (Hu et al., 2022) with a rank of 64, a weight decay of 0.1, and a warm-up ratio of 0.01 for 5 epochs.

**Training and Test Sets.** We employ two distinct strategies for splitting the training and test datasets. In the first strategy, the training set comprises prompts from all ten themes, ensuring that attributes and objects in the test set are not previously exposed in any training prompts. In the second strategy, the training set excludes the themes that are included in the test set, enabling us to assess whether a model fine-tuned on specific tasks can generalize to other tasks. Note that the tasks configured by the second approach are inherently more challenging.

(Data Split A) The training and test sets are constructed by splitting the predefined lists of text inputs and latent variables (from Table 5, denoted as $\mathcal{X}$ and $\Theta$) into training ($\mathcal{X}_{\text{train}}$, $\Theta_{\text{train}}$) and testing ($\mathcal{X}_{\text{test}}$, $\Theta_{\text{test}}$) subsets for each task, in a 1:1 ratio. Therefore, all the in-context demonstrations and textual queries in the test sets are unseen from the training set. For the training set, we create the dataset by considering all possible combinations and sequences of text inputs from $\mathcal{X}_{\text{train}}$ and latent variables $\Theta_{\text{train}}$ across all tasks. We fine-tune Qwen-VL on this unified training set containing the prompts from all tasks. For the testing set, we generate 250 prompts for each shot across various tasks. Each prompt is obtained by randomly sampling $\theta \in \Theta_{\text{test}}$ and $(x_n)_{n=1}^{N+1} \in \mathcal{X}_{\text{test}}^{N+1}$, which are then paired with the corresponding collected images $(y_n)_{n=1}^{N}$, where $y_n \sim f_\theta(x_n)$. This process results in $N$ in-context demonstrations $(x_n, y_n)_{n=1}^{N}$ and a single textual query $x_{N+1}$.

(Data Split B) In this data split, we intensify the challenge by increasing the disparity between the training and testing distributions. Instead of merely including unseen objects and attributes from the same themes in the testing dataset, this split introduces unseen themes. For example, the results shown in Table 9 for split B on color-themed tasks (i.e., Color-I and Color-II) are derived from a model fine-tuned on the other four themes (i.e., eight tasks). Thus, the model is not fine-tuned on color-themed tasks but is evaluated on them. This method is uniformly applied across all themes: each theme is evaluated using a model fine-tuned on tasks from the other four themes. Thus, the training set includes four themes, while the test set comprises a different, fifth theme. Consequently, the results in Table 9 for split B reflect different models, each fine-tuned on a distinct training set.

**Results.** The results are summarized in Table 9. For data split A, both models demonstrate significant improvements in T2I-ICL performance following fine-tuning. In the more challenging tasks defined by data split B, SEED-LLaMA demonstrates strong generalization to unseen tasks after fine-tuning, while Qwen-VL exhibits more difficulty in generalizing. Overall, these results suggest that fine-tuning MLLMs on a T2I-ICL dataset generally enhances their overall T2I-ICL capabilities. Example output images from the pre-trained and fine-tuned versions of SEED-LLaMA on split A are provided in Sec. H.2.

### F.3.2 Intergrating Chain-of-Thought with T2I-ICL

Another widely utilized method in prompt engineering is Chain-of-Thought (CoT) (Wei et al., 2022). This approach involves incorporating a simple instruction, such as "let's think step by step," prompting the model to sequentially generate concise sentences that outline the reasoning process, commonly referred to as reasoning chains or rationales. The chains are subsequently embedded into the subsequent prompt to obtain the final answer. CoT has been particularly effective in enhancing performance, especially for complex reasoning tasks,

| Model | Shot | Fine-tuned | Split | Object-Inference Task | | | | | Attribute-Inference Task | | | | |
|---|---|---|---|---|---|---|---|---|---|---|---|---|---|
| | | | | Color-I | Background-I | Style-I | Action-I | Texture-I | Color-II | Background-II | Style-II | Action-II | Texture-II |
| SEED-LLaMA | 2 | ✗ | - | .636 | .292 | .088 | .196 | .108 | .360 | .536 | .164 | **.196** | .080 |
| | | ✓ | A | **_.776_** | **_.540_** | _.164_ | **_.284_** | **_.208_** | _.468_ | **_.588_** | .108 | .192 | **_.140_** |
| | | ✓ | B | _.752_ | _.484_ | **_.208_** | _.272_ | _.200_ | **_.568_** | .376 | **_.240_** | .180 | _.104_ |
| | 4 | ✗ | - | .612 | .360 | .092 | .044 | .048 | .380 | .532 | **.140** | **.196** | .148 |
| | | ✓ | A | **_.784_** | _.516_ | .152 | .160 | .172 | _.504_ | **_.564_** | .104 | .192 | _.200_ |
| | | ✓ | B | _.748_ | **_.556_** | **_.208_** | **_.256_** | **_.244_** | **_.616_** | .488 | .112 | .132 | **_.216_** |
| Qwen-VL | 2 | ✗ | - | .540 | .236 | **.248** | .412 | .372 | .276 | .244 | .112 | .232 | .224 |
| | | ✓ | A | **_.852_** | **_.744_** | .212 | **_.856_** | **_.532_** | _.516_ | **_.344_** | _.148_ | **_.520_** | **_.284_** |
| | | ✓ | B | _.708_ | _.552_ | _.376_ | .308 | .328 | **_.592_** | _.272_ | **_.224_** | .212 | .172 |
| | 4 | ✗ | - | .680 | .492 | **.448** | .228 | .556 | .512 | **.448** | **.240** | .320 | .420 |
| | | ✓ | A | **_.876_** | _.604_ | .216 | **_.812_** | **_.588_** | _.696_ | .308 | .088 | **_.656_** | **_.480_** |
| | | ✓ | B | _.812_ | **_.728_** | .300 | _.352_ | .464 | **_.740_** | .380 | .240 | .212 | .308 |

Table 9: **T2I-ICL accuracy comparison of pretrained-only versus fine-tuned (FT) MLLMs.** Underlined numbers signify instances where the fine-tuned model surpasses the pretrained model in the same scenario, while bold numbers indicate the top performance for each shot across various methods within their tasks.

when applied to large-scale models (Wei et al., 2022). In our experiment, we investigate the impact of integrating CoT on the T2I-ICL performance of MLLMs.

**Prompt Design.** We employ a two-step inference process utilizing two distinct prompts. The initial prompt builds upon the default examples showcased in Figure 2. To this, we prepend the statement, *"We provide a few examples, each of which is an input-output pair where the output is a description of the image associated with the input. Based on the examples, the task is to predict the next image description.\n\n\n"* This is placed at the beginning of the prompt. Additionally, we append, *"\n\n\n Before predicting the next image, let's first think step by step and analyze the relationship between the text input and image output in each example.\n\n\n"* to the end of the prompt.

Following the MLLMs' responses, the second prompt comes into play. It includes the first prompt and the MLLM's response as part of the new prompt and extends it with, *"\n\n\n Based on the analysis, please generate the next image for the request 'red: ' "* for the image generation scenario, and *"\n\n\n Based on the analysis, please describe what the next image should look like for the request 'red: ' "* for the image description generation scenario when the textual query is 'red.' In each case, 'red' is replaced with the respective textual query according to different prompts.

| Model | Shot | Method | Object-Inference Task | | | | | Attribute-Inference Task | | | | |
|---|---|---|---|---|---|---|---|---|---|---|---|---|
| | | | Color-I | Background-I | Style-I | Action-I | Texture-I | Color-II | Background-II | Style-II | Action-II | Texture-II |
| SEED-LLaMA | 2 | T2I-ICL | .680 | **.348** | .203 | **.182** | .196 | **.287** | **_.467_** | **.297** | .261 | **.163** |
| | | CoT + T2I-ICL | **_.781_** | .179 | **.206** | .167 | **_.222_** | .179 | .389 | .195 | **.300** | .154 |
| | 4 | T2I-ICL | .482 | .211 | .141 | .053 | .122 | .252 | .076 | .268 | .207 | .105 |
| | | CoT + T2I-ICL | **_.650_** | **_.353_** | **_.244_** | **_.242_** | **.208** | **_.303_** | .370 | **_.335_** | .241 | **_.171_** |
| Qwen-VL | 2 | T2I-ICL | **.475** | .313 | .378 | **.464** | .338 | **.457** | **.379** | .258 | **.388** | **.316** |
| | | CoT + T2I-ICL | .281 | **_.494_** | **.387** | .217 | **.363** | .150 | .349 | **.260** | .176 | .181 |
| | 4 | T2I-ICL | **_.560_** | **.459** | **_.571_** | **_.679_** | .454 | **_.568_** | .364 | .341 | **_.546_** | **_.434_** |
| | | CoT + T2I-ICL | .548 | .379 | .274 | .404 | **_.573_** | .207 | **_.690_** | **_.409_** | .424 | .340 |
| Gemini | 2 | T2I-ICL | .865 | .794 | .315 | .517 | .704 | .555 | .583 | .360 | **.725** | .340 |
| | | CoT + T2I-ICL | **.938** | **.861** | **.647** | **.882** | **.731** | **.655** | **.908** | **_.672_** | .701 | **.445** |
| | 4 | T2I-ICL | .904 | .908 | .540 | .737 | .861 | .709 | .773 | **_.484_** | .818 | .553 |
| | | CoT + T2I-ICL | **_.986_** | **_.957_** | **_.799_** | **_.916_** | **_.945_** | **_.917_** | **_.977_** | .293 | **_.897_** | **_.755_** |

Table 10: **Assessing the impact of Chain-of-Thought (CoT) prompting on T2I-ICL.** The evaluation metric is accuracy, with the numbers in bold highlighting the highest accuracy achieved for each model, number of shots, and task, and underlined numbers indicate the highest accuracy achieved for each model and task across different numbers of shots. In this experiment, we evaluate three MLLMs: SEED-LLaMA for image generation, and Qwen-VL and Gemini for image description generation. Our findings reveal that CoT significantly improves Gemini's performance. For SEED-LLaMA and Qwen-VL, the enhancement offered by CoT is ambiguous in 2-shot scenarios. However, in 4-shot instances, CoT markedly enhances the performance of SEED-LLaMA, while it still shows no benefit for Qwen-VL.

**Results.** The results are reported in Table 10. With the integration of CoT, Gemini shows better performance across the most of tasks in both 2-shot and 4-shot scenarios. Similarly, SEED-LLaMA shows significant improvement in T2I-ICL performance across all ten tasks

in the 4-shot scenario. Conversely, for Qwen-VL, no concrete evidence suggests that CoT enhances its T2I-ICL performance. In fact, we find that Qwen-VL often avoids providing definitive answers in the second step of making predictions, and responds with general statements like *"Given the request 'black:', we can infer that the image output should be related to a black object or theme. However, without more specific information, it's difficult to determine the exact relationship between the text input and image output. Without additional context, it's impossible to accurately predict the next image."* Therefore, in many instances, standard T2I-ICL without CoT appears to outperform the version integrated with CoT for Qwen-VL. Exploring additional prompt engineering methods such as self-consistency sampling (Wang et al., 2023b) or Tree-of-Thought (Yao et al., 2023) to elicit more concrete responses from Qwen-VL is a possibility. Specifically, self-consistency sampling involves generating multiple outputs at a non-zero temperature setting and selecting the most appropriate one from these options. On the other hand, Tree-of-Thought expands upon CoT by considering multiple lines of reasoning at each step. However, such investigations fall outside the scope of this paper, and we identify it as one of the interesting future directions. We provide example conversations of integrating CoT and T2I-ICL in Sec. H.3.

### F.3.3 Articulating the Text-to-Image Relationship in Prompts

In our dataset, the goal is to check if the MLLMs are able to learn the mapping from the textual input and the visual output based on the in-context demonstrations. In this experiment, we investigate the performance of MLLMs on T2I-ICL if we explicitly write down this relationship in the text prompt. For instance, for the Color-I task, we directly add the following sentence to the beginning of the prompt: *"Please identify the common main object in the images, and generate another image of this object in the requested color."* The detailed prompts for all tasks are provided in Sec. D.1.2.

| Model | Shot | Explicit Instruction | Object-Inference Task | | | | | Attribute-Inference Task | | | | |
|---|---|---|---|---|---|---|---|---|---|---|---|---|
| | | | Color-I | Background-I | Style-I | Action-I | Texture-I | Color-II | Background-II | Style-II | Action-II | Texture-II |
| SEED-LLaMA | 2 | ✗ | .680 | .348 | .203 | .182 | .196 | **.287** | **.467** | .297 | .261 | .163 |
| | | ✓ | **.779** | **.391** | **.231** | **.301** | **.270** | .257 | .446 | **.350** | .249 | **.185** |
| | 4 | ✗ | .482 | .211 | .141 | .053 | .122 | .252 | .076 | .268 | .207 | .105 |
| | | ✓ | **.832** | **.408** | **.281** | **.318** | **.322** | **.388** | **.483** | **.406** | **.268** | **.228** |
| Qwen-VL | 2 | ✗ | **.475** | .313 | **.378** | .464 | **.338** | **.457** | .379 | .258 | .388 | .316 |
| | | ✓ | .407 | **.496** | .240 | **.516** | .300 | .240 | **.697** | **.317** | **.600** | **.373** |
| | 4 | ✗ | **.560** | **.459** | **.571** | **.679** | **.454** | **.568** | .364 | **.341** | **.546** | **.434** |
| | | ✓ | .315 | .291 | .341 | .475 | .473 | .277 | **.591** | .317 | .527 | .404 |
| Gemini | 2 | ✗ | **.865** | **.794** | .315 | .517 | **.704** | **.555** | .583 | **.360** | **.725** | **.340** |
| | | ✓ | .119 | .624 | **.553** | **.620** | .176 | .128 | **.735** | .155 | .373 | .118 |
| | 4 | ✗ | **.904** | **.908** | .540 | **.737** | **.861** | **.709** | .773 | **.484** | **.818** | **.553** |
| | | ✓ | .198 | .655 | **.564** | .675 | .356 | .125 | **.921** | .199 | .520 | .125 |

Table 11: **Effect of explicit instruction on T2I-ICL performance of MLLMs: articulating the text-to-image relationship in prompts.** The evaluation metric is accuracy, where underlined numbers denote the highest accuracy achieved by each model and task across varying shot numbers, and bold numbers represent the top accuracy for each specific combination of model, task, and shot count. This evaluation focuses on SEED-LLaMA for image generation, and Qwen-VL and Gemini for image description generation. We find that explicit instructions significantly enhance the T2I-ICL capability of SEED-LLaMA, especially in the 4-shot scenario. However, for Qwen-VL and Gemini, explicit instructions do not show similar performance gains.

**Results.**    We present the experiment results in Table 11. Results show that SEED-LLaMA's performance significantly improves with explicit instructions, surpassing its performance in T2I-ICL without instructions for seven out of ten tasks in the 2-shot cases and all tasks in the 4-shot cases. Notably, in the 4-shot case for the Color-I task, SEED-LLaMA achieves a high accuracy of 83.2% with explicit instructions, compared to only 48.2% without them. Furthermore, the performance of T2I-ICL with explicit instructions improves when moving from 2-shot to 4-shot scenarios, in contrast to the situation without explicit instructions where SEED-LLaMA's T2I-ICL performance declines as the number of demonstrations increases from 2 to 4. In contrast, Qwen-VL does not show comparable improvements, owing to reasons similar to those discussed in Sec. F.3.2, including the generation of irrelevant responses like *"Received."*. Similarly, Gemini also fails to demonstrate improvements. To be

more specific, we find that Gemini consistently ignores the textual query after articulating the text-to-image relationship in prompts. To handle these issues, more careful prompt engineering could be applied, although it is beyond the scope of this paper.

# G  Extended Discussion

This section is an expanded version of Sec 8, discussing the conclusion, limitations, and future works in greater detail.

## G.1  Conclusion

In this work, we identify an important yet underexplored problem — T2I-ICL, and explore the capability of MLLMs to solve it. To facilitate this investigation, we introduce CoBSAT, a comprehensive benchmark dataset encompassing ten tasks. Our experimental evaluation of MLLMs on this dataset reveals that while many MLLMs have difficulty in effectively learning from in-context demonstrations during text-to-image generation, a few MLLMs, such as GPT-4V, Qwen-VL, Gemini, Claude, and SEED-LaMA, show comparatively reasonable performance. Through further studies on free top-performing models SEED-LLaMA, Gemini, and Qwen-VL for both image and image description generation, we identify two key challenges in T2I-ICL: (i) the integration and understanding of multimodal information, evidenced by superior results achieved with textual ICL for the same tasks; and (ii, particularly for image generation models) the actual process of image creation, as even straightforward image requests with clear descriptions often yield suboptimal performances.

To improve MLLMs' performance in T2I-ICL, we carry out additional experimental studies. These studies suggest that fine-tuning and CoT can substantially enhance T2I-ICL capabilities. Meanwhile, it is worth noting that in our dataset, we intentionally exclude explicit task descriptions to assess whether MLLMs can autonomously adapt to tasks based solely on in-context demonstrations alone. In the ablation studies, we find that providing clearer task instructions might be a promising strategy for enhancing T2I-ICL performance. However, these prompting engineering strategies might need to be combined with others to achieve consistent improvements.

## G.2  Limitations and Future Works

While our study is an early attempt to explore the T2I-ICL benchmark dataset, many interesting questions remain open.

**Impact of Demonstration Selection on T2I-ICL Performance.**  Existing research in textual ICL has consistently demonstrated that the choice of demonstrations significantly influences ICL performances (Liu et al., 2022; Su et al., 2023; Rubin et al., 2022; Zhang et al., 2022b). In our study, we only employ random sampling to select in-context demonstrations. This opens an interesting question: to what extent does the selection of demonstrations affect T2I-ICL performance? Moreover, our evaluation primarily assesses whether MLLMs can accurately generate images with the current content, without a specific focus on the quality of these images. Another natural question arises: how significantly does the quality of images used in demonstrations influence the overall quality of the generated image output?

**Prompt Engineering Techniques for MLLMs.**  As discussed in Sec. 7, CoT demonstrates significant improvements in T2I-ICL performance for SEED-LLaMA and Gemini. However, the prompt sensitivity of models like Qwen-VL poses a challenge, as they tend to provide non-committal responses such as, *"Without additional context, it's impossible to accurately predict the next image."* This issue underscores the necessity of implementing more advanced prompt engineering techniques, including methods like Tree-of-Thought and self-consistency sampling, to address these limitations.

Once such issues are resolved, another interesting question arises: Is it feasible to enhance existing prompt engineering techniques with multimodal capabilities? For instance, while

Sec. 7 focuses on prompting MLLM to perform CoT through textual analysis (as further exemplified in Sec. F.3.2), expanding this approach to a multimodal CoT that integrates both textual analysis and image grounding could potentially yield better performance. These open questions are identified as interesting future directions.

**T2I-ICL for Image Editing.** One notable absence in our dataset is tasks related to image editing. For instance, in the Color-I task, the goal is to generate an image of a car in a color specified by the text query. In our evaluation, the car type and background in both the example images from the prompt and the newly generated image may differ. However, there is a growing need for image editing applications where the task is to alter specific attributes (e.g., the color of a car) in an otherwise unchanged image. For such tasks, selecting images that strictly adhere to the given criteria (identical images with only specific attributes or objects altered), coupled with the development of sophisticated metrics, is critical to assess these more complex challenges effectively.

**Exploring a Wider Range of Themes.** Our dataset primarily assesses MLLMs on elementary themes, incorporating a specific range of objects and attributes within narrowly defined categories. For instance, in the style task, we consider styles such as watercolor, sketch, pixel art, origami, and others. Nonetheless, the realm of styles in real-world applications is far more intricate and varied, extending to include oil painting, rococo, steampunk, and beyond. Additionally, our dataset encompasses a limited set of themes. There are also many other interesting themes such as counting. While it serves to test basic capabilities in T2I-ICL, a more comprehensive dataset, covering a broader spectrum of themes and a finer list of objects and attributes, will be crucial for evaluating more advanced model capabilities. In this scenario, the evaluation methodology may require refinement to more accurately identify the more fine-grained attributes and objects.

# H   Sample Outputs Generated by MLLMs

In this section, we provide sample outputs generated by the models under different scenarios. Specifically, Sec. H.1 contains a selection of sample prompts along with the corresponding responses generated by MLLMs across all ten tasks. Sec. H.2 displays selected sample prompts and the images produced by both the pretrained and fine-tuned SEED-LLaMA for all ten tasks. Furthermore, in Sec. H.3, we illustrate sample dialogues between users and MLLMs (including SEED-LLaMA and Qwen-VL) from our experiments that combine the CoT with T2I-ICL.

## H.1   Sample Prompts and Corresponding Outputs

Here, we showcase examples of outputs generated by various MLLMs, accompanied by their respective prompts.

**Image Generation**   For image generation, five examples are provided for each themed task, as depicted in Figures 14, 15, 16, 17, and 18 for color, background, style, action, and texture themes, respectively. Observing these figures, it is evident that SEED-LLaMA excels in image quality among the three MLLMs capable of image generation. Notably, SEED-LLaMA produces images that not only align with the true labels but also closely resemble the images in the prompts. For instance, in Figure 14, images with plain backgrounds in the prompts lead to similarly styled outputs.

**Image Description Generation**   In Figures 19, 20, 21, 22, and 23, we showcase two examples for each task, covering color, background, style, action, and texture themes. For brevity, some lengthy responses from MLLMs have been truncated in these figures, retaining only the key parts of the responses.

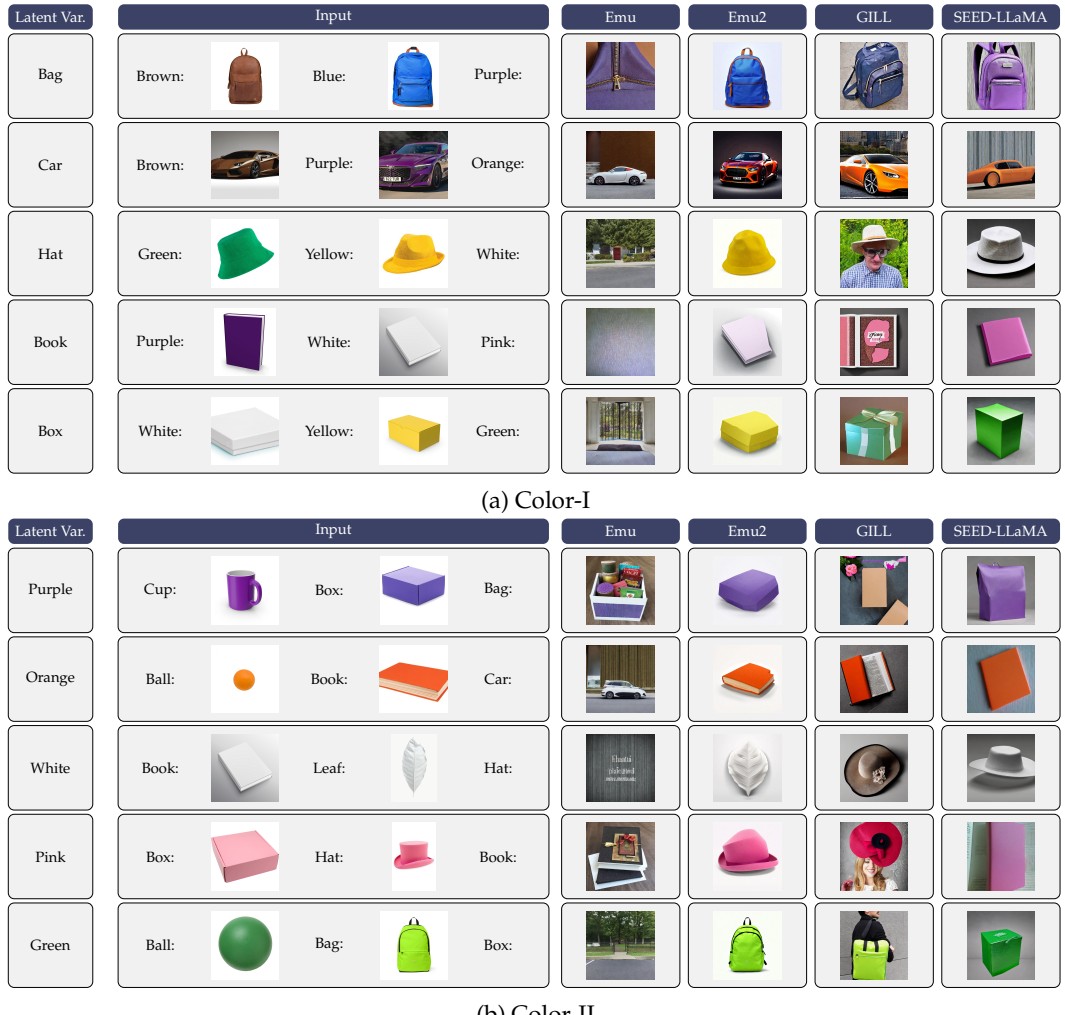

Figure 14: Examples of prompts and corresponding images generated by MLLMs for tasks within the Color theme.

## H.2 Sample Outputs from Fine-tuning SEED-LLaMA on CoBSAT

In Figure 24, we provide sample outputs generated by pretrained-only and fine-tuned SEED-LLaMA from the experiments described in Sec. 7 and F.3.1. We observe that the images generated by fine-tuned SEED-LLaMA generally align better with the expected output.

## H.3 Sample Outputs from Integrating CoT with T2I-ICL

In this section, we provide sample outputs generated by Qwen-VL and SEED-LLaMA from the experiment of integrating CoT with T2I-ICL, as detailed in Sec. 7. Specifically, Figures 25, 26, and 27 illustrate the example prompts and corresponding outputs generated by Qwen-VL for the tasks Color-I, Action-II, and Texture-II. Similarly, Figures 28, 29, and 30 display the example prompts and outputs produced by SEED-LLaMA for the tasks Color-I, Background-II, and Style-I.

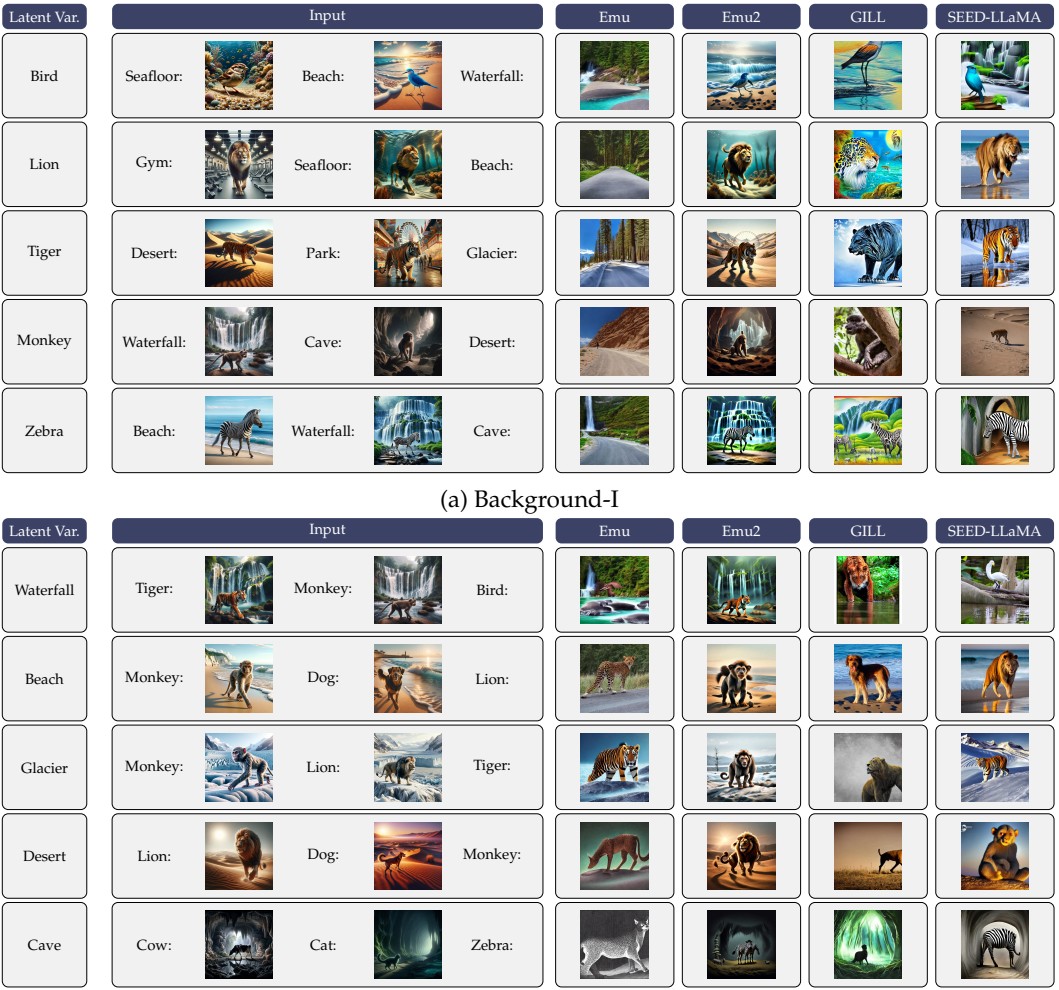

Figure 15: Examples of prompts and corresponding images generated by MLLMs for tasks within the Background theme.

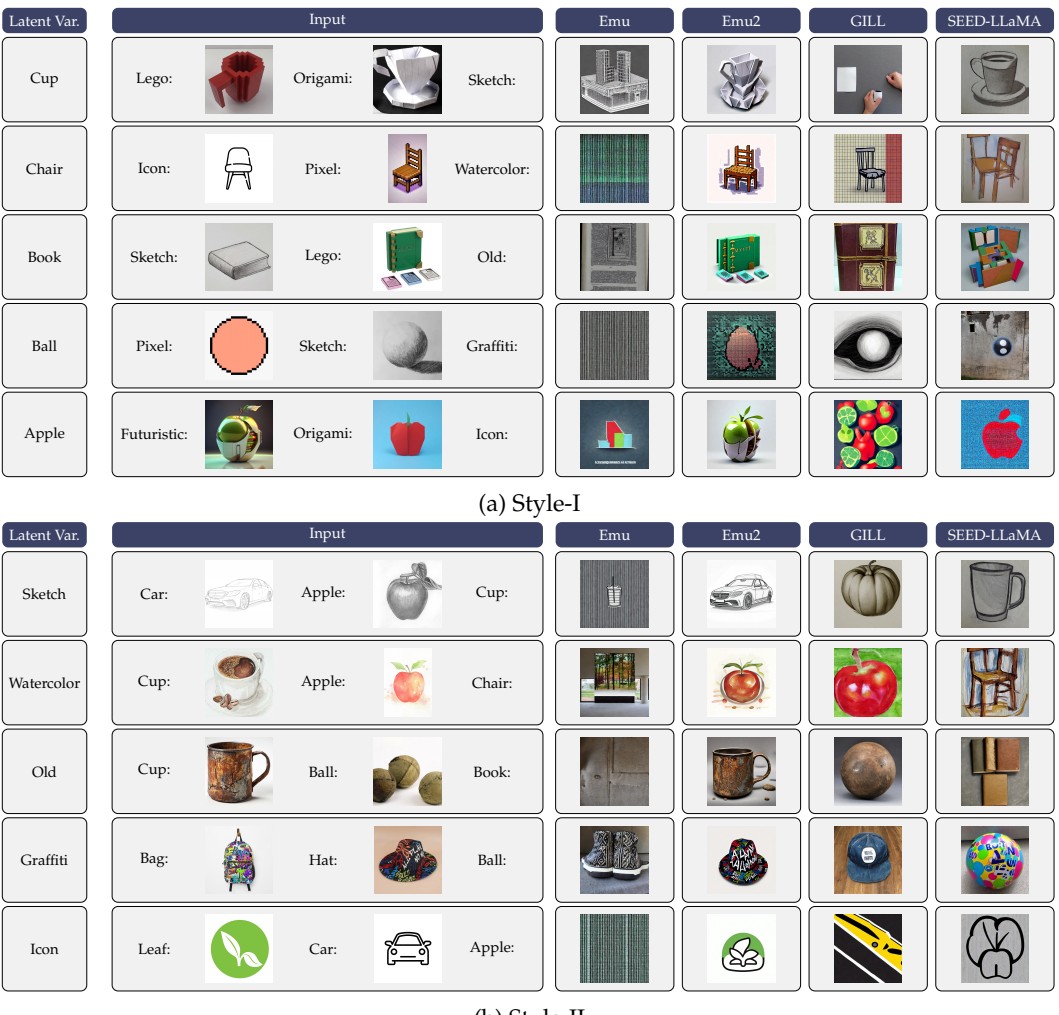

Figure 16: Examples of prompts and corresponding images generated by MLLMs for tasks within the Style theme.

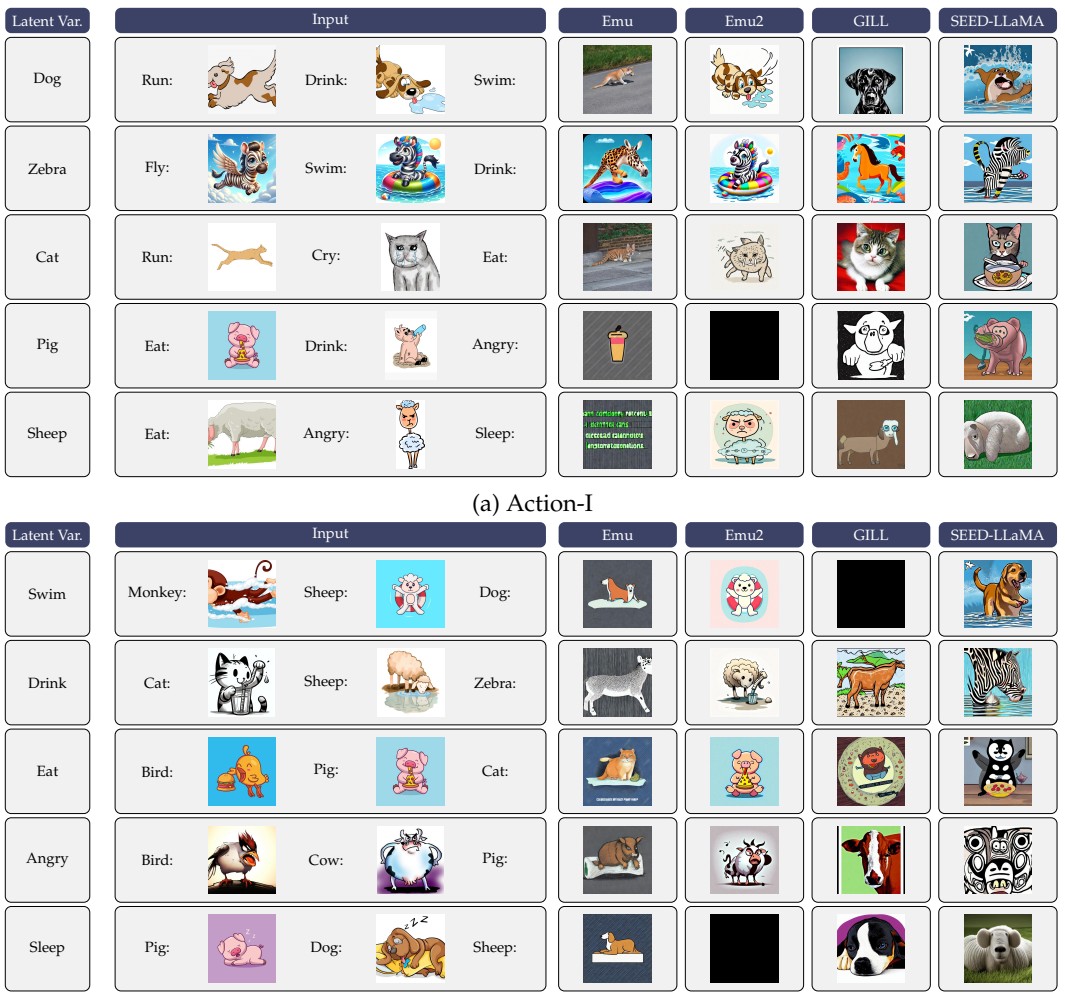

Figure 17: Examples of prompts and corresponding images generated by MLLMs for tasks within the Action theme.

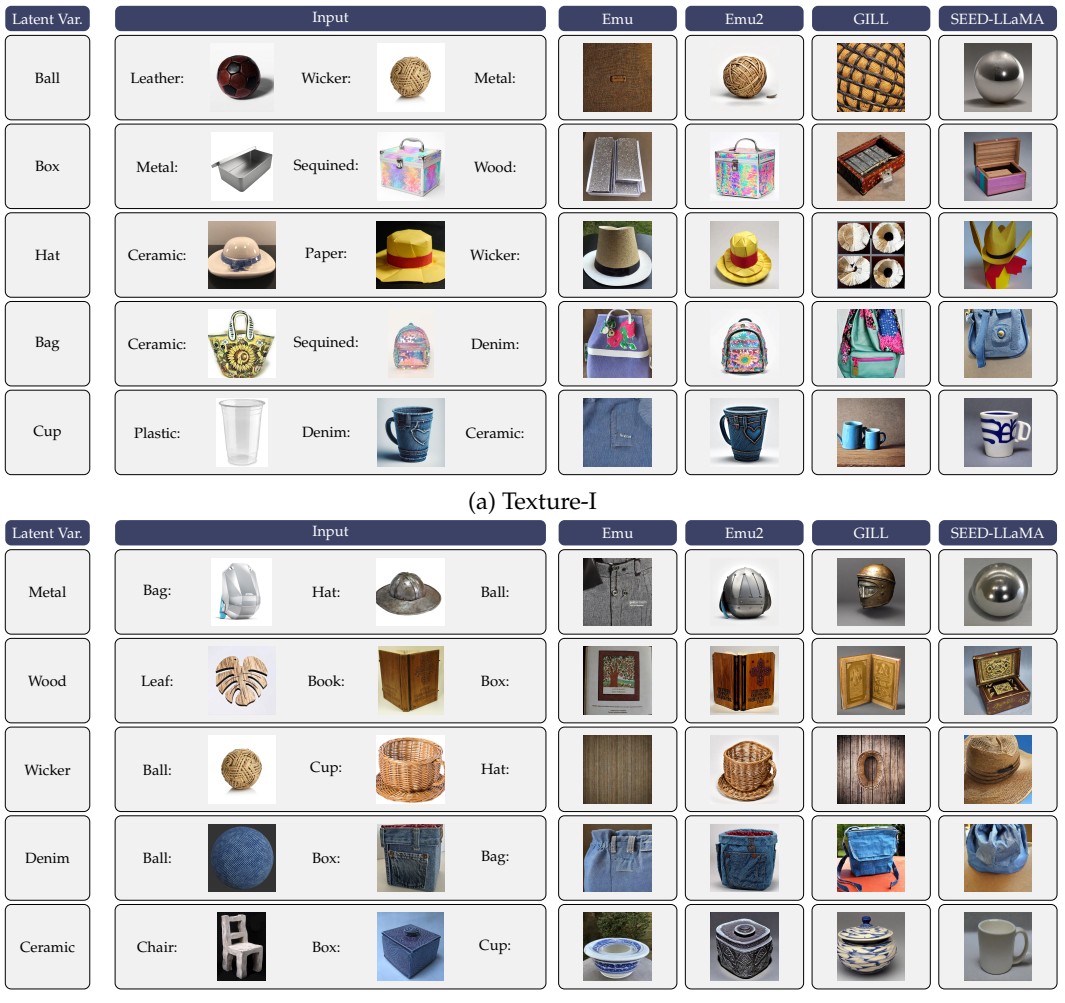

Figure 18: Examples of prompts and corresponding images generated by MLLMs for tasks within the Texture theme.

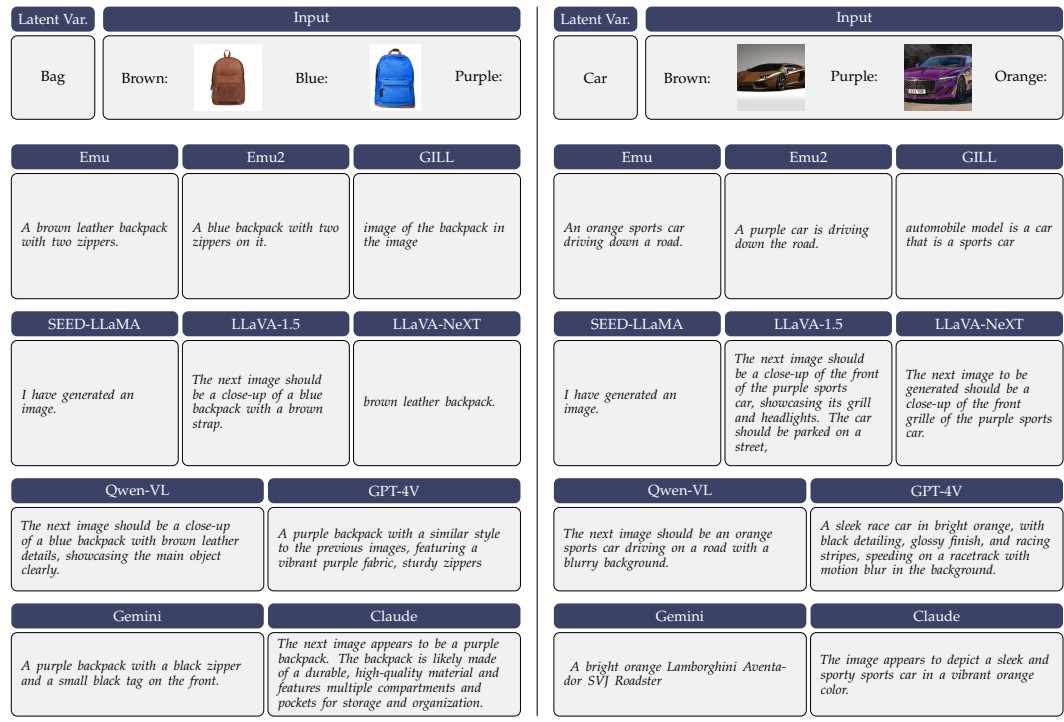

(a) Color-I

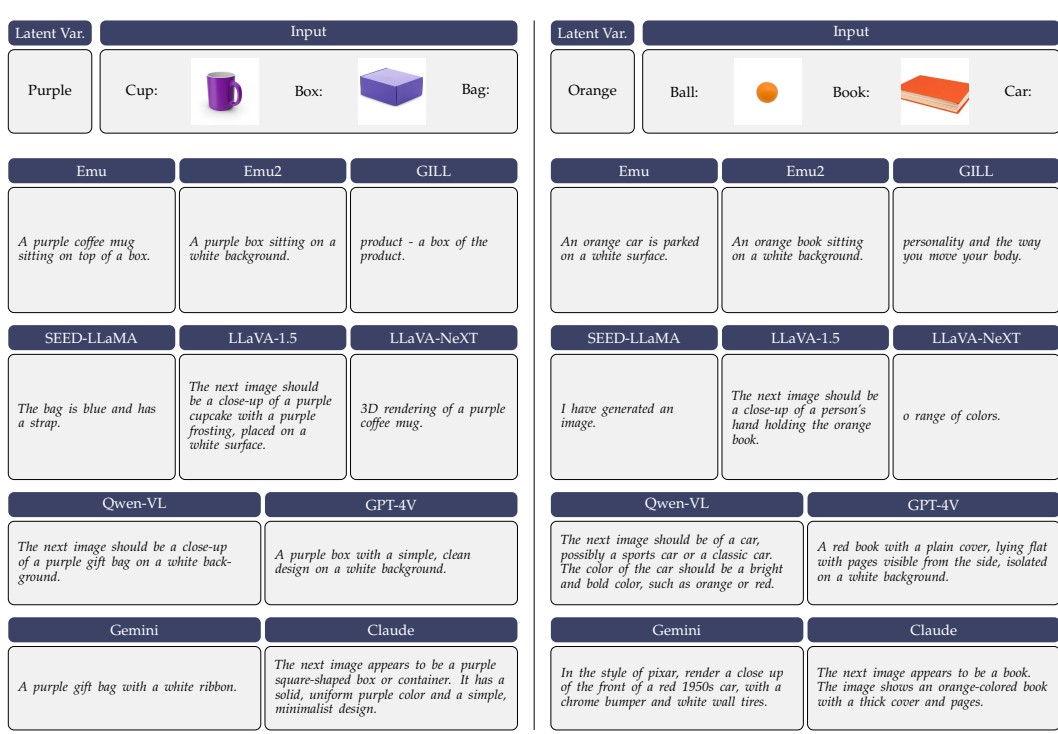

(b) Color-II

Figure 19: Examples of prompts and corresponding images description generated by MLLMs for tasks within the Color theme. For brevity, some lengthy responses from MLLMs have been truncated, retaining only the key parts of the responses.

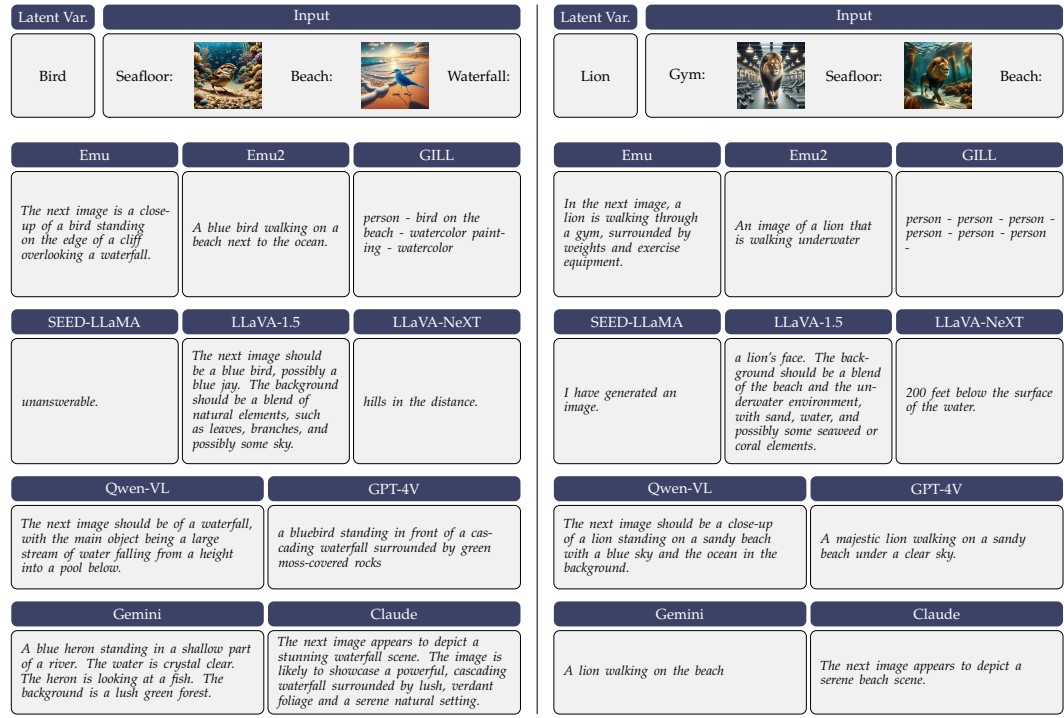

(a) Background-I

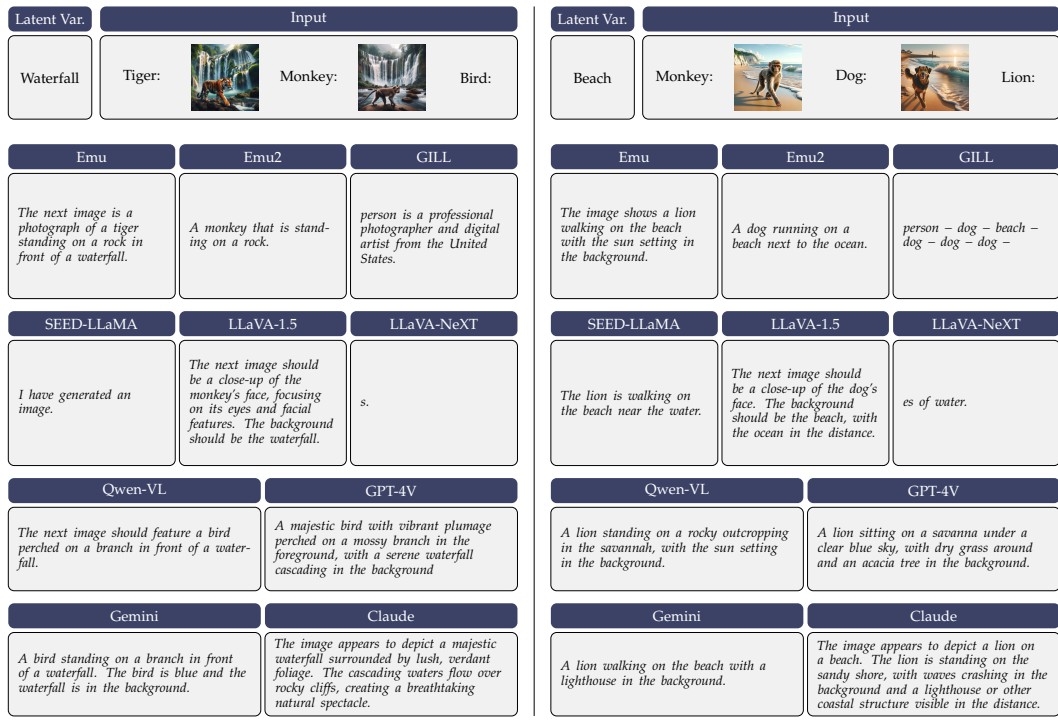

(b) Background-II

Figure 20: Examples of prompts and corresponding image descriptions generated by MLLMs for tasks within the Background theme. For brevity, some lengthy responses from MLLMs have been truncated, retaining only the key parts of the responses.

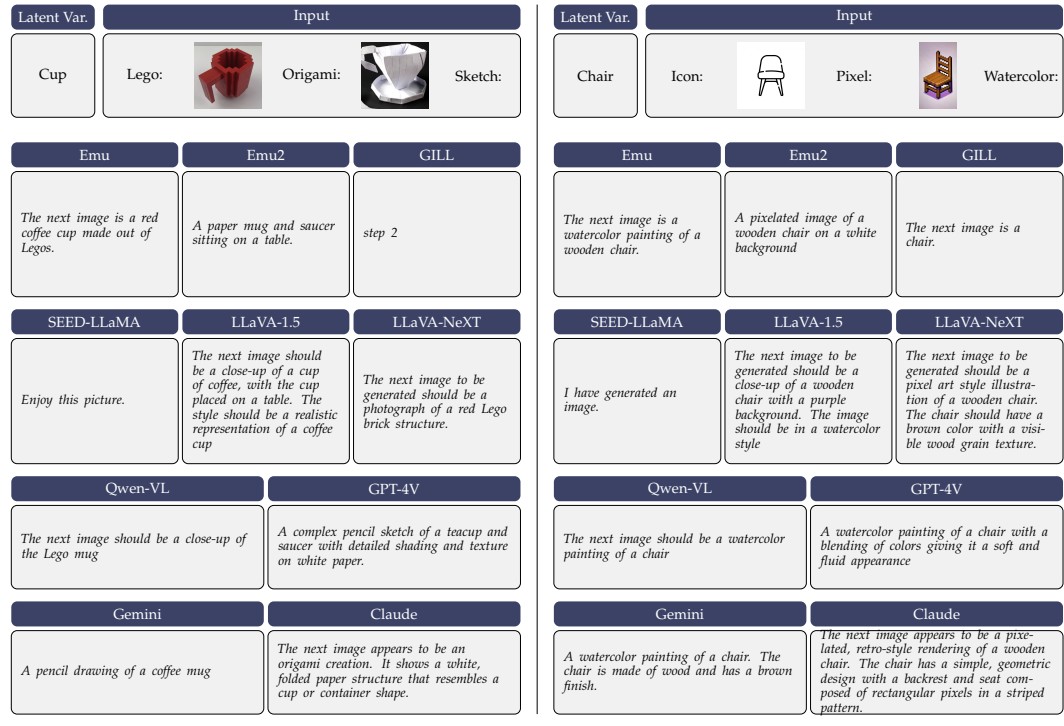

(a) Style-I

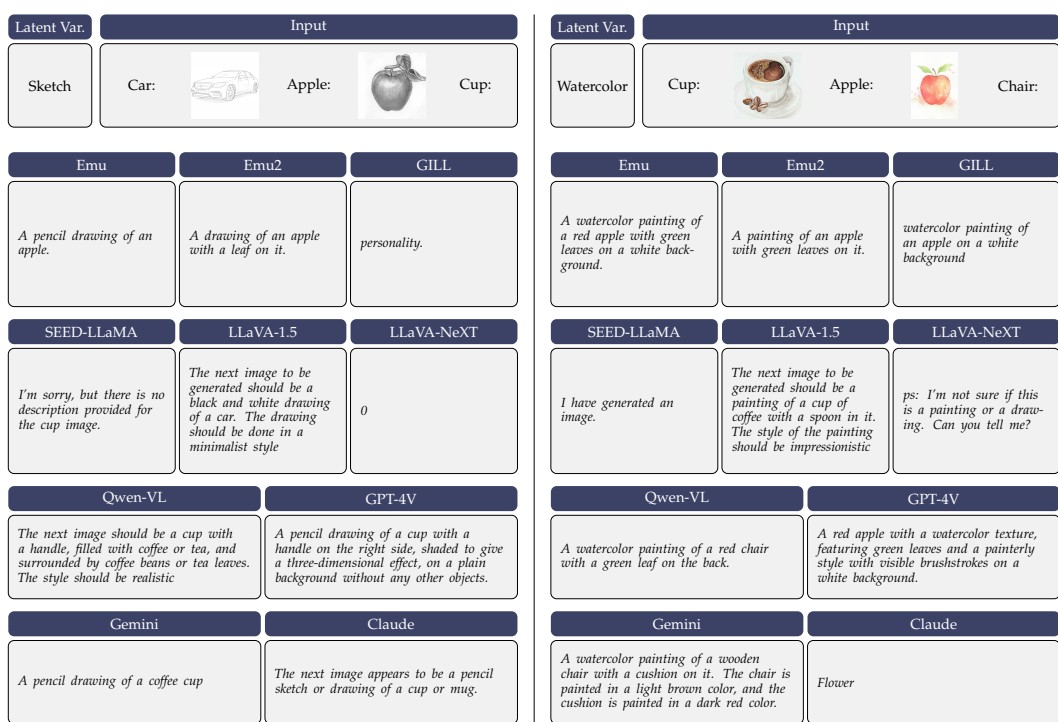

(b) Style-II

Figure 21: Examples of prompts and corresponding images description generated by MLLMs for tasks within the Style theme. For brevity, some lengthy responses from MLLMs have been truncated, retaining only the key parts of the responses.

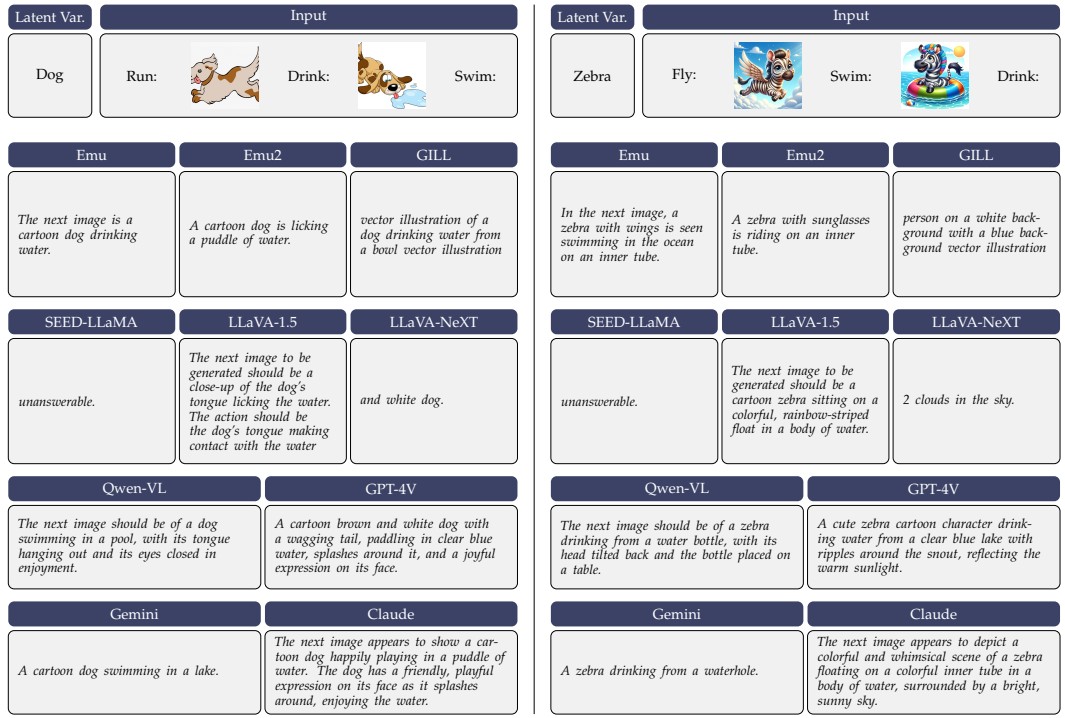

(a) Action-I

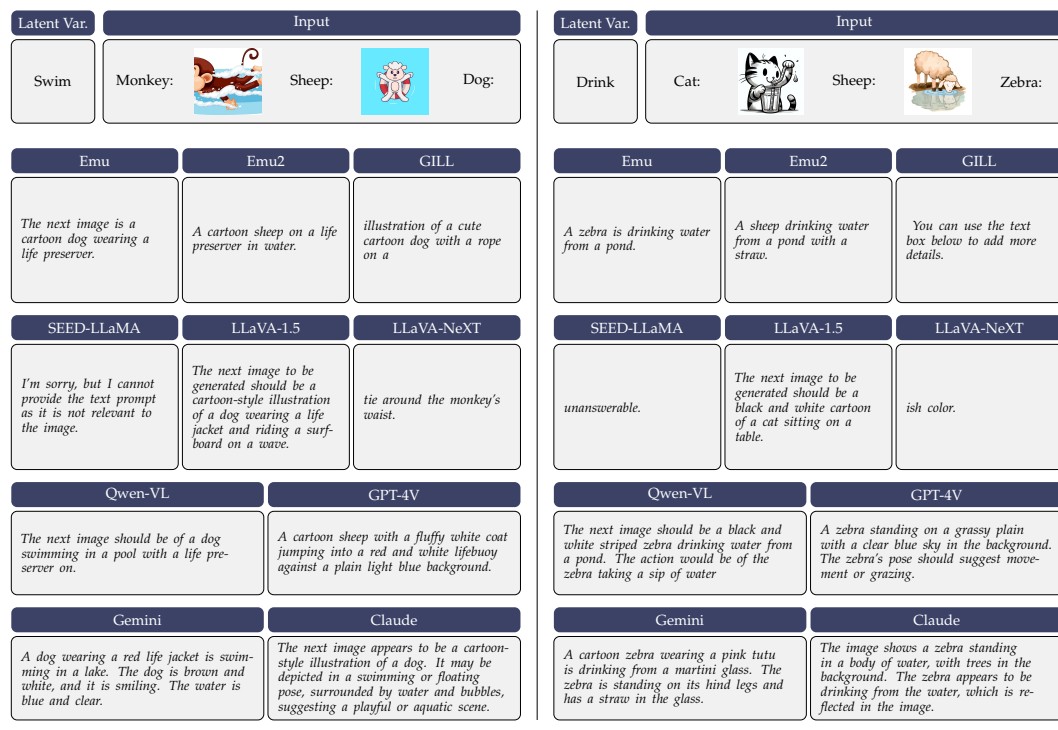

(b) Action-II

Figure 22: Examples of prompts and corresponding images description generated by MLLMs for tasks within the Action theme. For brevity, some lengthy responses from MLLMs have been truncated, retaining only the key parts of the responses.

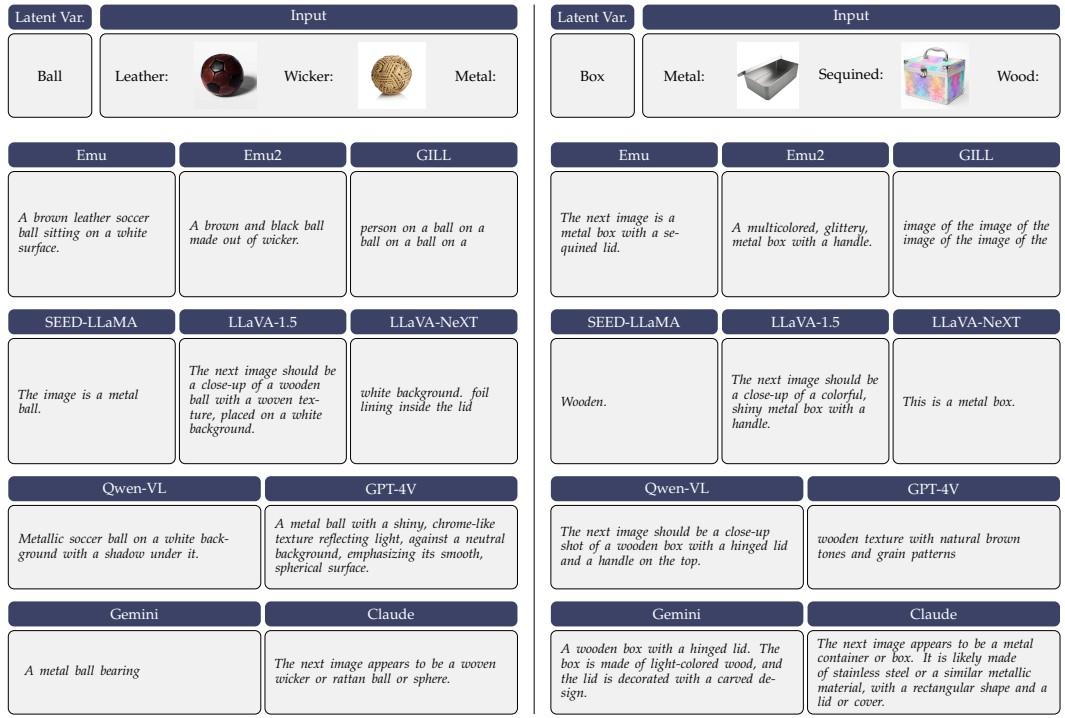

(a) Texture-I

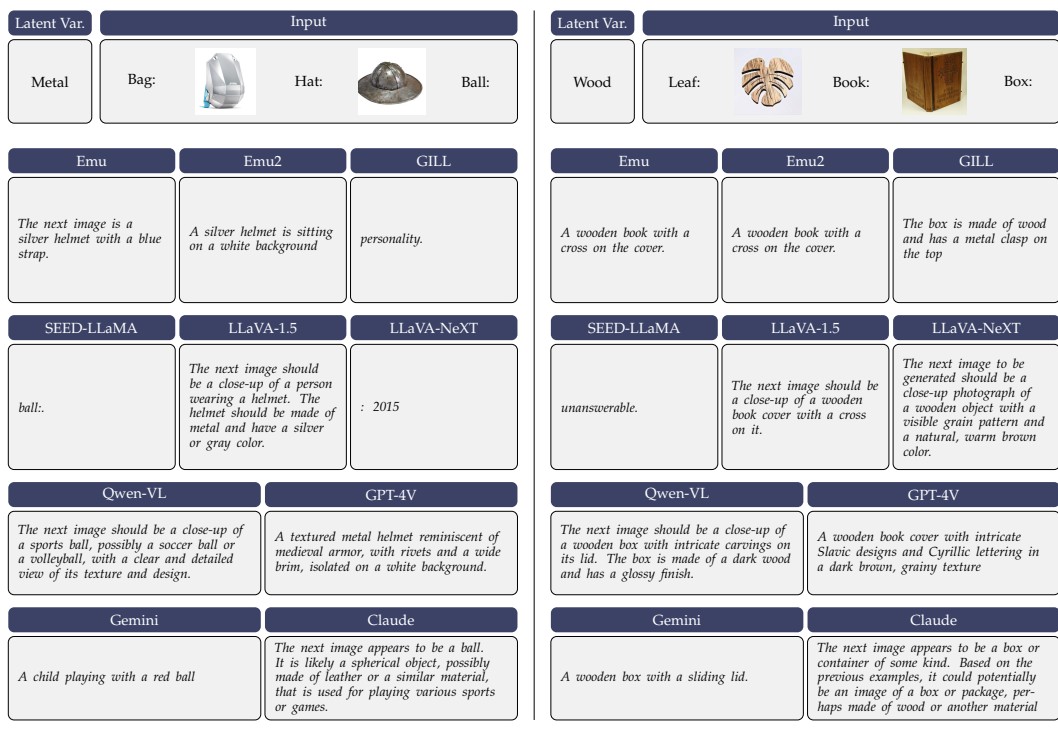

(b) Texture-II

Figure 23: Examples of prompts and corresponding images description generated by MLLMs for tasks within the Texture theme. For brevity, some lengthy responses from MLLMs have been truncated, retaining only the key parts of the responses.

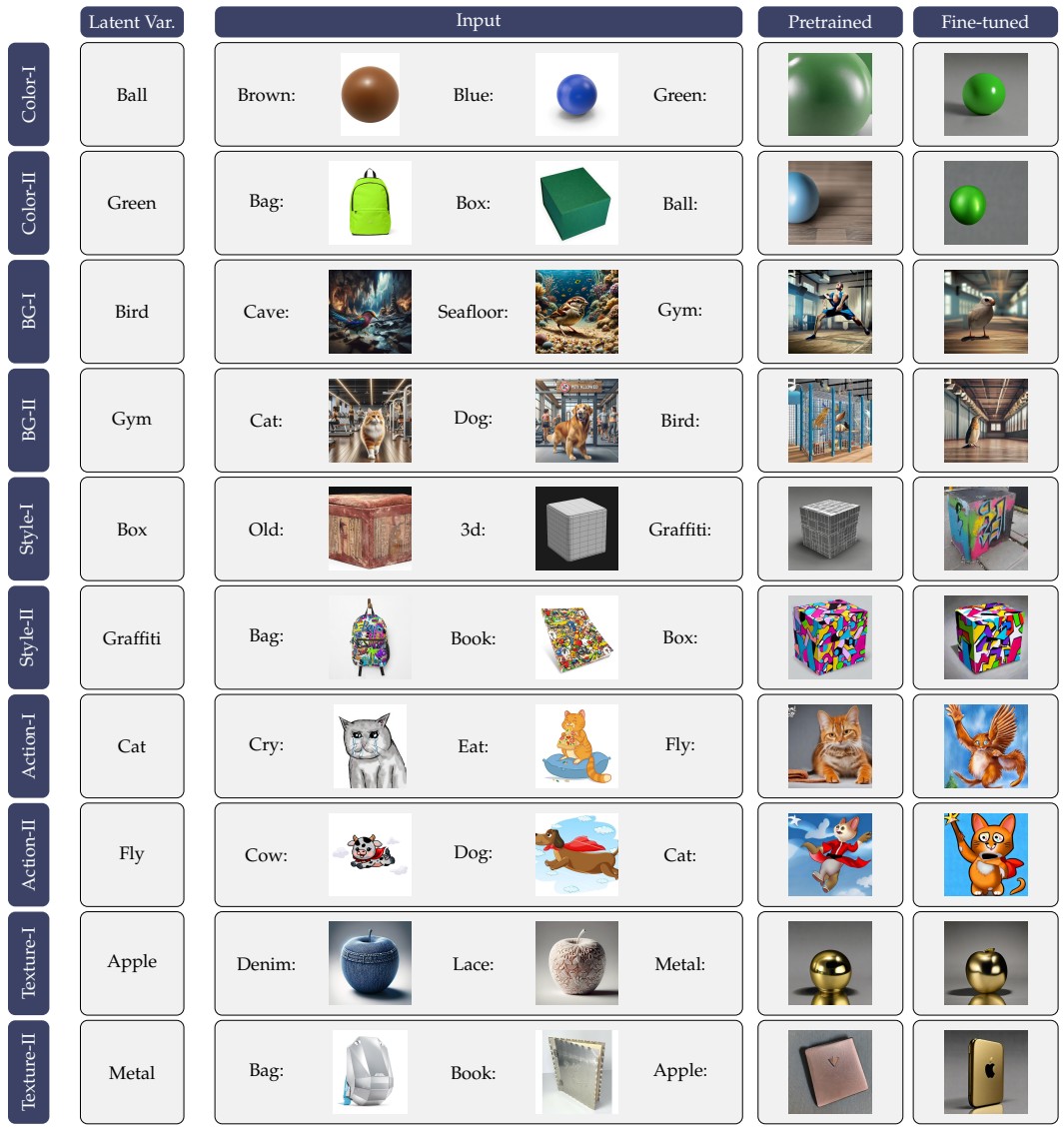

Figure 24: **Comparison between images generated by the pretrained-only SEED-LLaMA and its fine-tuned counterpart.** The shorthand 'BG' represents 'background'.

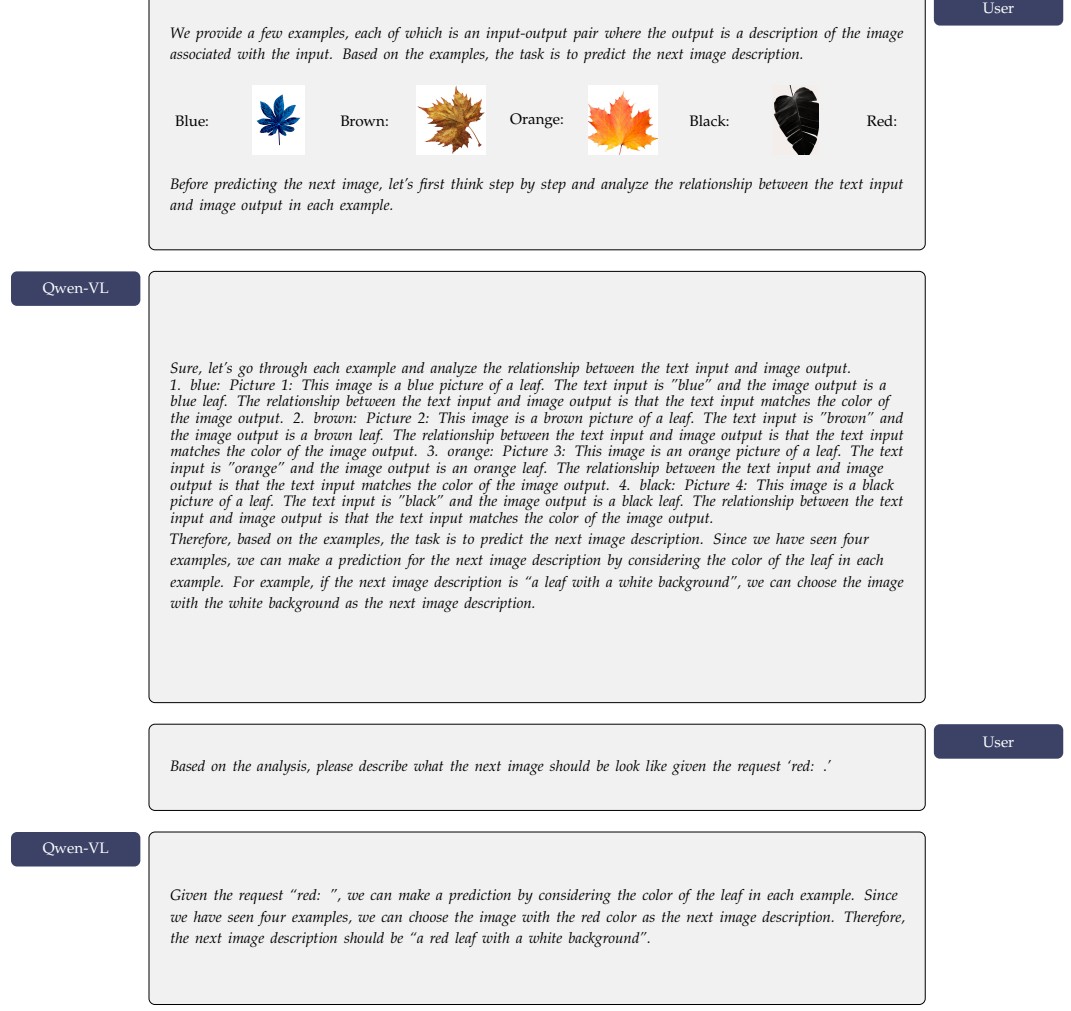

Figure 25: A four-shot example of outputs generated by Qwen-VL using CoT prompting for task Color-I.

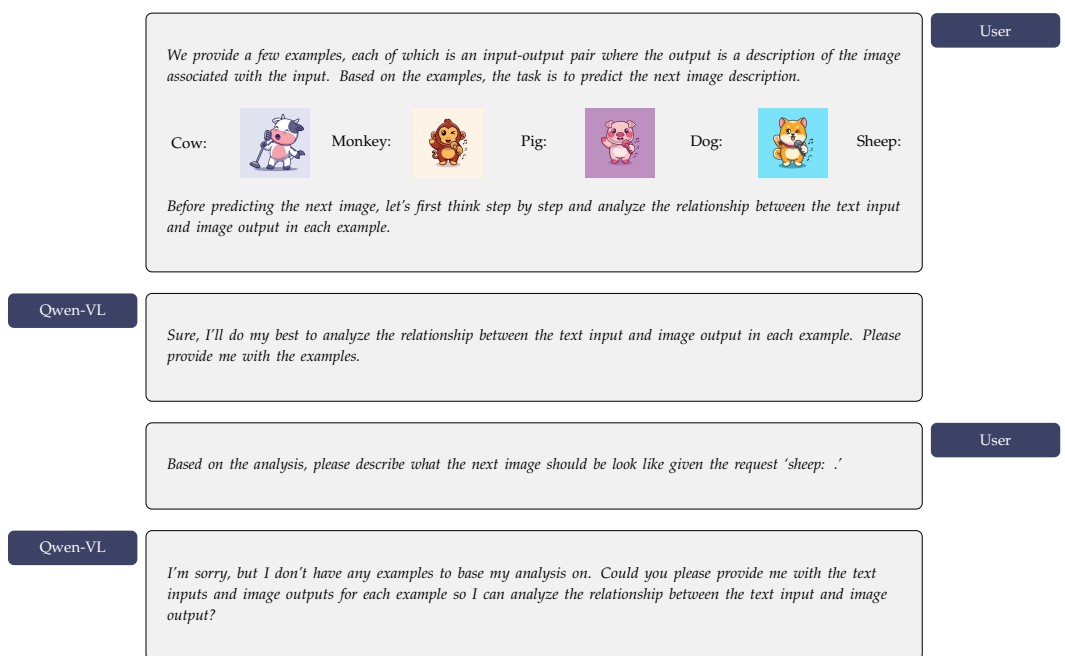

Figure 26: A four-shot example of outputs generated by Qwen-VL using CoT prompting for task Action-II.

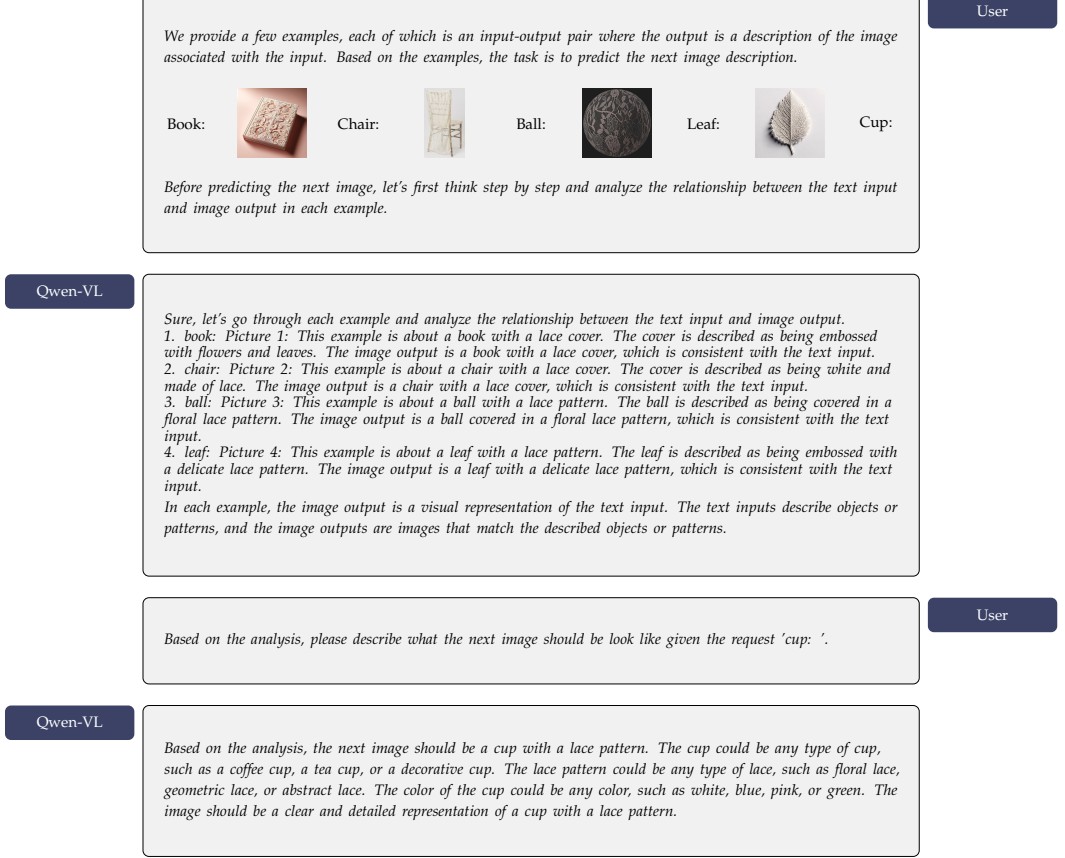

Figure 27: A four-shot example of outputs generated by Qwen-VL using CoT prompting for task Texture-II.

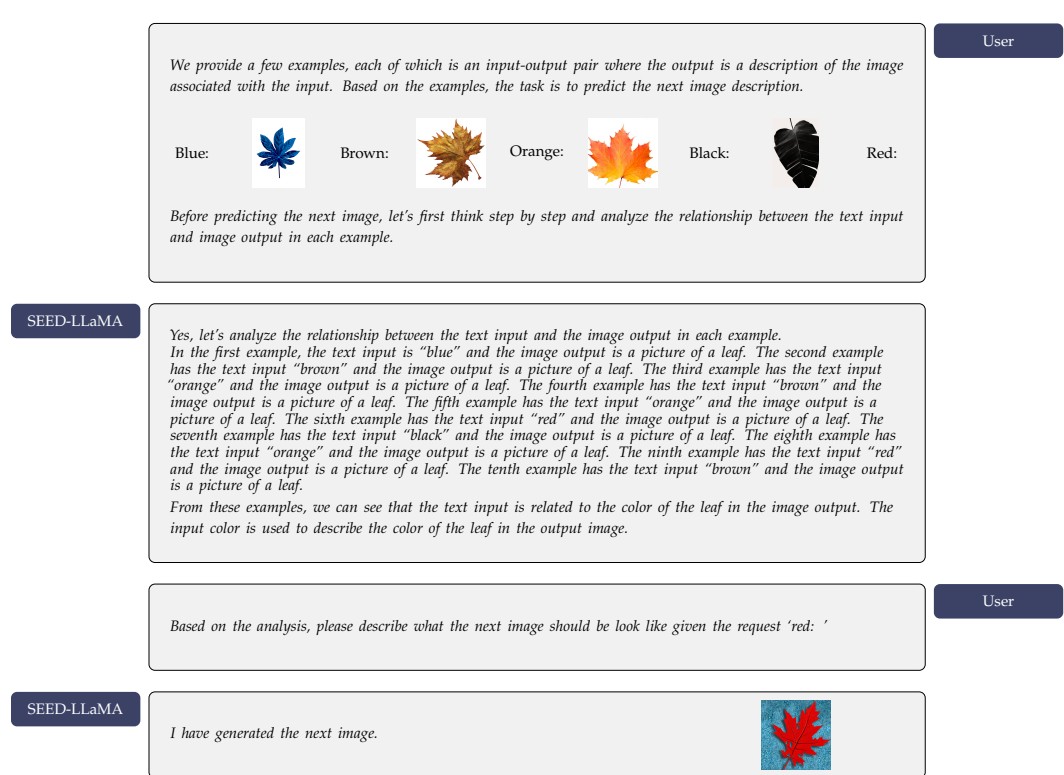

Figure 28: A four-shot example of outputs generated by SEED-LLaMA using CoT prompting for task Color-I.

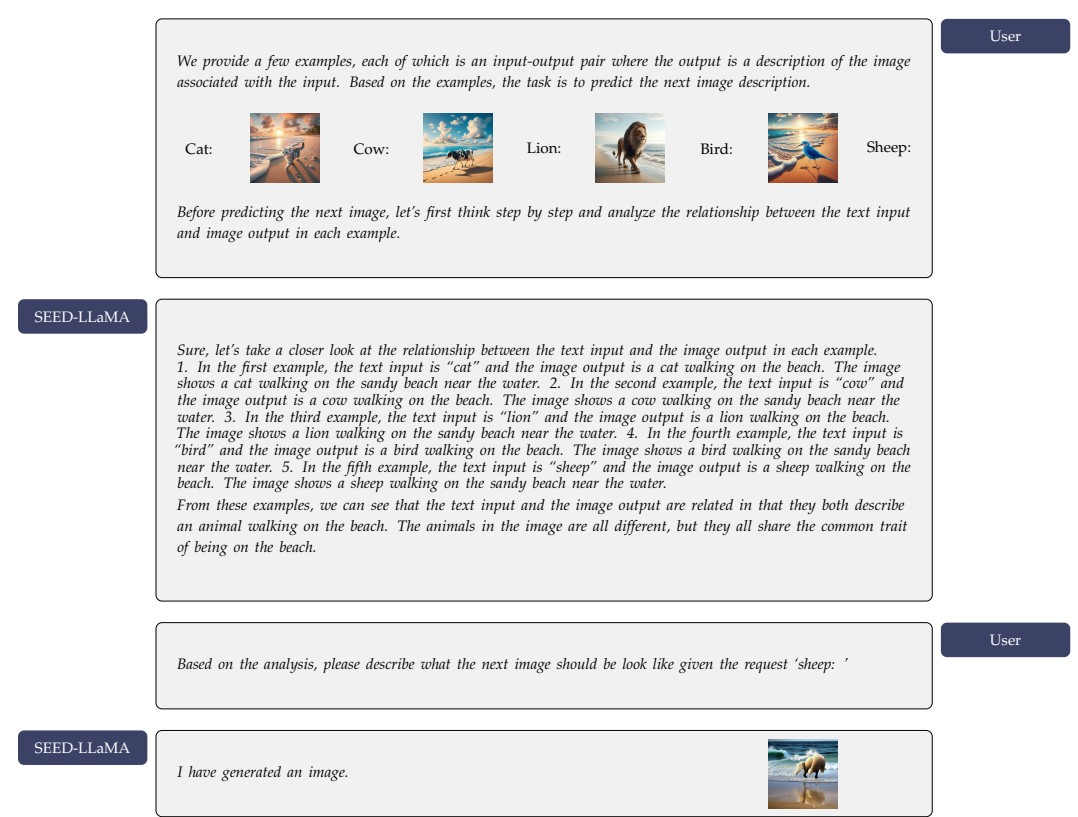

Figure 29: A four-shot example of outputs generated by SEED-LLaMA using CoT prompting for task Background-II.

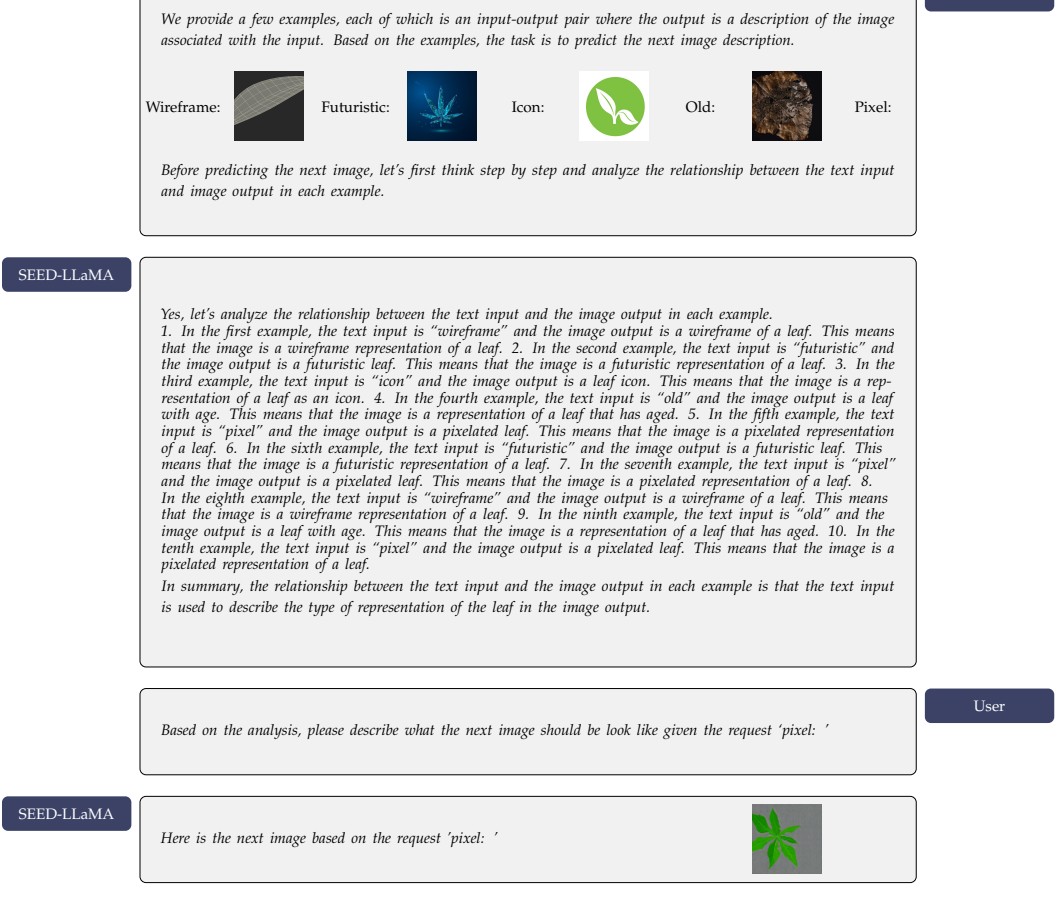

Figure 30: A four-shot example of outputs generated by SEED-LLaMA using CoT prompting for task Style-I.

