# OpenReview forum: "Can MLLMs Perform Text-to-Image In-Context Learning?"
_colmweb.org/COLM/2024/Conference — COLM_

### Official Review · Reviewer_8Cxt · 2024-04-29

**Rating:** 6
**Confidence:** 4
**Ethics Flag:** 1

**Summary:**

This paper identifies an important yet underexplored problem - T2I ICL. It introduces a benchmark for evaluating T2I-ICL ability of MLLMs. The authors introduce two approaches to measure the performance of MLLMs. For models with image generation ability, they evaluate the generated image under an I2T retrieval setting. For models that are only able to generate texts, they replace the image content with text descriptions.Finally, the authors identify the challenges of MLLMs performing T2I-ICL and provide two approaches to improve the performance.

**Questions To Authors:**

- In Figure 4(c) and 4(d), what is the potential reason of models without image generation
capability outperforming the models with such capability? On one hand, “capability tax” would be an interesting point, which means that a model capable of doing both may underperform a model that has a single capability. On the other hand, is it because the label itself is in textual modality, so performing a T2T evaluation would be an advantage for text-output-only models?

- To support your argument that “finetuning can enhance T2I-ICL ability”, it’s better to construct a test set which has distinct task types compared to the training set. However, the argument is still not strong and this finding still lacks excitement.
- The two challenges are not well solved by the two approaches the authors propose. While solving them is beyond the scope of this paper, it would be more insightful to discuss the potential solutions to the two challenges.

**Reasons To Accept:**

- The paper is well motivated. It identifies an important yet underexplored problem T2I-ICL.

- The settings of all experiments are well controlled and introduced, providing much soundness for this paper.

- The analysis in Section 5 seems reasonable and insightful.

- The paper is overall well written and easy to follow.

**Reasons To Reject:**

- As a benchmark, it would be better to test more models instead of just 6.

- The evaluation process is formulated as an Image-to-Text retrieval task via prompts (LLaVA) or embedding similarity (CLIP). But what if the generated image does not belong to any class in the predefined set? In this case, LLaVA may fail to follow instructions and thus fail to provide a valid evaluation result. Hence, it is a major concern that the authors do not discuss such cases.

- Texts in Figure 4 are almost unreadable.

- The first technique for enhancing T2I-ICL abilities is not surprising. ICL was an emergent ability of (M)LLMs to perform task learning without finetuning them. If we can finetune a model, then ICL would be needless.

---

> ### Author Rebuttal · Authors · 2024-05-31
>
> > Q: Insufficient MLLMs.
>
> We included nine MLLMs, detailing three (Emu2, LLaVA-1.5, and LLaVA-NeXT) in the appendix due to resource limitations. We then added Claude, bringing our total to ten MLLMs. The updated results are shown in [Figure 4(c,d)](https://hackmd.io/_uploads/H1wJmGvNC.png), though 8-shot results for Claude are not yet available.
>
> > Q: Experiments on fine-tuning.
>
> In response, we implemented a new leave-one-out setting for our experiments. We fine-tuned the model on all themed tasks except one (e.g., Texture-I and Texture-II) and tested on the excluded task to assess generalization.
>
> Below are the preliminary 2-shot results for SEED-LLaMA. Initial results indicate that fine-tuning various T2I-ICL tasks enhances the models' ability to perform T2I-ICL on unseen tasks, thus ICL remains necessary on unseen tasks for efficiency.
>
> | Fine-tuned SEED-LLaMA | Style-I  | Style-II | Texture-I | Texture-II |
> | -------- | -------- | -------- | -------- | -------- |
> | ✗ | .088 | .164 | .108 | .080 |
> | ✓ | **.208** | **.240** | **.200** | **.104** |
>
> > Q: Evaluation metric.
>
> To investigate, we manually reviewed 100 SEED-LLaMA-generated images, noting some ambiguities due to suboptimal quality. Despite this, our VQA system evaluations align strongly with human assessments, achieving an 89% concordance rate.
>
> > Q: Underperformance of image-generation MLLMs.
>
> The underperformance of MLLMs with image generation capabilities compared to those without is due to:
>
> * Image-generating MLLMs are newer and less mature, with smaller training data and model sizes compared to established models like GPT-4V.
>
> * They are primarily trained to generate images, which can hinder tasks like image description generation. For instance, SEED-LLaMA fails to generate text in our experiments.
>
> The performance gap is due to training disparities. With extensive training, image-generating MLLMs should match their text-only counterparts.
>
> > Q: Solution to two challenges.
>
> The main limitation is inadequate training data. We need large datasets that emphasize image interplay, which current VQA and image captioning datasets (used by Emu [[1]](https://arxiv.org/abs/2307.05222)) lack. Suitable sources include anime characters and product designs, where image interaction is crucial for context.
>
> > Q: Other.
>
> Fixed.
>
> ---
>
> **Final Note**: We sincerely thank you for your encouraging feedback and constructive suggestions which will significantly enhance our final paper.

---

> > ### Comment · Reviewer_8Cxt · 2024-06-07
> > **Thanks**
> >
> > Thanks for your response, I would like to keep the score as it is.

---

> > ### Author Response · Authors · 2024-06-07
> > **Thanks for your response. We now collected all experiment results.**
> >
> > Thank you for your response. We are here to provide the complete set of experimental results which we could not finish due to the time constraints during the rebuttal period.
> >
> > > Q. Insufficient MLLMs.
> >
> > Here we update the full results of Claude, including 8-shot. The updated figures are uploaded here: [Figure 4(c)](https://hackmd.io/_uploads/S1fF41lSC.png), [Figure 4(d)](https://hackmd.io/_uploads/HyutEyxBR.png). Claude's performance appears comparable to that of GPT-4V, while Gemini continues to outperform all the others.
> >
> > > Q. Experiments on fine-tuning.
> >
> > Below are the experimental results for SEED-LLaMA in both 2-shot and 4-shot scenarios. It is important to note that the the results for each theme are derived from different models. For example, for 2-shot, the results for Style-I and II come from models fine-tuned on all tasks except Style-I and II, while the results for Texture-I and II are from models fine-tuned on all tasks excluding Texture-I and II.
> >
> > | Shot | Fine-tuned SEED-LLaMA | Color-I | Background-I | Style-I | Action-I | Texture-I | Color-II | Background-II | Style-II | Action-II | Texture-II |
> > | -------- | -------- | -------- | -------- | -------- | -------- | -------- | -------- | -------- | -------- | -------- | -------- |
> > | 2 | ✗ | .636 | .292 | .088 | .196 | .108 | .360 | **.536** | .164 | **.196** | .080 |
> > | 2 | ✓ | **.752** | **.484** | **.208** | **.272** | **.200** | **.568** | .376 | **.240** | .180 | **.104** |
> > | 4 | ✗ | .612 | .360 | .092 | .044 | .048 | .380 | **.532** | **.140** | **.196** | .148 |
> > | 4 | ✓ | **.748** | **.556** | **.208** | **.256** | **.244** | **.616** | .488 | .112 | .132 | **.216** |
> >
> > After analyzing the complete set of results, our conclusion remains unchanged. The results indicate that fine-tuning on various T2I-ICL tasks generally enhances the models' ability to perform T2I-ICL on unseen tasks. Therefore, for unseen tasks that have not been included in the fine-tuning process, ICL remains necessary for efficiency.
> >
> > ---
> >
> > **Final Note**: Thanks again for providing insightful suggestions. All the discussions above will be included in our final version.

---

### Official Review · Reviewer_Nirr · 2024-05-08

**Rating:** 7
**Confidence:** 4
**Ethics Flag:** 1

**Summary:**

The paper proposes a new task, T2I-ICL where the goal is to generate an image given a demonstration. More precisely, the authors propose two tasks. In the first, the images define the object (e.g. a car, *object inference*) to generate, and in the second, they define the attribute (e.g. red, *attribute inference*) – and vice-versa for the text.

A dataset (CoBSAT) is proposed to evaluate such models, and the authors evaluate different types of models, namely (1) MLLMs than can generate images like Emu/Emu2, SEED-LLaMA, and GILL; (2) LLMs using only the textual description of images. The overall setting is clearly described, and the motivation – to analyze the limits of currents models in this simple setting, is interesting.

The evaluation of models is done automatically by using VQA systems, and show that object-inference tasks are more difficult for current state-of-the-art models.

The authors analyze why current MLLMs fail, i.e. infering attributes/objects from images and generating images are two difficult tasks and propose ways to improve the current models (by finetuning or using chain-of-thought).

Overall, the paper is interesting and well-constructed and demonstrates the limits of current MLLMs in a simple experimental setting.

**Reasons To Accept:**

- a new task with a new dataset
- the analysis of why current MLLMs fail

**Reasons To Reject:**

- the evaluation protocol is questionable (relying on a VQA system)
- the dataset might be too easy in the near future (as demonstrated when finetuning models on the data)

---

> ### Author Rebuttal · Authors · 2024-05-31
>
> We sincerely appreciate the reviewer’s encouraging feedback, particularly for recognizing our contributions in identifying a new task for MLLMs and providing benchmarks, as well as our analysis of the current limitations of these models. Our response is detailed below.
>
> ---
>
> > Q: The evaluation protocol is questionable (relying on a VQA system)
>
> To address the reviewer's concern, we manually labeled 100 images generated by SEED-LLaMA through T2I-ICL, selecting 10 random images from each task. It is important to note that some images were of suboptimal quality, presenting ambiguities that could be interpreted both as correct or incorrect. Despite these difficulties, our evaluations using the VQA system show strong alignment with human assessments, achieving a concordance rate of 89% (computed by the ratio of agreement between the two methods).
>
> > Q: The dataset might be too easy in the near future (as demonstrated when finetuning models on the data).
>
> In Sec. F.1.6, we have included a challenging version of our dataset, which incorporates misleading information in the prompts. This requires MLLMs to identify the correct information while disregarding the misleading content. Notably, almost all existing MLLMs struggle to perform well on this dataset. Based on your comment, we plan to highlight this further in the main body of our final version.
>
> Another way to increase the difficulty of our dataset is to incorporate more fine-grained attributes (e.g., dark red and light red in color-themed tasks) and more challenging themes (e.g., counting). Currently, these elements are excluded as our preliminary experiments indicate that existing MLLMs do not perform well on these more detailed tasks. We also identified this as a promising future research direction, which we have detailed in Sec. 8.
>
> ---
>
> **Final Notes**: We are delighted to hear that you appreciate our paper. We will incorporate these discussions into the revised version. Thank you once again for your insightful and encouraging feedback!

---

> > ### Comment · Reviewer_Nirr · 2024-06-05
> >
> > Thanks for your clarifications, it confirmed my overall rating of the paper.

---

> > > ### Author Response · Authors · 2024-06-07
> > >
> > > Thank you for reading our rebuttal and encouraging feedback. We will incorporate all of the above discussions in the final version of the paper.

---

### Official Review · Reviewer_u6Uf · 2024-05-12

**Rating:** 7
**Confidence:** 3
**Ethics Flag:** 1

**Summary:**

This paper proposes a new benchmark to evaluate MLLM's capability to perform in-context learning for text-to-image generation tasks，specifically it supports two types of visual inference tasks: object-inference and attribute-inference. The authors collect the images of the dataset from both internet resource and Generative model with a carefully designed object list and attribute lists to examine the MLLM's capability to properly generate the images or image descriptions that align with these proposed lists;
Finally, it benchmark both text-to-image generation model and closed source MLLM that can only generate text (e.g GPT4V) and identify some interesting insights: (1) The visual prompt post a significant challenge for in-context learning of the current MLLM; (2) The Image generation task is a much harder task than text-generation even when the required attributes are relatively constrained;
The author than proposed two simple solutions to address these two challenges including: chaini-of-thoughts & finetuning, both lead to obvious improvements.

**Reasons To Accept:**

1. A new benchmark that aims to evaluate MLLM's capability to perform in-context learning for text-to-image tasks, which was previously under-explored.
2. Through this benchmark, the author identifies two major challenge of the current MLLM: (1) Lack of capability to understand the visual prompt in the in-context learning examples (2)limited capability to generate image compared to the text generation capability.
3. Comprehensively compare multiple models on this new benchmark and identify the challenges the current methods encounter. It also proposes two effective method: fine-tuning and chain-of-thoughts to improve the model's capability of in-context learning.

**Reasons To Reject:**

1. Some of the benchmark images come from generative models (e.g., Dalle-E). The qualities of these generated images are questionable.
2. The paper mainly leverage LAVVA to evaluate the performance of the generated image or the generated caption of the image. However, given that LAVVA is a relatively worse model compared to some of the closed model (GPT-4V and Gemini), the evaluation results of LAVVA over these strong MLLM is less reliable.

---

> ### Author Rebuttal · Authors · 2024-05-31
>
> We sincerely thank the reviewer for acknowledging our contribution in i) a new benchmark for T2I-ICL that was underexplored, ii) identifying the challenges, and iii) suggesting effective methods. Your concerns are addressed below.
>
> ---
>
> > Q: The qualities of benchmark images are questionable.
>
> We avoided this issue through meticulous data collection. We manually scrutinized and selected each image in our dataset, regardless of whether they were sourced from the web or produced by generative models. We ensured that every image aligned with the intended object and attribute specifications.  Throughout the image collection process, we ensured all images in our dataset were of sufficient clarity to allow state-of-the-art MLLMs to identify their objects and attributes with 100% accuracy.
>
> > Q: The paper mainly leverages LLAVA to evaluate the performance. However, given that LLAVA is a relatively worse model compared to some of the closed models, the evaluation results of LLAVA over these strong MLLMs are less reliable.
>
> We appreciate the reviewer's insightful question. We believe in LLaVA's reliability for several reasons:
>
> 1. Performing T2I-ICL and visual question answering (VQA) requires different capabilities of MLLMs—T2I-ICL necessitates a deep understanding of the interplay between images, whereas VQA does not. Our evaluation primarily requires VQA capability, and LLaVA is well-known for excelling in visual question answering.
> 2. LLaVA achieves 100% accuracy in identifying the attributes and objects within our dataset.
>
> We also incorporated Gemini as an additional evaluation model. Our comparative analysis visualized in this [figure](https://hackmd.io/_uploads/r1RD30LV0.jpg) shows that the accuracy estimates from LLaVA align closely with those from Gemini. This correlation reinforces our confidence in LLaVA as a reliable and accessible open-source evaluation alternative to these closed-source models.
>
> Furthermore, we manually labeled 100 images generated by SEED-LLaMA through T2I-ICL, selecting 10 random images from each task to serve as a baseline. Our evaluations using the LLaVA show strong alignment with human assessments, achieving a concordance rate of 89%.
>
> ---
>
> **Final Note**: Thank you again for your encouraging feedback and great questions. If you have any remaining questions, please do not hesitate to let us know. If our responses have resolved your concerns, we kindly request you consider increasing your score and championing our paper.

---

### Official Review · Reviewer_6kcQ · 2024-05-12

**Rating:** 6
**Confidence:** 4
**Ethics Flag:** 1

**Summary:**

Quality:
The work presented is of high quality, demonstrating rigorous experimental design and thoughtful analysis of results. The introduction of the CoBSAT benchmark to systematically assess T2I-ICL capabilities across a range of MLLMs is a commendable effort. The methodology section is detailed and includes a clear explanation of the evaluation pipeline, which adds to the reliability of the findings.

Clarity:
The paper is well-structured and written with a clear narrative that guides the reader through the introduction of the problem, the development of the CoBSAT benchmark, experimentation, and conclusions. Each section builds logically on the last, with sufficient detail provided for replication of the experiments. Figures and examples are used effectively to illustrate key points, enhancing the overall readability and understanding.

Originality:
This work addresses a gap in the research on multimodal in-context learning by focusing on text-to-image tasks, which is less explored compared to image-to-text configurations. The creation of a dedicated benchmark for T2I-ICL is original and provides valuable resources to the research community. The exploration of fine-tuning and Chain-of-Thought prompting to enhance MLLM performance in T2I-ICL also contributes new insights into effective strategies for multimodal learning.

Significance:
The significance of this work lies in its potential to advance the field of multimodal learning. By identifying challenges and proposing methods to enhance T2I-ICL, the paper sets the groundwork for further developments in this area. The findings could influence future research directions and have practical implications for applications in product design, personalized content creation, and more. The availability of the CoBSAT dataset and the detailed analysis of MLLM performance also make significant contributions to ongoing discussions and research into multimodal AI systems.

**Questions To Authors:**

1. How do you ensure the diversity and representativeness of the CoBSAT dataset? Given the challenges of bias in AI, what measures were taken to ensure that the dataset does not reinforce stereotypical biases, particularly in the context of generating images from text?

2. Beyond the use of CLIP and LLaVA for validating the generated images, what other quantitative metrics were considered or used to assess the performance of the MLLMs on the CoBSAT benchmark? Please provide details on the statistical significance of the results, if applicable.

**Reasons To Accept:**

1. Significant Contribution to Multimodal Learning: The paper addresses a notably underexplored area in multimodal learning by focusing on Text-to-Image In-Context Learning (T2I-ICL). This study not only extends the current understanding of In-Context Learning (ICL) but also adapts it to a more challenging and less studied multimodal context, which is significant for the field.

2. Introduction of CoBSAT Benchmark: The creation of the CoBSAT benchmark is a major contribution of this work. It systematically assesses the T2I-ICL capabilities of Multimodal Large Language Models (MLLMs) across a variety of tasks. This benchmark will likely become a valuable resource for future research in the area.

**Reasons To Reject:**

1. Generalizability and Practical Relevance: The scenarios tested appear highly specific and might not generalize well to other text-to-image contexts. The practical applications described (e.g., product design, personalized content creation) are mentioned only briefly without exploring the operational challenges or effectiveness in real-world scenarios.

2. In summary, while the paper addresses an innovative and relevant topic, these concerns suggest it may not yet meet the high standards required for publication at COLM. Addressing these issues could potentially make the paper suitable for publication.

3. Too much content has been placed in the appendices.

---

> ### Author Rebuttal · Authors · 2024-05-31
>
> We are encouraged that you found our paper to be of high quality, well-structured, well-motivated, significant, and well-written.
>
> ---
>
> > Q: Generalizability and Practical Relevance.
>
> We intentionally designed our dataset for a simplified setting because:
>
> 1. It was unclear if existing models could perform T2I-ICL.
> 2. Practical applications are hard to evaluate; for example, no method exists to automatically assess generated anime character designs.
> 3. Current MLLMs generate unsatisfactory results even in simplified settings (Fig4a,b).
>
> > Q: Diversity, representativeness and stereotypical biases of dataset.
>
> To ensure the CoBSAT dataset's diversity and representativeness, we selected key image attributes: color, background, style, action, and texture, as supported by related works ([[1]](https://arxiv.org/abs/2208.12242), Figure 5 in [[2]](https://arxiv.org/abs/2310.01506)).
>
> While our dataset currently avoids stereotypical biases by not involving demographic groups, future expansions to include more attributes could introduce such challenges, which we acknowledge but are beyond our present scope.
>
> > Q: Other metrics.
>
> Evaluating images is inherently challenging. Initially, we only used CLIP and LLaVA for validating the generated images, and we validated the performance of CLIP and LLaVA using two methods:
>
> 1. LLaVA correctly identified all attributes and objects in our dataset with 100% accuracy.
> 2. In Sec. E, we showed a strong correlation between CLIP and LLaVA evaluations, validating LLaVA's consistency with CLIP and its systematic, rather than arbitrary, assessments.
>
> Following your suggestion, we included Qwen-VL and Gemini in our evaluation. We manually labeled 100 SEED-LLaMA-generated images. LLaVA's evaluations showed strong alignment with human assessments, and Gemini performed the best.
>
> | | CLIP | LLaVA | Qwen-VL | Gemini |
> |---|---|---|---|---|
> | Concordance Rate to Human Evaluation | .85 | .89 | .78 | **.92** |
>
> We conducted a statistical study with 20,000 images to compare automatic metrics, focusing on their relation to Gemini's results, which align closely with human evaluations. The [figure](https://hackmd.io/_uploads/BJJCTxw4C.jpg) shows a strong correlation between LLaVA's and Gemini's accuracy estimates, reinforcing our confidence in LLaVA as a reliable, open-source evaluation model.
>
> > Q: Long appendices.
>
> Will update.
>
> ---
>
> **Final Note**: Thanks for the suggestions. We will incorporate all discussions above in our final version.

---

### Decision · Program_Chairs · 2024-07-10

**Decision:**

Accept

**Comment:**

The work examines text-to-image in-context learning and introduces a benchmark to assess this capability of MLLM. All reviewers agree to accept the submission. It's a nice contribution to have a new benchmark dataset for the problem. As the release of gpt-4o, there will be more papers that are interested in this capability.